# Spectrally Transformed Kernel Regression

**Runtian Zhai, Rattana Pukdee, Roger Jin, Maria-Florina Balcan, Pradeep Ravikumar**
Carnegie Mellon University
`{rzhai,rpukdee,rrjin,ninamf,pradeepr}@cs.cmu.edu`

## Abstract

Unlabeled data is a key component of modern machine learning. In general, the role of unlabeled data is to impose a form of smoothness, usually from the similarity information encoded in a base kernel, such as the $\epsilon$-neighbor kernel or the adjacency matrix of a graph. This work revisits the classical idea of spectrally transformed kernel regression (STKR), and provides a new class of general and scalable STKR estimators able to leverage unlabeled data. Intuitively, via spectral transformation, STKR exploits the data distribution for which unlabeled data can provide additional information. First, we show that STKR is a principled and general approach, by characterizing a universal type of "target smoothness", and proving that any sufficiently smooth function can be learned by STKR. Second, we provide scalable STKR implementations for the inductive setting and a general transformation function, while prior work is mostly limited to the transductive setting. Third, we derive statistical guarantees for two scenarios: STKR with a known polynomial transformation, and STKR with kernel PCA when the transformation is unknown. Overall, we believe that this work helps deepen our understanding of how to work with unlabeled data, and its generality makes it easier to inspire new methods.

## 1 Introduction

The past decade has witnessed a surge of new and powerful algorithms and architectures for learning representations (Vaswani et al., 2017; Devlin et al., 2019; Chen et al., 2020; He et al., 2022); spurred in part by a boost in computational power as well as increasing sizes of datasets. Due to their empirical successes, providing an improved theoretical understanding of such representation learning methods has become an important open problem. Towards this, a big advance was made recently by HaoChen et al. (2021), who showed that when using a slight variant of popular contrastive learning approaches, termed spectral contrastive learning, the optimal learnt features are the top-$d$ eigenfunctions of a population augmentation graph. This was further extended to other contrastive learning approaches (Johnson et al., 2023; Cabannes et al., 2023), as well as more generally to all augmentation-based self-supervised learning methods (Zhai et al., 2024).

A high-level summary of this recent line of work is as follows: The self-supervised learning approaches implicitly specify inter-sample similarity encoded via a Mercer *base kernel*. Suppose this kernel has the spectral decomposition $K(x, x') = \sum_{i=1}^{\infty} \lambda_i \psi_i(x) \psi_i(x')$, where $\lambda_1 \geq \lambda_2 \geq \cdots \geq 0$. The above line of work then showed that recent representation learning objectives can learn the optimal $d$ features, which are simply the top-$d$ eigenfunctions $[\psi_1, \cdots, \psi_d]$ of this base kernel. Given these $d$ features, a "linear probe" is learned atop via regression. It can be seen that this procedure is equivalent to kernel regression with the truncated kernel $K_d(x, x') = \sum_{i=1}^{d} \lambda_i \psi_i(x) \psi_i(x')$. More generally, one can extend this to regression with a *spectrally transformed kernel* (STK) $K_s(x, x') = \sum_{i=1}^{\infty} s(\lambda_i) \psi_i(x) \psi_i(x')$, where $s : [0, +\infty) \to [0, +\infty)$ is a general transformation function. We call this generalized method *spectrally transformed kernel regression* (STKR). Then, $K_d$ amounts to an STK with the "truncation function" $s(\lambda_i) = \lambda_i \mathbf{1}_{\{i \leq d\}}$.

In fact, STK and STKR were quite popular two decades ago in the context of semi-supervised learning, which similar to more recent representation learning approaches, aims to leverage unlabeled data. Their starting point again was a base kernel encoding inter-sample similarity, but unlike recent representation learning approaches, at that period this base kernel was often explicitly rather than implicitly specified. For manifold learning this was typically the $\epsilon$-neighbor or the heat kernel (Belkin & Niyogi, 2003). For unlabeled data with clusters, this was the cluster kernel (Chapelle et al., 2002).

And for graph structured data, this was typically the (normalized) adjacency or Laplacian matrix of an explicitly specified adjacency graph (Chung, 1997; Belkin & Niyogi, 2003). A range of popular approaches then either extracted top eigenfunctions, or learned kernel machines. These methods include LLE (Roweis & Saul, 2000), Isomap (Tenenbaum et al., 2000), Laplacian eigenmap (Belkin & Niyogi, 2003) for manifold learning; spectral clustering (Ng et al., 2001) for clustered data; and label propagation (Zhu & Ghahramani, 2002; Zhou et al., 2003) for graph structured data. With respect to kernel machines, Bengio et al. (2004) linked these approaches to kernel PCA, and Chapelle et al. (2002); Smola & Kondor (2003); Zhu et al. (2006) proposed various types of STK.

In this work, we revisit STK and STKR, and provide three sets of novel results. Our first contribution is elevating STKR to be *a principled and general way of using unlabeled data*. Unlabeled data is useful as it provides additional information about the data distribution $P_\mathcal{X}$, but the kernel could be independent of $P_\mathcal{X}$. STKR implicitly mixes the information of $P_\mathcal{X}$ and the kernel in the process of constructing the STK. We then prove the generality of STKR via an existence result (Theorem 1): Suppose the target function satisfies a certain unknown "target smoothness" that preserves the relative smoothness at multiple scales, then there must exist an STK that describes this target smoothness.

Our second contribution is implementing STKR with *general transformations* for the *inductive setting*. Most prior work is limited to the transductive setting where test samples are known at train time (Zhou et al., 2003; Johnson & Zhang, 2008), in large part because it is easier to carry out spectral transformation of the finite-dimensional Gram matrix than the entire kernel function itself. But for practical use and a comprehensive analysis of STKR, we need inductive approaches as well. Towards this, Chapelle et al. (2002) solved an optimization problem for each test point, which is not scalable; Chapelle et al. (2006, Chapter 11.4) provided a more scalable extension that "propagates" the labels to unseen test points after transductive learning, but they still needed to implicitly solve a quadratic optimization program for each set of test points. These approaches moreover do not come with strong guarantees. Modern representation learning approaches that use deep neural networks to represent the STK eigenfunctions inductively do provide scalable approaches, but no longer have rigorous guarantees. To the best of our knowledge, this work develops the first inductive STKR implementation that (a) has closed-form formulas for the predictor, (b) works for very general STKs, (c) is scalable, and importantly, (d) comes with strong statistical guarantees. We offer detailed implementations with complexity analysis, and verify their efficiency with experiments on real tasks in Section 5.

Our third contribution is developing rigorous theory for this general inductive STKR, and proving nonparametric statistical learning bounds. Suppose the target function $f^*$ is smooth *w.r.t.* an STK $K_s$, and there are $n$ labeled and $m$ unlabeled samples both *i.i.d.*. We prove estimation and approximation error bounds for the STKR predictor (in $L^2$ norm) when $s(\lambda)$ is known or completely unknown. By incorporating recent theoretical progress, three of our four bounds have tightness results.

In a nutshell, this work conceptually establishes STKR as a general and principled way of learning with labeled and unlabeled data together with a similarity base kernel; algorithmically we provide scalable implementations for general inductive STKR, and verify them on real datasets; statistically we prove statistical guarantees, with technical improvements over prior work. Limitations and open problems are discussed in Section 6, and more related work can be found in Appendix A. We also provide a table of notations at the beginning of the Appendix for the convenience of our readers.

## 2 DERIVING STKR FROM DIFFUSION INDUCED MULTISCALE SMOOTHNESS

Let the input space $\mathcal{X}$ be a compact Hausdorff space, $\mathcal{Y} = \mathbb{R}$ be the label space, and $P_{\mathcal{X}\mathcal{Y}}$ be the underlying data distribution over $\mathcal{X} \times \mathcal{Y}$, whose marginal distribution $P_\mathcal{X}$ is a Borel measure with support $\mathcal{X}$. We will use the shorthand $dp(x)$ to denote $dP_\mathcal{X}(x)$. Let $L^2(P_\mathcal{X})$ be the Hilbert space of $L^2$ functions *w.r.t.* $P_\mathcal{X}$ that satisfy $\int f(x)^2 dp(x) < +\infty$, with $\langle f_1, f_2 \rangle_{P_\mathcal{X}} = \int f_1(x) f_2(x) dp(x)$ and $\|f\|_{P_\mathcal{X}} = \sqrt{\langle f, f \rangle_{P_\mathcal{X}}}$. $f \in L^2(P_\mathcal{X})$ also implies $f \in L^1(P_\mathcal{X})$, which guarantees that $\mathbb{E}_{X \sim P_\mathcal{X}}[f(X)]$ exists and is finite. Let a base kernel $K(x, x')$ encode inter-sample similarity information over $\mathcal{X}$. We assume full access to $K$ (*i.e.* we can compute $K(x, x')$ for all $x, x'$), and that $K$ satisfies:

(i) $K$ is a Mercer kernel, so it has the spectral decomposition: $K(x, x') = \sum_{i=1}^{\infty} \lambda_i \psi_i(x) \psi_i(x')$, where the convergence is absolute and uniform. Here $\lambda_i, \psi_i$ are the eigenvalues and orthonormal eigenfunctions of the integral operator $T_K : L^2(P_\mathcal{X}) \to L^2(P_\mathcal{X})$ defined as $(T_K f)(x) = \int f(x') K(x, x') dp(x')$, such that $\lambda_1 \geq \lambda_2 \geq \cdots \geq 0$, and $\langle \psi_i, \psi_j \rangle_{P_\mathcal{X}} = \delta_{i,j} = \mathbf{1}_{\{i=j\}}$.

(ii) $K$ is *centered*: Defined as $T_K \mathbf{1} = \mathbf{0}$, where $\mathbf{1}(x) \equiv 1$ and $\mathbf{0}(x) \equiv 0$. One can center any $K$ by $\tilde{K}(x_0, y_0) = K(x_0, y_0) - \int K(x, y_0) dp(x) - \int K(x_0, y) dp(y) + \iint K(x, y) dp(x) dp(y)$.

**Why assuming centeredness?** In this work, **we view the smoothness and scale of a function $f$ as two orthogonal axes**, since our smoothness pertains to the inter-sample similarity. Thus, *we view $f_1$ and $f_2$ as equally smooth if they differ by a constant a.e.*. If $K$ is not centered, then this will not be true under the RKHS norm (see Section 2.1). In practice centering is not a necessary step, though often recommended in kernel PCA.

This work investigates the regression function estimation problem in nonparametric regression, with error measured in $L^2$ norm (see Györfi et al. (2002) for an introduction of regression problems):

> **Problem.** Let $f^*(x) := \int y \, dP_{\mathcal{X}\mathcal{Y}}(y|x) \in L^2(P_{\mathcal{X}})$ be the *target regression function*. Given $n$ labeled samples $(x_1, y_1), \cdots, (x_n, y_n) \overset{i.i.d.}{\sim} P_{\mathcal{X}\mathcal{Y}}$, $m$ unlabeled samples $x_{n+1}, \cdots, x_{n+m} \overset{i.i.d.}{\sim} P_{\mathcal{X}}$, and access to $K(x, x')$ for any $x, x' \in \mathcal{X}$, find a predictor $\hat{f} \in L^2(P_{\mathcal{X}})$ with low *prediction error*:
> $$\mathrm{err}(\hat{f}, f^*) := \mathbb{E}_{X \sim P_{\mathcal{X}}}\left[\left(\hat{f}(X) - f^*(X)\right)^2\right] = \left\|\hat{f} - f^*\right\|^2_{P_{\mathcal{X}}}.$$

One can also think of $f^*$ as the target function, and $y = f^*(x) + \epsilon$, where $\epsilon$ is random noise with zero mean. Let $\{\lambda_i : i \in \mathbb{I}\}$ be the set of non-zero eigenvalues of $T_K$, then define $K^p(x, x') := \sum_{i \in \mathbb{I}} \lambda_i^p \psi_i(x) \psi_i(x')$ for all $p \in \mathbb{R}$, which corresponds to an STK with $s(\lambda) = \lambda^p$. The set $\{K^p\}$ delineates a *diffusion process w.r.t. $K$*, because $K^{p+1}(x, x') = \int K^p(x, x_0)K(x', x_0)dp(x_0)$, so that $K^{p+1}$ captures similarity with one additional hop to $K^p$. For continuous diffusion, $p$ can be real-valued. Then, the *reproducing kernel Hilbert space (RKHS)* associated with $K^p$ for any $p \geq 1$ is:

$$\mathcal{H}_{K^p} := \left\{ f = \sum_{i \in \mathbb{I}} u_i \psi_i \, \middle| \, \sum_i \frac{u_i^2}{\lambda_i^p} < \infty \right\}, \quad \left\langle \sum_i u_i \psi_i, \sum_i v_i \psi_i \right\rangle_{\mathcal{H}_{K^p}} = \sum_i \frac{u_i v_i}{\lambda_i^p}, \quad (1)$$

and $\|f\|^2_{\mathcal{H}_{K^p}} = \langle f, f \rangle_{\mathcal{H}_{K^p}}$. $K^p$ is the reproducing kernel of $\mathcal{H}_{K^p}$, as one can verify for all $f \in \mathcal{H}_{K^p}$ and $x$ that $\langle f, K^p_x \rangle_{\mathcal{H}_{K^p}} = f(x)$, for $K^p_x(z) := K^p(x, z)$. $\mathcal{H}_{K^1}$ is also denoted by $\mathcal{H}_K$. $\mathcal{H}_{K^p}$ are called power spaces (Fischer & Steinwart, 2020) or interpolation Sobolev spaces (Jin et al., 2023). *Kernel ridge regression* (KRR) is a classical least-squares algorithm. KRR with $K$ is given by:

$$\hat{f} \in \arg\min_{f \in \mathcal{H}_K} \left\{ \frac{1}{n} \sum_{i=1}^n (f(x_i) - y_i)^2 + \beta_n \|f\|^2_{\mathcal{H}_K} \right\}$$

for some $\beta_n > 0$. Although KRR is very widely used, the problem is that it does not leverage the unlabeled data, because the optimal solution of KRR only depends on $x_1, \cdots, x_n$ but not $x_{n+1}, \cdots, x_{n+m}$, as is explicitly shown by the Representer Theorem (Schölkopf & Smola, 2002, Theorem 4.2): *All minimizers of KRR admit the form $\hat{f}^*(x) = \sum_{j=1}^n \alpha_j^* K(x, x_j)$, where*

$$\boldsymbol{\alpha}^* \in \arg\inf_{\boldsymbol{\alpha} \in \mathbb{R}^n} \left\{ \frac{1}{n} \sum_{i=1}^n \left[ \sum_{j=1}^n \alpha_j K(x_i, x_j) - y_i \right]^2 + \beta_n \sum_{i,j=1}^n \alpha_i \alpha_j K(x_i, x_j) \right\}. \quad (2)$$

One consequence is that for KRR, the whole base kernel could be useless. Consider the graph example on the right, where only the three shaded nodes are labeled, and $K$ is the adjacency matrix. With KRR, the unlabeled nodes are useless and can be removed. Then, the graph becomes three isolated nodes, so it has zero impact on the learned predictor.

Figure 1: Sample graph.

## 2.1 Diffusion Induced Multiscale Smoothness

Let us use this graph example to motivate STKR. Unlabeled samples are useful as they offer more information about the marginal distribution $P_{\mathcal{X}}$. The problem is that we don't know the connection between $K$ and $P_{\mathcal{X}}$. So while KRR can leverage $K$, it does not necessarily exploit more information about $P_{\mathcal{X}}$ than supervised learning over the $n$ labeled samples, which is why the unlabeled samples are useless in our graph example. To address this, the seminal work Belkin et al. (2006) proposed this elegant idea of explicitly including another regularizer $\|f\|^2_{\mathcal{I}}$ that reflects the intrinsic structure of $P_{\mathcal{X}}$. For instance, $\|f\|^2_{\mathcal{I}}$ can be defined with the Laplace-Beltrami operator in manifold learning, or the graph Laplacian for graphs. In comparison, STKR also exploits $P_{\mathcal{X}}$, but in an implicit way—the construction of the STK mixes $K$ with $P_{\mathcal{X}}$. To see this: In our graph example, suppose we were to use the STK $K^2$, *i.e.* a two-step random walk. Then, (a) the graph would be useful again because the three labeled nodes were connected in $K^2$, and (b) we mixed $K$ with $P_{\mathcal{X}}$ since $K^2$ is essentially an

integral of $K \times K$ over $P_{\mathcal{X}}$. The main takeaway from the above analysis is: *With STKR, we impose another kind of smoothness we call the **target smoothness**, and it mixes the information of $K$ with the information of $P_{\mathcal{X}}$.* In the rest of this section, we formally characterize this target smoothness.

We start with formally defining "smoothness". Suppose the inter-sample similarity is characterized by a metric $d(x, x')$ over the input space $\mathcal{X}$, then one can naturally measure the smoothness of $f$ by its Lipschitz constant $\text{Lip}_d(f) = \sup_{x,x' \in \mathcal{X}, x \neq x'} \frac{|f(x) - f(x')|}{d(x,x')}$. So it suffices to specify $d(x, x')$. If $\mathcal{X}$ is an Euclidean space, then one can choose $d$ to be the Euclidean distance, which is used in lots of prior work (Tenenbaum et al., 2000; Belkin & Niyogi, 2003). The caveat is that the Euclidean distance is not guaranteed to be correlated with similarity, and $\mathcal{X}$ is not necessarily Euclidean in the first place.

Instead, one can use $K$ to define the metric, which should align better with inter-sample similarity by the definition of $K$. And if one further assumes transitivity of similarity, *i.e.* $(a, b)$ and $(b, c)$ being similar implies that $(a, c)$ are similar, then $K^p$ also aligns with similarity. The kernel metric of $K^p$ is given by $d_{K^p}(x, x') := \|K_x^p - K_{x'}^p\|_{\mathcal{H}_{K^p}} = \sum_i \lambda_i^p (\psi_i(x) - \psi_i(x'))^2$, which is equivalent to the diffusion distance defined in Coifman & Lafon (2006), and $p$ can be real-valued. Thus, kernel diffusion $\{K^p\}$ induces a multiscale metric geometry over $\mathcal{X}$, where a larger $p$ induces a weaker metric. Here "weaker" means $d_{K^p} = O(d_{K^q})$ if $p > q$. One can also think of $\{K^p\}_{p \geq 1}$ as forming a chain of smooth function classes: $L^2(P_{\mathcal{X}}) \supset \mathcal{H}_{K^1} \supset \mathcal{H}_{K^2} \supset \cdots$, and for continuous diffusion we can also have sets like $\mathcal{H}_{K^{1.5}}$. A larger $p$ imposes a stronger constraint since $\mathcal{H}_{K^p}$ is smaller.

Now we show: $\|f\|_{\mathcal{H}_{K^p}}$ is equal to its Lipschitz constant. But this is not true for $\text{Lip}_d(f)$, which is not very tractable under the topological structure of $\mathcal{X}$. Thus, we consider the space of finite signed measures over $\mathcal{X}$, denoted by $\bar{\mathcal{X}}$. For any function $f$ on $\mathcal{X}$, define its mean $\bar{f}$ as a linear functional over $\bar{\mathcal{X}}$, such that $\bar{f}(\mu) = \int_{\mathcal{X}} f(x)\mu(x)$. Then, define $d_{K^p}(\mu, \nu) := \|\int K_x^p \mu(x) - \int K_x^p \nu(x)\|_{\mathcal{H}_{K^p}}$ for $\mu, \nu \in \bar{\mathcal{X}}$, and $\overline{\text{Lip}}_{d_{K^p}}(f) := \sup_{\mu,\nu \in \overline{\mathcal{X}}, \mu \neq \nu} \frac{|\bar{f}(\mu) - \bar{f}(\nu)|}{d_{K^p}(\mu,\nu)}$. In other words, $f$ is smooth if its mean *w.r.t.* $\mu$ does not change too much when the measure $\mu$ over $\mathcal{X}$ changes by a little bit. Then, we have:

**Proposition 1** (Proofs in Appendix B). *This* $\overline{\text{Lip}}_{d_{K^p}}(f)$ *satisfies:* $\overline{\text{Lip}}_{d_{K^p}}(f) = \|f\|_{\mathcal{H}_{K^p}}, \forall f \in \mathcal{H}_{K^p}$.

We define $r_{K^p}(f) := \|f - \mathbb{E}_{P_{\mathcal{X}}}[f]\|_{P_{\mathcal{X}}}^2 / \overline{\text{Lip}}_{d_{K^p}}(f)^2 = \|f - \mathbb{E}_{P_{\mathcal{X}}}[f]\|_{P_{\mathcal{X}}}^2 / \|f\|_{\mathcal{H}_{K^p}}^2$, and use it to measure the smoothness of any $f \in L^2(P_{\mathcal{X}})$ at scale $p \geq 1$. Here $\|f\|_{\mathcal{H}_{K^p}}$ is extended to all $f \in L^2(P_{\mathcal{X}})$: If $\exists f_p \in \mathcal{H}_{K^p}$ such that $f - \mathbb{E}_{P_{\mathcal{X}}}[f] = f_p$ ($P_{\mathcal{X}}$-a.e.), then $\|f\|_{\mathcal{H}_{K^p}} := \|f_p\|_{\mathcal{H}_{K^p}}$; If there is no such $f_p$, then $\|f\|_{\mathcal{H}_{K^p}} := +\infty$. Since $K$ is centered, for any $f_1$ and $f_2$ that differ by a constant $P_{\mathcal{X}}$-a.e., there is $r_{K^p}(f_1) = r_{K^p}(f_2)$. This would not be true without the centeredness assumption. We define $r_{K^p}(f)$ as a ratio to make it scale-invariant, *i.e.* $f$ and $2f$ are equally smooth, for the same purpose of decoupling smoothness and scale. And in Appendix B.2, we will discuss the connection between $r_{K^p}(f)$ and discriminant analysis, as well as the Poincaré constant.

Now we characterize "target smoothness", an unknown property that the target $f^*$ possesses. We assume that it has the same form $r_t(f) := \|f - \mathbb{E}_{P_{\mathcal{X}}}[f]\|_{P_{\mathcal{X}}}^2 / \overline{\text{Lip}}_{d_t}(f)^2$, for some metric $d_t$ over $\bar{\mathcal{X}}$. Then, we assume all functions with "target smoothness" belong to a Hilbert space $\mathcal{H}_t$, and $x, x'$ are similar if all functions in $\mathcal{H}_t$ give them similar predictions, *i.e.* $d_t(\mu, \nu) = \sup_{\|f\|_{\mathcal{H}_t}=1} |\bar{f}(\mu) - \bar{f}(\nu)|$. We also assume that target smoothness implies base smoothness, *i.e.* $\mathcal{H}_t \subset \mathcal{H}_K$ (this is relaxable).

## 2.2 TARGET SMOOTHNESS CAN ALWAYS BE OBTAINED FROM STK: SUFFICIENT CONDITION

Let $r_t(f)$ be defined as above. Our first theorem gives the following sufficient condition: If the target smoothness preserves relative multiscale smoothness, then it must be attainable with an STK.

---

**Theorem 1.** *If $r_t(f)$ preserves relative smoothness: "$\forall f_1, f_2 \in L^2(P_{\mathcal{X}})$, if $r_{K^p}(f_1) \geq r_{K^p}(f_2)$ for **all** $p \geq 1$, then $r_t(f_1) \geq r_t(f_2)$", and $\mathcal{H}_t \subset \mathcal{H}_K$, then $r_t(f) = \|f - \mathbb{E}_{P_{\mathcal{X}}}[f]\|_{P_{\mathcal{X}}}^2 / \|f\|_{\mathcal{H}_t}^2$, and $\mathcal{H}_t$ must be an RKHS, whose reproducing kernel is an STK that admits the following form:*

$$K_s(x, x') = \sum_{i:\lambda_i > 0} s(\lambda_i)\psi_i(x)\psi_i(x'), \tag{3}$$

*for a transformation function $s : [0, +\infty) \to [0, +\infty)$ that is: (i) monotonically non-decreasing, (ii) $s(\lambda) \leq M\lambda$ for some constant $M > 0$, (iii) continuous on $[0, +\infty)$, and (iv) $C^\infty$ on $(0, +\infty)$.*

---

The proof is done by sequentially showing that (i) $\mathcal{H}_t$ is an RKHS; (ii) Its reproducing kernel is $K_s(x, x') := \sum_i s_i \psi_i(x)\psi_i(x')$, with $s_1 \geq s_2 \geq \cdots \geq 0$; (iii) $s_i = O(\lambda_i)$; (iv) There exists such a function $s(\lambda)$ that interpolates all $s_i$. From now on, we will use $\mathcal{H}_{K_s}$ to denote $\mathcal{H}_t$. This theorem

implies that $s(\lambda)$ makes the eigenvalues decay faster than the base kernel, but it does not imply that $K_s$ is a linear combination of $\{K^p\}_{p\geq 1}$. This result naturally leads to KRR with $\|f\|^2_{\mathcal{H}_{K_s}} = \|f\|^2_{\mathcal{H}_t}$:

$$\tilde{f} \in \underset{f - \mathbb{E}_{P_{\mathcal{X}}}[f] \in \mathcal{H}_{K_s}}{\arg\min} \left\{ \frac{1}{n} \sum_{i=1}^{n} (f(x_i) - y_i)^2 + \beta_n \|f - \mathbb{E}_{X \sim P_{\mathcal{X}}}[f(X)]\|^2_{\mathcal{H}_{K_s}} \right\}, \qquad (4)$$

which we term *spectrally transformed kernel regression* (STKR). One could also relax the assumption $\mathcal{H}_t \subset \mathcal{H}_K$ by considering $\mathcal{H}_{K^p}$ for $p \geq p_0$ where $p_0 < 1$. Assuming that $\mathcal{H}_{K^{p_0}}$ is still an RKHS and $\mathcal{H}_t \subset \mathcal{H}_{K^{p_0}}$, one can prove the same result as Theorem 1, with *(ii)* changed to $s(\lambda) \leq M\lambda^{p_0}$.

Now we develop theory for STKR, and show how it exploits the unlabeled data. Here is a road map:

(a) We first study the easier transform-aware setting in Section 3, where a good $s(\lambda)$ is given by an oracle. But even though $s(\lambda)$ is known, $K_s$ is inaccessible as one cannot obtain $\psi_i$ with finite samples. Unlabeled data becomes useful when one constructs a kernel $\hat{K}_s$ to approximate $K_s$.

(b) In reality, such an oracle need not exist. So in Section 4, we study the harder transform-agnostic setting where we have no knowledge of $s(\lambda)$ apart from Theorem 1. We examine two methods:

 (i) STKR with inverse Laplacian (Example 1), which is popular in semi-supervised learning and empirically works well on lots of tasks though the real $s$ might not be inverse Laplacian.

 (ii) STKR with kernel PCA, which extracts the top-$d$ eigenfunctions to be an encoder and then learns a linear probe atop. This is used in many manifold and representation learning methods. Here, unlabeled data is useful when approximating $\psi_1, \cdots, \psi_d$ in kernel PCA.

**Notation:** For any kernel $K$, we use $\boldsymbol{G}_K \in \mathbb{R}^{(n+m)\times(n+m)}, \boldsymbol{G}_{K,n} \in \mathbb{R}^{n\times n}, \boldsymbol{G}_{K,m} \in \mathbb{R}^{m\times m}$ to respectively denote its Gram matrix on all, labeled and unlabeled samples, *i.e.* $\boldsymbol{G}_K[i,j] = K(x_i, x_j)$.

## 3 Transform-aware: STKR with Known Polynomial Transform

Let the scale of $f^*$ be measured by $B$. This section supposes that $s(\lambda)$ is known, and the following:

**Assumption 1.** $s(\lambda) = \sum_{p=1}^{\infty} \pi_p \lambda^p$ is a polynomial, with $\pi_p \geq 0$.

**Assumption 2.** *There exists a constant $\kappa > 0$ such that $K(x,x) \leq \kappa^2$ for $P_{\mathcal{X}}$-almost all $x$.*

**Assumption 3.** $\mathbb{E}_{P_{\mathcal{X}}}[f^*] = 0$, *and there exist constants $B, \epsilon > 0$ such that:* $\|f^*\|_{P_{\mathcal{X}}} \leq B$, $f^* \in \mathcal{H}_{K_s}$, *and* $\|f^*\|^2_{\mathcal{H}_{K_s}} \leq \epsilon \|f^*\|^2_{P_{\mathcal{X}}}$ *(i.e. $r_t(f^*) \geq \epsilon^{-1}$). (cf. the **isometry property** in Zhai et al. (2024))*

**Assumption 4.** $P_{\mathcal{X}\mathcal{Y}}$ *satisfies the **moment condition** for $\sigma, L > 0$:* $\mathbb{E}[|y - f^*(x)|^r] \leq \frac{1}{2} r! \sigma^2 L^{r-2}$ *for all $r \geq 2$ and $P_{\mathcal{X}}$-almost all $x$. (e.g. For $y - f^*(x) \sim \mathcal{N}(0, \sigma^2)$, this holds with $L = \sigma$.)*

Assumption 1 is a natural condition for discrete diffusion, such as a multi-step random walk on a graph, and $p$ starts from 1 because $s(0) = 0$. The assumption $\mathbb{E}_{P_{\mathcal{X}}}[f^*] = 0$ in Assumption 3 is solely for the simplicity of the results, without which one can prove the same but more verbose bounds. The moment condition Assumption 4 is essentially used to control the size of the label noise.

➤ **Method:** We implement inductive STKR by constructing a computable kernel $\hat{K}_s(x,x')$ to approximate the inaccessible $K_s$. For example, if $\check{K}_s(x,x') = K^2(x,x') = \int K(x,x_0)K(x',x_0)dp(x_0)$, then a Monte-Carlo approximation can be done by replacing the integral over $x_0$ with an average over $x_1, \cdots, x_{n+m}$. Computing this average leverages the unlabeled data. Specifically, we define:

$$\hat{f} \in \underset{f \in \mathcal{H}_{\hat{K}_s}}{\arg\min} \left\{ \frac{1}{n} \sum_{i=1}^{n} (f(x_i) - y_i)^2 + \beta_n \|f\|^2_{\mathcal{H}_{\hat{K}_s}} \right\}, \qquad (5)$$

where $\hat{K}_s(x,x') := \sum_{p=1}^{\infty} \pi_p \hat{K}^p(x,x')$; $\hat{K}^1 = K$; $\forall p \geq 2, \hat{K}^p(x,x') = \dfrac{\boldsymbol{v}_K(x)^\top \boldsymbol{G}_K^{p-2} \boldsymbol{v}_K(x')}{(n+m)^{p-1}}$.

Here, $\boldsymbol{v}_K(x) \in \mathbb{R}^{n+m}$ such that $\boldsymbol{v}_K(x)[i] = K(x,x_i), i \in [n+m]$. One can compute $\hat{K}_s(x,x')$ for any $x, x'$ with full access to $K(x,x')$. Let $\boldsymbol{y} = [y_1, \cdots, y_n] \in \mathbb{R}^n$, and $\boldsymbol{v}_{K_s,n}(x) \in \mathbb{R}^n$ be defined as $\boldsymbol{v}_{K_s,n}(x)[i] = K_s(x,x_i)$ for $i \in [n]$. The following closed-form solutions can be derived from the Representer Theorem. While they are not necessarily unique, we will use them throughout this work:

$$\begin{cases} \tilde{f}(x) = \boldsymbol{v}_{K_s,n}(x)^\top \tilde{\boldsymbol{\alpha}}, & \tilde{\boldsymbol{\alpha}} = (\boldsymbol{G}_{K_s,n} + n\beta_n \boldsymbol{I}_n)^{-1} \boldsymbol{y}; & (6) \\ \hat{f}(x) = \boldsymbol{v}_{\hat{K}_s,n}(x)^\top \hat{\boldsymbol{\alpha}}, & \hat{\boldsymbol{\alpha}} = (\boldsymbol{G}_{\hat{K}_s,n} + n\beta_n \boldsymbol{I}_n)^{-1} \boldsymbol{y}. & (7) \end{cases}$$

➤ **Results overview:** Now, for all $s$ and $f^*$ that satisfy the above assumptions, we bound the prediction error $\|\hat{f} - f^*\|_{P_\mathcal{X}}^2$. The bound has two parts and here is a synopsis: In Part 1 (Theorem 2), we assume access to $K_s$, and use the general results in Fischer & Steinwart (2020) to bound the estimation error entailed by KRR with finite samples and label noise; In Part 2 (Theorem 3), we bound the approximation error entailed by using $\hat{K}_s$ to approximate the inaccessible $K_s$.

> **Theorem 2.** *Let $M$ be given by Theorem 1. If eigenvalues of $K_s$ decay by order $p^{-1}$ for $p \in (0, 1]$, i.e. $s(\lambda_i) = O(i^{-\frac{1}{p}})$ for all $i$, then under Assumptions 2 and 4, for a sequence of $\{\beta_n\}_{n \geq 1}$ with $\beta_n = \Theta(n^{-\frac{1}{1+p}})$, there is a constant $c_0 > 0$ independent of $n \geq 1$ and $\tau \geq \kappa^{-1} M^{-\frac{1}{2}}$ such that*
> $$\left\| \tilde{f} - f^* \right\|_{P_\mathcal{X}}^2 \leq c_0 \tau^2 \kappa^2 M \left[ (\epsilon B^2 + \sigma^2) n^{-\frac{1}{1+p}} + \max \left\{ L^2, \kappa^2 M \epsilon B^2 \right\} n^{-\frac{1+2p}{1+p}} \right]$$
> *holds for all $f^*$ satisfying Assumption 3 and sufficiently large $n$ with probability at least $1 - 4e^{-\tau}$.*

*Remark.* The $O(n^{-\frac{1}{1+p}})$ learning rate is *minimax optimal* as shown in Fischer & Steinwart (2020), *i.e.* one can construct an example where the learning rate is at most $\Omega(n^{-\frac{1}{1+p}})$. And under Assumption 2, one can always choose $p = 1$ since $i \cdot s(\lambda_i) \leq \sum_{j=1}^i s(\lambda_j) \leq M \sum \lambda_j = M \operatorname{Tr}(T_K) \leq M\kappa^2$. So one statistical benefit of using an appropriate $s$ is to make the eigenvalues decay faster (*i.e.* make $p$ smaller). Also note that the random noise should scale with $f^*$, which means that $\sigma, L = \Theta(B)$.

> **Theorem 3.** *Let $\hat{\lambda}_1$ be the largest eigenvalue of $\frac{G_K}{n+m}$, and denote $\lambda_{\max} := \max \left\{ \lambda_1, \hat{\lambda}_1 \right\}$. Then, under Assumptions 1 and 2, for any $\delta > 0$, it holds with probability at least $1 - \delta$ that:*
> $$\left\| \hat{f} - \tilde{f} \right\|_{P_\mathcal{X}}^2 \leq 8s(\lambda_{\max}) \left. \nabla_\lambda \left( \frac{s(\lambda)}{\lambda} \right) \right|_{\lambda = \lambda_{\max}} \frac{\beta_n^{-2} \kappa^4}{\sqrt{n+m}} \left( 2 + \sqrt{2 \log \frac{1}{\delta}} \right) \frac{\|y\|_2^2}{n}.$$

*Remark.* The key to prove this is to first prove a uniform bound for $|\hat{K}_s(x, x_j) - K_s(x, x_j)|$ over all $x$ and $j$. With Assumptions 3 and 4, an $O(B^2 + \sigma^2 + L^2)$ bound for $\frac{\|y\|_2^2}{n}$ can be easily obtained. If $\beta_n = \Theta(n^{-\frac{1}{1+p}})$ as in Theorem 2, then with $m = \omega(n^{\frac{4}{1+p}})$ this bound vanishes, so more unlabeled samples than labeled ones are needed. Moreover, $\hat{\lambda}_1$ is known to be close to $\lambda_1$ when $n + m$ is large:

**Lemma 2.** *(Shawe-Taylor et al., 2005, Theorem 2) For any $\delta > 0$, with probability at least $1 - \delta$,*
$$\hat{\lambda}_1 \leq \lambda_1 + \frac{\kappa^2}{\sqrt{n+m}} \left[ 2\sqrt{2} + \sqrt{19 \log \frac{2(n+m+1)}{\delta}} \right].$$

➤ **Implementation:** STKR amounts to solving $A\hat{\alpha} = y$ for $A = G_{\hat{K}_s, n} + n\beta_n I_n$ by Eqn. (7). There are two approaches: (i) Directly computing $A$ (Algorithm 3 in Appendix C) can be slow due to costly matrix multiplication; (ii) Iterative methods are faster by only performing matrix-vector multiplication. Algorithm 1 solves $A\hat{\alpha} = y$ via Richardson iteration. We name it STKR-Prop as it is very similar to label propagation (Label-Prop) (Zhou et al., 2003). If $s(\lambda) = \sum_{p=1}^q \pi_p \lambda^p$ and $q < \infty$, and computing $K(x, x')$ for any $x, x'$ takes $O(1)$ time, then Algorithm 1 has a time complexity of $O(q(n+m)^2 \beta_n^{-1} s(\lambda) \log \frac{1}{\epsilon})$ for achieving error less than $\epsilon$, where $\lambda$ is a known upper bound of $\lambda_1$ (see derivation in Appendix C). Besides, STKR-Prop is much faster when $K$ is sparse. In particular, for a graph with $|E|$ edges, STKR-Prop runs in $\tilde{O}(q|E|\beta_n^{-1})$ time, which is as fast as Label-Prop.

At inference time, one can store $v$ computed in line 4 of Algorithm 1 in the memory. Then for any $x$, there is $\hat{f}(x) = \sum_{i=1}^{n+m} K(x_i, x) v_i + \pi_1 \sum_{j=1}^n K(x_j, x) \hat{\alpha}_j$, which takes $O(n + m)$ time to compute. This is much faster than Chapelle et al. (2002) who solved an optimization problem for each new $x$.

For some other transformations, including the inverse Laplacian we are about to discuss, $s$ is complex, but $s^{-1}(\lambda) = \sum_{p=0}^{q-1} \xi_p \lambda^{p-r}$ is simple. For this type of $s(\lambda)$, Algorithm 1 is infeasible, but there is a viable method in Algorithm 2: It finds $\theta \in \mathbb{R}^{n+m}$ such that $Q\theta = [\hat{\alpha}, 0_m]^\top$ and $M\theta = \tilde{y}$, where $Q := \sum_{p=0}^{q-1} \xi_p \left( \frac{G_K}{n+m} \right)^p$, $M := (n+m)\tilde{I}_n \left( \frac{G_K}{n+m} \right)^r + n\beta_n Q$, $\tilde{I}_n := \operatorname{diag}\{1, \cdots, 1, 0, \cdots, 0\}$ with $n$ ones and $m$ zeros, and $\tilde{y} := [y, 0_m]^\top$. In Appendix C we will derive these formulas step by step, and prove its time complexity to be $\tilde{O}(\max\{q, r\}(n+m)^2 \beta_n^{-1})$. And at inference time, one can compute $\hat{f}(x) = v_K(x)^\top \left( \frac{G_K}{n+m} \right)^{r-1} \theta$ in $O(n+m)$ time for any $x$ by storing $\left( \frac{G_K}{n+m} \right)^{r-1} \theta$ in the

| **Algorithm 1** STKR-Prop for simple $s$ | **Algorithm 2** STKR-Prop for simple $s^{-1}$ |
|---|---|
| **Input:** $\boldsymbol{G}_K, s(\lambda), \beta_n, \boldsymbol{y}, \gamma, \epsilon$ | **Input:** $\boldsymbol{G}_K, s^{-1}(\lambda), \beta_n, \boldsymbol{y}, \gamma, \epsilon$ |
| 1: Initialize: $\hat{\boldsymbol{\alpha}} \leftarrow \boldsymbol{0} \in \mathbb{R}^n$ | 1: Initialize: $\boldsymbol{\theta} \leftarrow \boldsymbol{0} \in \mathbb{R}^{n+m}, \tilde{\boldsymbol{y}} \leftarrow [\boldsymbol{y}, \boldsymbol{0}_m]^\top$ |
| 2: **while** True **do** | 2: **while** True **do** |
|    # Compute $\boldsymbol{u} = (\boldsymbol{G}_{\hat{K}_s,n} + n\beta_n \boldsymbol{I}_n)\hat{\boldsymbol{\alpha}}$ |    # Compute $\boldsymbol{u} = \boldsymbol{M}\boldsymbol{\theta}$ |
| 3:    $\tilde{\boldsymbol{\alpha}} \leftarrow \frac{1}{n+m}\boldsymbol{G}_{K,n+m,n}\hat{\boldsymbol{\alpha}}, \boldsymbol{v} \leftarrow \boldsymbol{0} \in \mathbb{R}^{n+m}$ | 3:    $\boldsymbol{v} \leftarrow \boldsymbol{0} \in \mathbb{R}^{n+m}$ |
| 4:    **for** $p = q, \cdots, 2$ **do** $\boldsymbol{v} \leftarrow \frac{\boldsymbol{G}_K \boldsymbol{v}}{n+m} + \pi_p \tilde{\boldsymbol{\alpha}}$ | 4:    **for** $p = q-1, \cdots, 0$ **do** $\boldsymbol{v} \leftarrow \frac{\boldsymbol{G}_K \boldsymbol{v}}{n+m} + \xi_p \boldsymbol{\theta}$ |
| 5:    $\boldsymbol{u} \leftarrow \boldsymbol{G}_{K,n+m,n}^\top \boldsymbol{v} + \pi_1 \boldsymbol{G}_{K,n}\hat{\boldsymbol{\alpha}} + n\beta_n \hat{\boldsymbol{\alpha}}$ | 5:    $\boldsymbol{u} \leftarrow \left[\left(\frac{\boldsymbol{G}_K^r}{(n+m)^{r-1}}\boldsymbol{\theta}\right)[1:n], \boldsymbol{0}_m\right]^\top + n\beta_n \boldsymbol{v}$ |
| 6:    **if** $\|\boldsymbol{u} - \boldsymbol{y}\|_2 < \epsilon \|\boldsymbol{y}\|_2$ **then return** $\hat{\boldsymbol{\alpha}}$ | 6:    $\boldsymbol{a} \leftarrow \boldsymbol{u} - \tilde{\boldsymbol{y}}, \boldsymbol{\theta} \leftarrow \boldsymbol{\theta} - \gamma \boldsymbol{a}$ |
| 7:    $\hat{\boldsymbol{\alpha}} \leftarrow \hat{\boldsymbol{\alpha}} - \gamma(\boldsymbol{u} - \boldsymbol{y})$ | 7:    **if** $\|\boldsymbol{a}\|_2 < \epsilon \|\boldsymbol{y}\|_2$ **then return** $\boldsymbol{\theta}$ |

memory, where $\boldsymbol{v}_K$ is defined as in Eqn. (5). Once again, for a graph with $|E|$ edges, STKR-Prop has a time complexity of $\tilde{O}(\max\{q,r\}|E|\beta_n^{-1})$, which is as fast as Label-Prop. Finally, here we showed the existence of a good solver (Richardson), but practitioners could surely use other linear solvers.

# 4 TRANSFORM-AGNOSTIC: INVERSE LAPLACIAN AND KERNEL PCA

We have derived learning guarantees for general inductive STKR when $s$ is known. This is useful, but in reality, it is unreasonable to presume that such an oracle $s$ will be given. What should one do if one has zero knowledge of $s(\lambda)$ but still want to enforce target smoothness? Here we provide two parallel methods. The first option one can try is STKR with the canonical inverse Laplacian transformation. Laplacian as a regularizer has been widely used in various context (Zhou et al., 2003; Johnson & Zhang, 2008; HaoChen et al., 2021; Zhai et al., 2024). For our problem, we want $\|f\|_{\mathcal{H}_{K_s}}^2 = f^\top K_s^{-1} f$ to be the Laplacian, so the kernel $K_s$ should be the inverse Laplacian:

**Example 1** (Inverse Laplacian for the inductive setting). *For $\eta \in (0, \lambda_1^{-1})$, define $K_s$ such that $K_s^{-1}(x,x') = K^{-1}(x,x') - \eta K^0(x,x')$. $K^{-1}$ and $K^0$ are STKs with $s(\lambda) = \lambda^{-1}$ and $s(\lambda) = \lambda^0$. Then, $s^{-1}(\lambda) = \lambda^{-1} - \eta > 0$ for $\lambda \in (0, \lambda_1]$ ($s^{-1}$ is the reciprocal, not inverse), $s(\lambda) = \frac{\lambda}{1-\eta\lambda} = \sum_{p=1}^\infty \eta^{p-1}\lambda^p$, and $\|f\|_{\mathcal{H}_{K_s}}^2 = \|f\|_{\mathcal{H}_K}^2 - \eta\|f\|_{P_\mathcal{X}}^2$. Classical Laplacian has $\eta = 1$ and $\lambda_1 < 1$. For the connection between transductive and inductive versions of Laplacian, see Appendix B.3.*

This canonical transformation empirically works well on lots of tasks, and also have this guarantee:

**Proposition 3.** *Let $s$ be the inverse Laplacian (Example 1), and $s^*$ be an arbitrary oracle satisfying Theorem 1. Suppose $f^*$ satisfies Assumption 3 w.r.t. $s^*$, but STKR (Eqn. (7)) is performed with $s$. Then, Theorem 3 still holds for $\tilde{f}$ given by Eqn. (6), and Theorem 2 holds by replacing $\epsilon$ with $M\epsilon$.*

Note that this result does not explain why inverse Laplacian is so good — its superiority is mainly an empirical observation, so it could still be bad on some tasks, for which there is the second option. The key observation here is that since $s$ is proved in Theorem 1 to be monotonic, the order of $\psi_1, \psi_2, \cdots$ must remain unchanged. So if one is asked to choose $d$ functions to represent the target function, regardless of $s$ the best choice with the lowest worst-case approximation error must be $\psi_1, \cdots, \psi_d$:

**Proposition 4.** *Let $s$ be any transformation function that satisfies Theorem 1. Let $\mathcal{F}_s$ be the set of functions that satisfy Assumption 3 for this $s$. Then, the following holds for all $\hat{\Psi} = [\hat{\psi}_1, \cdots, \hat{\psi}_d]$ such that $\hat{\psi}_i \in L^2(P_\mathcal{X})$, as long as $s(\lambda_1)\epsilon > 1$ and $\frac{s(\lambda_{d+1})}{s(\lambda_1)-s(\lambda_{d+1})}[s(\lambda_1)\epsilon - 1] \leq \frac{1}{2}$:*

$$\max_{f \in \mathcal{F}_s} \min_{\boldsymbol{w} \in \mathbb{R}^d} \left\|\boldsymbol{w}^\top \hat{\Psi} - f\right\|_{P_\mathcal{X}}^2 \geq \frac{s(\lambda_{d+1})}{s(\lambda_1) - s(\lambda_{d+1})}[s(\lambda_1)\epsilon - 1]B^2.$$

*To attain equality, it is sufficient for $\hat{\Psi}$ to span $\text{span}\{\psi_1, \cdots, \psi_d\}$, and necessary if $s(\lambda_d) > s(\lambda_{d+1})$.*

➢ **Method:** This result motivates using representation learning with two stages: A self-supervised pretraining stage that learns a $d$-dimensional encoder $\hat{\Psi} = [\hat{\psi}_1, \cdots, \hat{\psi}_d]$ with the unlabeled samples, and a supervised fitting stage that fits a linear probe on $\hat{\Psi}$ with the labeled samples. The final predictor is $\hat{f}_d(x) = \hat{\boldsymbol{w}}^\top \hat{\Psi}(x)$, for which we do not include a bias term since $f^*$ is assumed to be centered.

For pretraining, the problem boils down to extracting the top-$d$ eigenfunctions of $T_K$, for which a classical method is kernel PCA (Schölkopf & Smola, 2002, Chapter 14). Indeed, kernel PCA has been widely applied in manifold learning (Belkin & Niyogi, 2003; Bengio et al., 2004), and more recently self-supervised pretraining (Johnson et al., 2023). Suppose that $\boldsymbol{G}_{K,m} \in \mathbb{R}^{m \times m}$, the Gram matrix of $K$ over $x_{n+1}, \cdots, x_{n+m}$, is at least rank-$d$. Then, kernel PCA can be formulated as:

$$\hat{\psi}_i(x) = \sum_{j=1}^{m} \boldsymbol{v}_i[j] K(x_{n+j}, x), \tag{8}$$

$$\text{where } \boldsymbol{G}_{K,m} \boldsymbol{v}_i = m \tilde{\lambda}_i \boldsymbol{v}_i; \; \tilde{\lambda}_1 \geq \cdots \geq \tilde{\lambda}_d > 0; \; \boldsymbol{v}_i \in \mathbb{R}^m; \; \forall i, j \in [d], \langle \boldsymbol{v}_i, \boldsymbol{v}_j \rangle = \frac{\delta_{i,j}}{m \tilde{\lambda}_i}.$$

For any $i, j \in [d]$, there is $\langle \hat{\psi}_i, \hat{\psi}_j \rangle_{\mathcal{H}_K} = \boldsymbol{v}_i^\top \boldsymbol{G}_{K,m} \boldsymbol{v}_j = \delta_{i,j}$. Consider running KRR *w.r.t.* $K$ over all $f = \boldsymbol{w}^\top \hat{\Psi}$. For $\hat{f} = \hat{\boldsymbol{w}}^\top \hat{\Psi}$, there is $\|\hat{f}\|_{\mathcal{H}_K}^2 = \sum_{i,j=1}^{d} \hat{w}_i \hat{w}_j \langle \hat{\psi}_i, \hat{\psi}_j \rangle_{\mathcal{H}_K} = \|\hat{\boldsymbol{w}}\|_2^2$. So it amounts to minimize $\frac{1}{n} \sum_{i=1}^{n} (\hat{\boldsymbol{w}}^\top \hat{\Psi}(x_i) - y_i)^2 + \beta_n \|\hat{\boldsymbol{w}}\|_2^2$ as in ridge regression, which is an approximation of STKR with a "truncation function" $s(\lambda_i) = \lambda_i$ if $i \leq d$, and 0 otherwise (not a real function if $\lambda_d = \lambda_{d+1}$). Denote $\hat{\Psi}(\boldsymbol{X}_n) = [\hat{\Psi}(x_1), \cdots, \hat{\Psi}(x_n)] \in \mathbb{R}^{d \times n}$. Then, the final predictor is given by:

$$\hat{f}_d = \hat{\boldsymbol{w}}^{*\top} \hat{\Psi}, \quad \hat{\boldsymbol{w}}^* = \left( \hat{\Psi}(\boldsymbol{X}_n) \hat{\Psi}(\boldsymbol{X}_n)^\top + n \beta_n \boldsymbol{I}_d \right)^{-1} \hat{\Psi}(\boldsymbol{X}_n) \boldsymbol{y}. \tag{9}$$

➤ **Results overview:** We now bound the prediction error of $\hat{f}_d$ for all $f^*$ satisfying Assumption 3, with no extra knowledge about $s(\lambda)$. The bound also has two parts. In Part 1 (Theorem 4), we bound the estimation error entailed by KRR over $\mathcal{H}_{\hat{\Psi}}$ given by Eqn. (9), where $\mathcal{H}_{\hat{\Psi}}$ is the RKHS spanned by $\hat{\Psi} = [\hat{\psi}_1, \cdots, \hat{\psi}_d]$, which is a subspace of $\mathcal{H}_K$; In Part 2 (Theorem 5), we bound the approximation error, which is the distance from $f^*$ to this subspace $\hat{\Psi}$. Note that if $\hat{\Psi}$ has insufficient representation capacity (*e.g.* $d$ is small), then the approximation error will not vanish. Specifically, let $\tilde{f}_d$ be the projection of $f^*$ onto $\mathcal{H}_{\hat{\Psi}}$, *i.e.* $\tilde{f}_d = \tilde{\boldsymbol{w}}^\top \hat{\Psi}$, and $\langle \tilde{f}_d, f^* - \tilde{f}_d \rangle_{\mathcal{H}_K} = 0$. Then, Part 1 bounds the KRR estimation error with $\tilde{f}_d$ being the target function, and Part 2 bounds $\|f^* - \tilde{f}_d\|^2$.

---

**Theorem 4.** *Let $M$ be given by Theorem 1. Then, under Assumptions 2 and 4, for Eqn. (9) with a sequence of $\{\beta_n\}_{n \geq 1}$ with $\beta_n = \Theta(n^{-\frac{1}{1+p}})$ for any $p \in (0, 1]$, and any $\delta > 0$ and $\tau \geq \kappa^{-1}$, if $n \geq 16\kappa^4 \tilde{\lambda}_d^{-2} \left(2 + \sqrt{2\log\frac{2}{\delta}}\right)^2$, then there is a constant $c_0 > 0$ independent of $n, \tau$ such that:*

$$\left\| \hat{f}_d - \tilde{f}_d \right\|_{P_\mathcal{X}}^2 \leq 3\left( \left\| f^* - \tilde{f}_d \right\|_{P_\mathcal{X}}^2 + \frac{\tilde{\lambda}_d}{4} \left\| f^* - \tilde{f}_d \right\|_{\mathcal{H}_K}^2 \right)$$

$$+ c_0 \tau^2 \left[ \left( \kappa^2 M \epsilon B^2 + \kappa^2 \sigma^2 \right) n^{-\frac{1}{1+p}} + \kappa^2 \max\left\{ L^2, \kappa^2 M \epsilon B^2 \right\} n^{-\frac{1+2p}{1+p}} \right]$$

*holds for all $f^*$ under Assumption 3 and sufficiently large $n$ with probability at least $1 - 4e^{-\tau} - \delta$.*

---

*Remark.* This bound has two terms. The first term bounds the gap between $\boldsymbol{y}$ and new labels $\tilde{\boldsymbol{y}}$, where $\tilde{y}_i = y_i - f^*(x_i) + \tilde{f}_d(x_i)$. The second term again comes from the results in Fischer & Steinwart (2020). Comparing the second term to Theorem 2, we can see that it achieves the fastest minimax optimal learning rate (*i.e.* $p$ can be arbitrarily close to 0), as the eigenvalues decay the fastest with $s$ being the "truncation function". But the side effect of this statistical benefit is the first term, as the $d$-dimensional $\hat{\Psi}$ has limited capacity. The coefficient 3 can be arbitrarily close to 1 with larger $n, c_0$. Our astute readers might ask why $\hat{\Psi}$ is learned only with the unlabeled samples, while in the last section STKR was done with both labeled and unlabeled samples. This is because in the supervised fitting stage, the function class is the subspace spanned by $\hat{\Psi}$. To apply uniform deviation bounds in Theorem 4, this function class, and therefore $\hat{\Psi}$, must not see $x_1, \cdots, x_n$ during pretraining. On the contrary, the function class in Theorem 2 is $\mathcal{H}_{K_s}$, which is independent of $x_1, \cdots, x_n$ by definition.

---

**Theorem 5.** *Let $M$ be given by Theorem 1. Let $f^* - \tilde{f}_d = bg$, where $b \in \mathbb{R}$, and $g \in \mathcal{H}_K$ such that $\|g\|_{\mathcal{H}_K} = 1$ and $\langle g, \hat{\psi}_i \rangle_{\mathcal{H}_K} = 0$ for $i \in [d]$. Then, $\|f^* - \tilde{f}_d\|_{P_\mathcal{X}}^2 = b^2 \|g\|_{P_\mathcal{X}}^2$, and*

$$\|f^* - \tilde{f}_d\|_{\mathcal{H}_K}^2 = b^2 \leq \frac{\epsilon M \lambda_1 - \frac{1}{2}}{\lambda_1 - \|g\|_{P_\mathcal{X}}^2} B^2 \qquad \text{for all } f^* \text{ satisfying Assumption 3.} \tag{10}$$

*And if Assumption 2 holds, then for any $\delta > 0$, it holds with probability at least $1 - \delta$ that:*

$$\lambda_{d+1} \leq \|g\|_{P_\mathcal{X}}^2 \leq \lambda_{d+1} + \frac{\kappa^2}{\sqrt{m}} \left( 2\sqrt{d} + 3\sqrt{\log\frac{6}{\delta}} \right).$$

---

*Remark.* When $m$ is sufficiently large, $\|g\|_{P_\mathcal{X}}^2$ can be very close to $\lambda_{d+1}$. Compared to Proposition 4, one can see that the bound for $\|f^* - \tilde{f}_d\|_{P_\mathcal{X}}^2 = b^2 \|g\|_{P_\mathcal{X}}^2$ given by this result is near tight provided that $\frac{s(\lambda_1)}{\lambda_1} = \frac{s(\lambda_{d+1})}{\lambda_{d+1}} = M$: The only difference is that Eqn. (10) has $\epsilon M \lambda_1 - \frac{1}{2}$ instead of $\epsilon M \lambda_1 - 1$.

Table 1: Experiment results. We compare Label-Prop (LP) to STKR-Prop (SP) with inverse Laplacian (Lap), with polynomial $s(\lambda) = \lambda^8$ (poly), with kernel PCA (topd), and with $s(\lambda) = \lambda$ (KRR) (*i.e.* KRR with base kernel). (t) and (i) indicate transductive and inductive settings. Test samples account for 1% of all samples. We report the accuracies of the argmax prediction of the estimators (%). Optimal hyperparameters are selected using a validation set (see Appendix D for details). Standard deviations are given across ten random seeds.

| Dataset | LP (t) | SP-Lap (t) | SP-poly (t) | SP-topd (t) | SP-Lap (i) | SP-poly (i) | SP-topd (i) | KRR (i) |
|---|---|---|---|---|---|---|---|---|
| **Computers** | $77.30_{3.05}$ | $77.81_{3.94}$ | $76.72_{4.12}$ | $80.80_{3.06}$ | $77.15_{2.64}$ | $71.91_{4.13}$ | $80.80_{3.28}$ | $26.35_{4.34}$ |
| **Cora** | $73.33_{6.00}$ | $77.04_{5.74}$ | $71.48_{5.80}$ | $69.26_{7.82}$ | $67.78_{7.62}$ | $65.19_{9.11}$ | $63.70_{6.00}$ | $28.52_{8.56}$ |
| **DBLP** | $66.44_{3.78}$ | $65.42_{5.02}$ | $64.52_{4.20}$ | $64.86_{4.60}$ | $65.20_{4.92}$ | $64.51_{4.05}$ | $63.16_{3.41}$ | $44.80_{3.86}$ |

Our analysis in this section follows the framework of Zhai et al. (2024), but we have the following technical improvements: (a) Estimation error: They bound with classical local Gaussian and localized Rademacher complexity, while we use the tighter bound in Fischer & Steinwart (2020) that is minimax optimal; (b) Approximation error: Our Theorem 5 has three improvements. (i) $\|g\|_{P_{\mathcal{X}}}^2 - \lambda_{d+1}$ is $O(\sqrt{d})$ instead of $O(d)$; (ii) It does not require delocalization of the top-$d$ eigenfunctions, thereby removing the dependence on the covariance matrix; (iii) Our bound does not depend on $\lambda_d^{-1}$.

Eigen-decomposition of $\boldsymbol{G}_{k,m}$ takes $O(m^3)$ time in general, though as of today the fastest algorithm takes $O(m^\omega)$ time with $\omega < 2.38$ (Demmel et al., 2007), and could be faster if the kernel is sparse.

## 5 EXPERIMENTS

We implement STKR-Prop (SP) with inverse Laplacian (Lap), polynomial (poly) $s(\lambda) = \lambda^p$, and kernel PCA (topd). We run them on several node classification tasks, and compare them to Label-Prop (LP) and KRR with the base kernel (*i.e.* STKR with $s(\lambda) = \lambda$). Details and full results are deferred to Appendix D, and here we report a portion of the results in Table 1, in which the best and second-best performances for each dataset are marked in red and blue. We make the following observations:

(a) STKR works pretty well with general polynomial $s(\lambda)$ in the inductive setting. In the transductive setting, the performance of SP-Lap is similar to LP, and SP-poly is slightly worse. The inductive performance is slightly worse than transductive, which is reasonable since there is less information at train time for the inductive setting. Note that LP does not work in the inductive setting.

(b) STKR with $s(\lambda) = \lambda^p$ for $p > 1$ is much better than KRR (*i.e.* $p = 1$). In fact, we observe that for STKR with $s(\lambda) = \lambda^p$, a larger $p$ performs better (see Figure 2 in Appendix D). This suggests one possible reason why inverse Laplacian works so well empirically: It contains $K^p$ for $p = 1, 2, \cdots$, so it can use multi-step similarity information up to infinitely many steps.

(c) STKR also works pretty well with kernel PCA. Specifically, on 3 of the 9 datasets we use, such as `Computers`, kernel PCA is better than LP and STKR with inverse Laplacian. This shows that inverse Laplacian and kernel PCA are two parallel methods — neither is superior.

## 6 CONCLUSION

This work revisited the classical idea of STKR, and proposed a new class of general and scalable STKR estimators able to leverage unlabeled data with a base kernel. We established STKR as a general and principled approach, provided scalable implementations for general transformation and inductive settings, and proved statistical bounds with technical improvements over prior work.

**Limitations and open problems.** This work assumes full access to $K(x, x')$, but in some cases computing $K(x, x')$ might be slow or impossible. The positive-pair kernel in contrastive learning (Johnson et al., 2023) is such an example, for which computing $K$ is hard but computing $\|f\|_{\mathcal{H}_K}^2$ is easy, so our methods need to be modified accordingly. Also, this work does not talk about how to choose the right base kernel $K$, which is a critical yet difficult open problem. For graph tasks, STKR like label propagation only leverages the graph, but it does not utilize the node features that are usually provided, which are important for achieving high performances in practice. Finally, this work focuses on the theoretical part, and a more extensive empirical study on STKR is desired, especially within the context of manifold learning, and modern self-supervised and semi-supervised learning.

There are three open problems from this work. (i) Improving the minimax optimal learning rate: In this work, we provided statistical bounds *w.r.t.* $n, m$ jointly, but one question we did not answer is: If $m$ is sufficiently large, can we improve the minimax optimal learning rate *w.r.t.* $n$ proved in prior work on supervised learning? (ii) Distribution shift: Diffusion induces a chain of smooth function classes $L^2(P_{\mathcal{X}}) \supset \mathcal{H}_{K^1} \supset \mathcal{H}_{K^2} \supset \cdots$, but this chain will collapse if $P_{\mathcal{X}}$ changes. Can one learn predictors or encoders that are robust to the shift in $P_{\mathcal{X}}$? (iii) Combining multiple kernels: In practice, usually the predictor is expected to satisfy multiple constraints. For example, an image classifier should be invariant to small rotation, translation, perturbation, etc. When each constraint induces a kernel, how should a predictor or encoder be learned? We leave these problems to future work.

CODE

The code of Section 5 can be found at https://colab.research.google.com/drive/1m8OENF2lvxW3BB6CVEu45SGeK9IoYpd1?usp=sharing.

ACKNOWLEDGMENTS

We would like to thank Zico Kolter, Andrej Risteski, Bingbin Liu, Elan Rosenfeld, Shanda Li, Yuchen Li, Tanya Marwah, Ashwini Pokle, Amrith Setlur and Xiaoyu Huang for their feedback on the early draft of this work, and Yiping Lu and Fanghui Liu for their useful discussions. We are grateful to our anonymous ICLR reviewers, with whose help this work has been greatly improved. We acknowledge the support of NSF via IIS-1909816, IIS-2211907, ONR via N00014-23-1-2368, DARPA under cooperative agreement HR00112020003, and Bloomberg Data Science PhD fellowship.

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

APPENDIX

Table 2: Table of notations.

| Notation | Description |
|---|---|
| $\delta_{i,j}$ | Kronecker delta. Equal to 1 if $i = j$, and 0 otherwise |
| $\mathcal{X}$ | Input space, assumed to be a compact Hausdorff space |
| $\mathcal{Y}$ | Label space, assumed to be $\mathbb{R}$ |
| $P_{\mathcal{X}\mathcal{Y}}$ | Underlying data distribution over $\mathcal{X} \times \mathcal{Y}$ |
| $P_{\mathcal{X}}, P_{\mathcal{Y}}$ | Marginal distributions of $P_{\mathcal{X}\mathcal{Y}}$ |
| $dp(x)$ | A shorthand of $dP_{\mathcal{X}}(x)$ |
| $n, m$ | Numbers of labeled and unlabeled *i.i.d.* samples from $P_{\mathcal{X}\mathcal{Y}}$ and $P_{\mathcal{X}}$, respectively |
| $x_1, \cdots, x_{n+m}$ | $x_1, \cdots, x_n$ are the labeled samples, and $x_{n+1}, \cdots, x_{n+m}$ are unlabeled |
| $y_1, \cdots, y_n$ | The observed labels. Also define $\boldsymbol{y} = [y_1, \cdots, y_n] \in \mathbb{R}^n$ |
| $L^2(P_{\mathcal{X}})$ | $L^2$ function space *w.r.t.* $P_{\mathcal{X}}$, with inner product $\langle \cdot, \cdot \rangle_{P_{\mathcal{X}}}$ and norm $\| \cdot \|_{P_{\mathcal{X}}}$ |
| $K(x, x')$ | A Mercer base kernel that encodes inter-sample similarity information |
| $\boldsymbol{G}_K$ | $(n+m) \times (n+m)$ Gram matrix over all samples. $\boldsymbol{G}_K[i, j] = K(x_i, x_j)$ |
| $\boldsymbol{G}_{K,n}$ | $n \times n$ Gram matrix over the $n$ labeled samples |
| $\boldsymbol{G}_{K,m}$ | $m \times m$ Gram matrix over the $m$ unlabeled samples |
| $T_K$ | An integral operator *w.r.t.* $K$, $(T_K f)(x) = \int f(x')K(x, x')dp(x)$ |
| $(\lambda_i, \psi_i)$ | Eigenvalue and eigenfunction of $T_K$ such that $T_K \psi_i = \lambda_i \psi_i$. $\lambda_1 \geq \lambda_2 \geq \cdots \geq 0$ |
| $\mathbb{I}$ | The set $\{i \in \mathbb{N} : \lambda_i > 0\}$ |
| $K_s(x, x')$ | A spectrally transformed kernel of $K$, whose formula is given by Eqn. (3) |
| $s(\lambda)$ | The transformation function of $K_s(x, x')$ |
| $M$ | $s(\lambda) \leq M\lambda$ as proved in Theorem 1. Frequently used in other theorems |
| $K^p(x, x')$ | Equivalent to $K_s(x, x')$ with $s(\lambda) = \lambda^p$, for all $p \in \mathbb{R}$ |
| $\mathcal{H}_{K^p}$ | The RKHS associated with $K^p(x, x')$ when $p \geq 1$. $\mathcal{H}_{K^1}$ is also denoted by $\mathcal{H}_K$ |
| $\mathcal{H}_{K_s}$ | The RKHS associated with $K_s(x, x')$ that encodes target smoothness |
| $d_{K^p}$ | The kernel metric of $K^p$, defined as $d_{K^p}(x, x') = \sum_i \lambda_i^p(\psi_i(x) - \psi_i(x'))$ |
| $\overline{\mathrm{Lip}}_{d_{K^p}}(f)$ | The Lipschitz constant of $f$ *w.r.t.* metric $d_{K^p}$ over $\mathcal{X}$ defined in Section 2.1 |
| $r_{K^p}(f)$ | Defined as $\|f - \mathbb{E}_{P_{\mathcal{X}}}[f]\|_{P_{\mathcal{X}}}^2 / \overline{\mathrm{Lip}}_{d_{K^p}}(f)^2$, similar to the discriminant function |
| $r_t(f)$ | Defined as $\|f - \mathbb{E}_{P_{\mathcal{X}}}[f]\|_{P_{\mathcal{X}}}^2 / \overline{\mathrm{Lip}}_{d_t}(f)^2$. $d_t$ is defined in Section 2.1 |
| $f^*$ | Regression function defined as $f^*(x) := \int y \, dP_{\mathcal{X}\mathcal{Y}}(y\|x) \in L^2(P_{\mathcal{X}})$ |
| $\hat{f}$ | A predictor that approximates $f^*$. For STKR, its formula is given by Eqn. (4) |
| $B$ | The scale of $f^*$, defined as $\|f^*\|_{P_{\mathcal{X}}} \leq B$ |
| $\pi_p$ | The coefficient of $K_s$ when it is assumed to be polynomial in Assumption 1 |
| $\kappa^2$ | The upper bound of $K(x, x)$ for $P_{\mathcal{X}}$-*a.e.* $x \in \mathcal{X}$. See Assumption 2 |
| $\epsilon$ | $f^*$ is assumed to satisfy $r_t(f^*) \geq \epsilon^{-1}$ in Assumption 3 |
| $\sigma, L$ | Used in the moment condition in Assumption 4 |
| $\beta_n$ | The regularization coefficient in KRR and STKR. Can change with $n$ |
| $\hat{K}_s$ | A computable kernel used to approximate $K_s$, defined in Eqn. (5) |
| $\boldsymbol{v}_K(x)$ | An $\mathbb{R}^{n+m}$ vector. An important component of $\hat{K}_s$ defined after Eqn. (5) |
| $\tilde{f}$ | A predictor defined in Eqn. (6) that is inaccessible, but important in our analysis |
| $\tilde{\boldsymbol{\alpha}}, \hat{\boldsymbol{\alpha}}$ | Defined in Eqns. (6) and (7) |
| $\hat{\lambda}_1$ | The largest eigenvalue of $\frac{\boldsymbol{G}_K}{n+m}$ |
| $\gamma$ | The "step size" in STKR-Prop implemented with Richardson iteration |
| $\eta$ | A hyperparameter of inverse Laplacian, which is defined with $s^{-1}(\lambda) = \lambda^{-1} - \eta$ |
| $\hat{\Psi}$ | A pretrained $d$-dimensional representation $[\hat{\psi}_1, \cdots, \hat{\psi}_d]$ in representation learning |
| $\hat{\boldsymbol{w}}$ | A linear probe on top of $\hat{\Psi}$ trained during downstream, and $\hat{f}_d(x) = \hat{\boldsymbol{w}}^\top \hat{\Psi}(x)$ |
| $\mathcal{H}_{\hat{\Psi}}$ | A $d$-dimensional RKHS spanned by $\hat{\Psi}$, and it is a subspace of $\mathcal{H}_K$ |
| $\tilde{f}_d$ | Projection of $f^*$ onto $\mathcal{H}_{\hat{\Psi}}$ |

# A    RELATED WORK

## A.1    LEARNING WITH UNLABELED DATA

There are two broad classes of methods of learning with unlabeled data, namely semi-supervised learning and representation learning. Semi-supervised learning focuses on how to improve supervised learning by incorporating unlabeled samples, while representation learning focuses on how to extract a low-dimensional representation of data using huge unlabeled data.

Semi-supervised learning has long been used in different domains to enhance the performance of machine learning (Elworthy, 1994; Ranzato & Szummer, 2008; Shi & Zhang, 2011). While there are a wide variety of semi-supervised learning algorithms, the main differences between them lie in their way of realizing the assumption of consistency (Zhou et al., 2003), *i.e.* soft constraints that the predictor is expected to satisfy by prior knowledge. Enforcing consistency can be viewed as an auxiliary task (Goodfellow et al., 2013; Rasmus et al., 2015), since the unlabeled samples cannot be used in the "main task" involving the labels. The smoothness studied in this work can be regarded as one type of consistency.

While there are various types of consistency, most of them can be put into one of three categories: Euclidean-based consistency, prediction-based consistency and similarity-based consistency.

Euclidean-based consistency supposes $\mathcal{X}$ to be an Euclidean space, and relies on the assumption that $x, x'$ have the same label with high probability if they are close in the Euclidean distance, which is the foundation of many classical methods including nearest neighbor, clustering and RBF kernels. In the context of semi-supervised learning, virtual adversarial training (VAT) learns with unlabeled samples by perturbing them with little noise and forcing the model to give consistent outputs (Miyato et al., 2018). Mixup (Zhang et al., 2018) uses a variant of Euclidean-based consistency, which assumes that for two samples $(x_1, y_1)$ and $(x_2, y_2)$ and any $\theta \in (0, 1)$, the label of $\theta x_1 + (1 - \theta)x_2$ should be close to $\theta y_1 + (1 - \theta)y_2$. Mixup has been proved to be empirically successful, and was later further improved by Mixmatch (Berthelot et al., 2019).

Prediction-based consistency assumes that deep learning models enjoy good generalization: When a deep model is well trained on a small set of labeled samples, it should be able to achieve fairly good accuracy on the much larger set of unlabeled samples. The most simple method is pseudo labeling (Lee, 2013), which first trains a model only on the labeled samples, then uses it to label the unlabeled samples, and finally trains a second label on all samples. There are a large number of variants of pseudo labeling, also known as self-training. For instance, temporal ensembling (Laine & Aila, 2017) pseudo labels the unlabeled samples with models from previous epochs; Mean teacher (Tarvainen & Valpola, 2017) improves temporal ensembling for large datasets by using a model that averages the weights of previous models to generate pseudo labels; And NoisyStudent (Xie et al., 2020b), which does self-training iteratively with noise added at each iteration, reportedly achieves as high as 88.4% top-1 accuracy on ImageNet.

Finally, similarity-based consistency assumes prior knowledge about some kind of similarity over samples or transformations of the samples. A classical type of similarity-based consistency is based on graphs, and there is a rich body of literature on semi-supervised learning on graphs (Zhou et al., 2003; Johnson & Zhang, 2008) and GNNs (Kipf & Welling, 2016; Hamilton et al., 2017; Veličković et al., 2018). The most popular type of similarity-based consistency in deep learning is based on random data augmentation, also known as invariance-based consistency, where all augmentations of the same sample are assumed to be similar and thus should share the same label. This simple idea has been reported to achieve good performances by a lot of work, including UDA (Xie et al., 2020a), ReMixMatch (Berthelot et al., 2020) and FixMatch (Sohn et al., 2020). People have also experimented a variety of augmentation techniques, ranging from simple ones like image translation, flipping and rotation to more complicated ones like Cutout (DeVries & Taylor, 2017), AutoAugment (Cubuk et al., 2019) and RandAugment (Cubuk et al., 2020).

Representation learning aims to learn low-dimensional representations of data that concisely encodes relevant information useful for building classifiers or other predictors (Bengio et al., 2013). By this definition, in order to learn a good representation, we need to have information about what classifiers or predictors we want to build, for which consistency also plays a very important role. Popular representation learning algorithms based on consistency can be roughly classified into Euclidean-

based or similarity-based consistency. There is no prediction-based consistency for unsupervised representation learning because there is nothing to "predict" in the first place.

Euclidean-based consistency has been widely applied in manifold learning, such as LLE (Roweis & Saul, 2000), Isomap (Tenenbaum et al., 2000) and Laplacian eigenmaps (Belkin & Niyogi, 2003), whose goal is to determine the low-dimensional manifold on which the observed high-dimensional data resides. See Bengio et al. (2004) for descriptions of these methods.

Similarity-based consistency is the basis of most deep representation learning algorithms. In fact, Euclidean-based consistency can be viewed as a special case of similarity-based consistency, which assumes near samples under the Euclidean distance to be similar. The most obvious similarity-based method is contrastive learning (Oord et al., 2018; Hjelm et al., 2019; Chen et al., 2020), which requires the model to assign similar representations to augmentations of the same sample. But in fact, lots of representation learning methods that make use of certain auxiliary tasks can be viewed as enforcing similarity-based consistency. If the auxiliary task is to learn a certain multi-dimensional target function $g(x)$, then by defining kernel $K_g(x, x') := \langle g(x), g(x') \rangle$, the task can also be seen as approximating $K_g$, and $K_g$ encodes some kind of inter-sample similarity. For example, even supervised pretraining such as ImageNet pretraining, can be viewed as enforcing similarity-based consistency (defined by the ImageNet labels) over the data. In fact, Papyan et al. (2020) showed that if supervised pretraining with cross entropy loss is run for sufficiently long time, then the representations will *neural collapse* to the class labels (*i.e.* samples of the same class share exactly the same representation), which is a natural consequence of enforcing class-based similarity. Zhai et al. (2024) also showed that mask-based auxiliary tasks such as BERT (Devlin et al., 2019) and MAE (He et al., 2022) can also be viewed as approximating a similarity kernel.

Finally, for semi-supervised and self-supervised learning on graphs, diffusion has been widely used, from the classical works of Page et al. (1999); Kondor & Lafferty (2002) to more recent papers such as Gasteiger et al. (2019); Hassani & Khasahmadi (2020). For graph tasks, STKR can be viewed as a generalization of these diffusion-based methods.

## A.2 STATISTICAL LEARNING THEORY ON KERNEL METHODS

Kernel methods have a very long history and there is a very rich body of literature on theory pertaining to kernels, which we shall not make an exhaustive list here. We point our readers who want to learn more about kernels to two books that are referred to a lot in this work: Schölkopf & Smola (2002); Steinwart & Christmann (2008). Also, Chapters 12-14 of Wainwright (2019) can be more helpful and easier to read for elementary readers.

Here we would like to briefly talk about recent theoretical progress on kernel methods, especially on interpolation Sobolev spaces. This work uses the results in Fischer & Steinwart (2020), in which minimax optimal learning rates for regularized least-squares regression under Sobolev norms were proved. Meanwhile, Lin et al. (2020) proved that KRR converges to the Bayes risk at the best known rate among kernel-based algorithms. Addition follow-up work includes Jun et al. (2019); Cui et al. (2021); Talwai et al. (2022); Li et al. (2022); de Hoop et al. (2023); Jin et al. (2023).

However, there are also counter arguments to this line of work, and kernel-based learning is by no means a solved problem. As Jun et al. (2019) pointed out, these optimal rates only match the lower bounds under certain assumptions (Steinwart et al., 2009). Besides, most of these minimax optimal learning rates are for KRR, which requires a greater-than-zero regularization term, but empirical observations suggest that kernel regression with regularization can perform equally well, if not better (Zhang et al., 2017; Belkin et al., 2018; Liang & Rakhlin, 2020). What makes things even more complicated is that kernel "ridgeless" regression seems to only work for high-dimensional data under some conditions, as argued in Rakhlin & Zhai (2019); Buchholz (2022) who showed that minimum-norm interpolation in the RKHS *w.r.t.* Laplace kernel is not consistent if the input dimension is constant. Overall, while kernel methods have a very long history, they still have lots of mysteries, which we hope that future work can shed light on.

Another class of methods in parallel with STKR is random features, first introduced in the seminal work Rahimi & Recht (2007). We refer our readers to Liu et al. (2021) for a survey on random features. The high-level idea of random features is the following: One first constructs a class of features $\{\varphi(x, \theta)\}$ parameterized by $\theta$, *e.g.* Fourier features. Then, randomly sample $D$ features from

this class, and perform KRR or other learning algorithms on this set of random features. Now, why would such method have good generalization? As explained in Mei et al. (2022), if $D$ is very large (overparameterized regime), then the approximation error is small since $f^*$ should be well represented by the $D$ features, but the estimation error is large because more features means more complexity; If $D$ is very small (underparameterized regime), then the model is simple so the estimation error is small, but the approximation error could be large because the $D$ features might be unable to represent $f^*$. However, since the class of features $\{\varphi(x,\theta)\}$ is of our choice, we can choose them to be "good" features (*e.g.* with low variance) so that the estimation error will not be very large even if $D$ is big. Rahimi & Recht (2007) used Fourier features that are good, and there are lots of other good features we could use. One can even try to learn the random features: For example, Sinha & Duchi (2016) proposed to learn a distribution over a pre-defined class of random features such that the resulting kernel aligns with the labels.

## B  PROOFS

### B.1  PROOF OF PROPOSITION 1

*Proof.* First, for all $f \in \mathcal{H}_{K^p}$, and $\mu, \nu \in \bar{\mathcal{X}}$ we have

$$\left| \bar{f}(\mu) - \bar{f}(\nu) \right| = \left| \int_{\mathcal{X}} f(x) d\mu(x) - \int_{\mathcal{X}} f(x) d\nu(x) \right| = \left| \left\langle f, \int_{\mathcal{X}} K_x^p d\mu(x) - \int_{\mathcal{X}} K_x^p d\nu(x) \right\rangle_{\mathcal{H}_{K^p}} \right|$$

$$\leq \|f\|_{\mathcal{H}_{K^p}} \left\| \int_{\mathcal{X}} K_x^p d\mu(x) - \int_{\mathcal{X}} K_x^p d\nu(x) \right\|_{\mathcal{H}_{K^p}} = \|f\|_{\mathcal{H}_{K^p}} d_{K^p}(\mu, \nu),$$

which means that $\overline{\mathrm{Lip}}_{d_{K^p}}(f) \leq \|f\|_{\mathcal{H}_{K^p}}$.

Second, $\{K_x^p - K_{x'}^p\}_{x,x' \in \mathcal{X}}$ span the entire $\mathcal{H}_{K^p}$, because if $f \in \mathcal{H}_{K^p}$ satisfies $\langle f, K_x^p - K_{x'}^p \rangle_{\mathcal{H}_{K^p}} = 0$ for all $x, x'$, then $f \equiv c$ for some $c$, which means that $f \equiv 0$ since $\mathbf{0}$ is the only constant function in $\mathcal{H}_{K^p}$ (because $K$ is centered). Thus, any $f \in \mathcal{H}_{K^p}$ can be written as $f = \iint (K_x^p - K_{x'}^p) d\xi(x, x')$ for some finite signed measure $\xi$ over $\mathcal{X} \times \mathcal{X}$, and thus $f = \int_{\mathcal{X}} K_x^p d\mu(x) - \int_{\mathcal{X}} K_x^p d\nu(x)$, where $\mu, \nu$ are the marginal measures of $\xi$. By using such defined $\mu, \nu$ in the above formula, we can see that $\overline{\mathrm{Lip}}_{d_{K^p}}(f) = \|f\|_{\mathcal{H}_{K^p}}$. □

### B.2  PROOF OF THEOREM 1

First, we show that $\mathcal{H}_t$ must be an RKHS. Since $\psi_1$ is the top-1 eigenfunction of $K^p$ for all $p \geq 1$, we have $r_{K^p}(\psi_1) \geq r_{K^p}(f)$ for all $f \in \mathcal{H}_K$, which implies that $r_t(\psi_1) \geq r_t(f)$ for all $f \in \mathcal{H}_t \subset \mathcal{H}_K$. Let $C_0 := r_t(\psi_1)$. Then, for all $f \in \mathcal{H}_t$, since $f$ must be centered, we have $\|f\|_{P_{\mathcal{X}}}^2 \leq C_0 \cdot \overline{\mathrm{Lip}}_{d_t}(f)^2$. Let $\tilde{f} = f / \|f\|_{\mathcal{H}_t}$, then:

$$\|f\|_{P_{\mathcal{X}}} \leq \sqrt{C_0} \cdot \sup_{x,x' \in \overline{\mathcal{X}}, x \neq x'} \inf_{\|f_1\|_{\mathcal{H}_t} = 1} \frac{|f(x) - f(x')|}{|f_1(x) - f_1(x')|}$$

$$\leq \sqrt{C_0} \cdot \sup_{x,x' \in \overline{\mathcal{X}}, x \neq x'} \frac{|f(x) - f(x')|}{|\tilde{f}(x) - \tilde{f}(x')|} = \sqrt{C_0} \|f\|_{\mathcal{H}_t}.$$

This implies that $\| \cdot \|_{\mathcal{H}_t}$ is stronger than $\| \cdot \|_{P_{\mathcal{X}}}$. Thus, if there is a sequence of points $\{x_i\}$ such that $x_i \to x$ w.r.t. $\| \cdot \|_{\mathcal{H}_t}$, then (a) $x \in \mathcal{H}_t$ because $\mathcal{H}_t$ is a Hilbert space, and (b) $x_i \to x$ w.r.t. $\| \cdot \|_{P_{\mathcal{X}}}$. Meanwhile, we know that $\| \cdot \|_{\mathcal{H}_K}$ is stronger than $\| \cdot \|_{P_{\mathcal{X}}}$, so $x_i \to x$ w.r.t. $\| \cdot \|_{\mathcal{H}_K}$ also implies $x_i \to x$ w.r.t. $\| \cdot \|_{P_{\mathcal{X}}}$.

Now, consider the inclusion map $I : \mathcal{H}_t \to \mathcal{H}_K$, such that $Ix = x$. For any sequence $\{x_i\} \subset \mathcal{H}_t$ such that $x_i \to x$ w.r.t. $\| \cdot \|_{\mathcal{H}_t}$ and $Ix_i \to y$ w.r.t. $\| \cdot \|_{\mathcal{H}_K}$, we have $x_i \to x$ w.r.t. $\| \cdot \|_{P_{\mathcal{X}}}$, and $x_i \to y$ w.r.t. $\| \cdot \|_{P_{\mathcal{X}}}$. Thus, we must have $y = x = Ix$, meaning that the graph of $I$ is closed. So the closed graph theorem says that $I$ must be a bounded operator, meaning that there exists a constant $C > 0$ such that $\|f\|_{\mathcal{H}_K} \leq C \|f\|_{\mathcal{H}_t}$ for all $f \in \mathcal{H}_t$. (For an introduction of the closed graph theorem, see Chapter 2 of Brezis (2011).)

For all $f \in \mathcal{H}_K$, let $\delta_x : f \mapsto f(x)$ be the evaluation functional at point $x$. Since $\mathcal{H}_K$ is an RKHS, for any $x \in \mathcal{X}$, $\delta_x$ is a bounded functional on $\mathcal{H}_K$, which means that there exists a constant

$M_x > 0$ such that $|f(x)| \le M_x \|f\|_{\mathcal{H}_K}$ for all $f \in \mathcal{H}_K$. So for any $f \in \mathcal{H}_t \subset \mathcal{H}_K$, there is $|f(x)| \le M_x \|f\|_{\mathcal{H}_K} \le M_x C \|f\|_{\mathcal{H}_t}$, which means that $\delta_x$ is also a bounded functional on $\mathcal{H}_t$. Thus, $\mathcal{H}_t$ must be an RKHS. Let $K_s$ be the reproducing kernel of $\mathcal{H}_t$. From now on, we will use $\mathcal{H}_{K_s}$ to denote $\mathcal{H}_t$. Now that $\mathcal{H}_{K_s}$ is an RKHS, we can use the proof of Proposition 1 to show that $\overline{\mathrm{Lip}}_{d_t}(f) = \|f\|_{\mathcal{H}_{K_s}}$, which implies that $r_t(f) = \|f - \mathbb{E}_{P_{\mathcal{X}}}[f]\|_{P_{\mathcal{X}}}^2 / \|f\|_{\mathcal{H}_{K_s}}^2$.

Second, we prove by induction that $\psi_1, \cdots, \psi_d$ are the top-$d$ eigenfunctions of $K_s$, and they are pairwise orthogonal *w.r.t.* $\mathcal{H}_{K_s}$. When $d = 1$, $\psi_1$ is the top-1 eigenfunction of $K^p$ for all $p \ge 1$, so $\psi_1 \in \arg\max_{f \in L^2(P_{\mathcal{X}})} r_{K^p}(f)$. By the relative smoothness preserving assumption, this implies that $\psi_1 \in \arg\max_{f \in L^2(P_{\mathcal{X}})} r_t(f)$. Therefore, $\psi_1$ must be the top-1 eigenfunction of $K_s$. Now for $d \ge 2$, suppose $\psi_1, \cdots, \psi_{d-1}$ are the top-$(d-1)$ eigenfunctions of $K_s$ with eigenvalues $s_1, \cdots, s_{d-1}$, and they are pairwise orthogonal *w.r.t.* $\mathcal{H}_{K_s}$. Let $\mathcal{H}_0 = \{f : \langle f, \psi_i \rangle_{P_{\mathcal{X}}} = 0, \forall i \in [d-1]\}$. Obviously, $\mathcal{H}_0 \cap \mathcal{H}_{K^p}$ is a closed subspace of $\mathcal{H}_{K^p}$ for all $p \ge 1$. And for any $f \in \mathcal{H}_0 \cap \mathcal{H}_{K_s}$ and any $i \in [d-1]$, we have $\langle f, \psi_i \rangle_{\mathcal{H}_{K_s}} = f^\top (K_s^{-1} \psi_i) = f^\top s_i^{-1} \psi_i = 0$, so $\mathcal{H}_{K_s} \cap \mathcal{H}$ is a closed subspace of $\mathcal{H}_{K_s}$. Applying the assumption with this $\mathcal{H}_0$, we can see that $\psi_d$ is the top-$d$ eigenfunction of $K_s$, and is orthogonal to $\psi_1, \cdots, \psi_{d-1}$ *w.r.t.* both $\mathcal{H}_{K_s}$. Thus, we complete our proof by induction. And if $\lambda_d = \lambda_{d+1}$, then we can show that both $\psi_1, \cdots, \psi_{d-1}, \psi_d$ and $\psi_1, \cdots, \psi_{d-1}, \psi_{d+1}$ are the top-$d$ eigenfunctions of $K_s$, meaning that $s_d = s_{d+1}$. Thus, $K_s$ can be written as $K_s(x, x') = \sum_i s_i \psi_i(x) \psi_i(x')$, where $s_1 \ge s_2 \ge \cdots \ge 0$, and $s_i = s_{i+1}$ if $\lambda_i = \lambda_{i+1}$. Moreover, by $\mathcal{H}_{K_s} \subset \mathcal{H}_K$, it is obviously true that $\lambda_i = 0$ implies $s_i = 0$.

Third, we prove by contradiction that $s_i \le M\lambda_i$ for all $i$. If this statement is false, then obviously one can find $t_1 < t_2 < \cdots$ such that $s_{t_i} \ge i \cdot \lambda_{t_i}$ for all $i$. Consider $f = \sum_i \sqrt{i^{-1} \lambda_{t_i}} \psi_i$, for which $\|f\|_{\mathcal{H}_K}^2 = \sum_i i^{-1} = +\infty$. Since $\mathcal{H}_{K_s} \subset \mathcal{H}_K$, this implies that $\|f\|_{\mathcal{H}_{K_s}}^2 = +\infty$, so we have $+\infty = \sum_i \frac{\lambda_{t_i}}{i \cdot s_{t_i}} \le \sum_i \frac{\lambda_{t_i}}{i^2 \lambda_{t_i}} = \sum_i \frac{1}{i^2} < +\infty$, which gives a contradiction and proves the claim.

Fourth, we find a function $s(\lambda)$ that satisfies the conditions in the theorem to interpolate those points. Before interpolation, we first point out that we can WLOG assume that $\lambda_i < 2\lambda_{i+1}$ for all $i$: If there is an $i$ that does not satisfy this condition, we simply insert some new $\lambda$'s between $\lambda_i$ and $\lambda_{i+1}$, whose corresponding $s$'s are the linear interpolations between $s_i$ and $s_{i+1}$, so that $s_i \le M\lambda_i$ still holds. With this assumption, it suffices to construct a series of bump functions $\{f_i\}_{i=1}^\infty$, where $f_i \equiv 0$ if $\lambda_i = \lambda_{i+1}$; otherwise, $f_i(\lambda) = s_i - s_{i+1}$ for $\lambda \ge \lambda_i$ and $f_i(\lambda) = 0$ for $\lambda \le \lambda_{i+1}$. Such bump functions are $C^\infty$ and monotonically non-decreasing. Then, define $s(\lambda) = \sum_i f_i(\lambda)$ for $\lambda > 0$, and $s(0) = 0$. This sum of bump functions converges everywhere on $(0, +\infty)$, since it is a finite sum locally everywhere. Clearly this $s$ is monotonic, interpolates all the points, continuous on $[0, +\infty)$ and $C^\infty$ on $(0, +\infty)$. And for all $\lambda$ that is not $\lambda_i$, for instance $\lambda \in (\lambda_{i+1}, \lambda_i)$, there is $s(\lambda) \le s(\lambda_i) \le M\lambda_i \le 2M\lambda_{i+1} \le 2M\lambda$. Thus, $s(\lambda) = O(\lambda)$ for $\lambda \in [0, +\infty)$. $\qquad\square$

*Remark.* In general, we cannot guarantee that $s(\lambda)$ is differentiable at $\lambda = 0$. Here is a counterexample: $\lambda_i = 3^{-i}$, and $s_i = 3^{-i}$ if $i$ is odd and $2 \cdot 3^{-i}$ if $i$ is even. Were $s(\lambda)$ to be differentiable at $\lambda = 0$, its derivative would be 1 and also would be 2, which leads to a contradiction. Nevertheless, if the target smoothness is strictly stronger than base smoothness, *i.e.* $\overline{\mathrm{Lip}}_{d_K}(f) = o(\overline{\mathrm{Lip}}_{d_t}(f))$, then $s$ can be differentiable at $\lambda = 0$ but still not $C^\infty$.

**Link with discriminant analysis and the Poincaré constant.** First, we point out the connection between $r_{K^p}(f)$ and the discriminant function in discriminant analysis (see Chapters 3-4 of Huberty & Olejnik (2006)). We can see that $r_{K^p}(f)$ is defined as the proportion of variance of $f$ *w.r.t.* $L^2(P_{\mathcal{X}})$ and *w.r.t.* $\mathcal{H}_{K^p}$, so essentially $r_{K^p}(f)$ measures how much variance of $f$ is kept by the inclusion map $\mathcal{H}_{K^p} \hookrightarrow L^2(P_{\mathcal{X}})$. Meanwhile, the discriminant function is the proportion of variance of $f$ in the grouping variable and in total. Thus, similar to PCA which extracts the $d$ features that keep the most variance (*i.e.* the top-$d$ singular vectors), kernel PCA also extracts the $d$ features that keep the most variance (*i.e.* the top-$d$ eigenfunctions). This is also closely related to the ratio trace defined in Zhai et al. (2024), whose Appendix C showed that lots of existing contrastive learning methods can be viewed as maximizing the ratio trace, *i.e.* maximizing the variance kept. Zhai et al. (2024) also showed that for supervised pretraining with multi-class classification, we can also define an "augmentation kernel", and then $r_{K^p}(f)$ is equivalent to the discriminant function (*i.e.* the $\eta^2$ defined in Section 4.2.1 of Huberty & Olejnik (2006)).

Moreover, $r_{K^p}(f)$ defined for the interpolation Sobolev space $\mathcal{H}_{K^p}$ is analogous to the Poincaré constant for the Sobolev space. Specifically, the Poincaré-Wirtinger inequality states that for any

$1 \leq p < \infty$ and any bounded connected open set of $C^1$ functions $\Omega$ , there exists a constant $C$ depending on $p$ and $\Omega$ such that for all $u$ in the Sobolev space $W^{1,p}(\Omega)$, there is $\|u - \bar{u}\|_{L^p(\Omega)} \leq C\|\nabla u\|_{L^p(\Omega)}$ where $\bar{u} = \frac{1}{|\Omega|} \int_\Omega u$ is the mean, and the smallest value of such $C$ is called the Poincaré constant (see Chapter 9.4, Brezis (2011)). Consider the special case $p = 2$, and replace $L^2(\Omega)$ with $L^2(\mu)$ for a probability measure $d\mu$ such that $d\mu(x) = \exp(-V(x))dx$, where $V$ is called the potential function. Then, with proper boundary conditions, we can show with integration by parts that $\|\nabla u\|_{L^2(\mu)}^2 = \langle u, \mathcal{L}u \rangle_{L^2(\mu)}$, where $\mathcal{L}$ is the diffusion operator defined as $\mathcal{L}f = -\Delta f + \nabla V \cdot \nabla f$ for all $f$. Here, $\Delta = \sum_j \frac{\partial^2}{\partial x_j^2}$ is the Laplace-Beltrami operator. Now, by replacing $\mathcal{L}$ with the integral operator $T_{K^p}$, we can see that $C^2$ is equivalent to $\sup_f r_{K^p}(f)$ for the interpolation Sobolev space $\mathcal{H}_{K^p}$, which is known to be $r_{K^p}(\psi_1) = \lambda_1$ if $f \in L^2(P_\mathcal{X})$, and $r_{K^p}(\psi_j) = \lambda_j$ if $f$ is orthogonal to $\psi_1, \cdots, \psi_{j-1}$ (*i.e.* $\mathcal{H}_0$ defined in the proof above).

## B.3   CONNECTION BETWEEN TRANSDUCTIVE AND INDUCTIVE FOR INVERSE LAPLACIAN

**Proposition 5.** *Let $\mathcal{X} = \{x_1, \cdots, x_{n+m}\}$, and $\mathcal{G}$ be a graph with node set $\mathcal{X}$ and edge weights $w_{ij}$. Define $\boldsymbol{W}, \boldsymbol{D} \in \mathbb{R}^{(n+m) \times (n+m)}$ as $\boldsymbol{W}[i,j] = w_{ij}$, and $\boldsymbol{D}$ be a diagonal matrix such that $\boldsymbol{D}[i,i] = \sum_j w_{ij}$. Suppose $\boldsymbol{W}$ is p.s.d., and $\boldsymbol{D}[i,i] > 0$ for all $i$. Let $\boldsymbol{L} := \boldsymbol{I}_{n+m} - \eta \boldsymbol{D}^{-1/2} \boldsymbol{W} \boldsymbol{D}^{-1/2}$ for a constant $\eta$. Let $P_\mathcal{X}$ be a distribution over $\mathcal{X}$ such that $p(x_i) = \frac{\boldsymbol{D}[i,i]}{\text{Tr}(\boldsymbol{D})}$. Define a kernel $K : \mathcal{X} \times \mathcal{X} \to \mathbb{R}$ as $K(x_i, x_j) = \text{Tr}(\boldsymbol{D})(\boldsymbol{D}^{-1} \boldsymbol{W} \boldsymbol{D}^{-1})[i,j]$. For any $\boldsymbol{u} \in \mathbb{R}^{n+m}$ and $\hat{\boldsymbol{y}} = \left(\boldsymbol{D}^{-\frac{1}{2}} \boldsymbol{W} \boldsymbol{D}^{-\frac{1}{2}}\right)^{\frac{1}{2}} \boldsymbol{u}$, define $f : \mathcal{X} \to \mathbb{R}$ as: $[f(x_1), \cdots, f(x_{n+m})] = \sqrt{\text{Tr}(\boldsymbol{D})} \boldsymbol{D}^{-\frac{1}{2}} \left(\boldsymbol{D}^{-\frac{1}{2}} \boldsymbol{W} \boldsymbol{D}^{-\frac{1}{2}}\right)^{\frac{1}{2}} \hat{\boldsymbol{y}}$. Then, we have*

$$\hat{\boldsymbol{y}}^\top \boldsymbol{L} \hat{\boldsymbol{y}} = \|f\|_{\mathcal{H}_K}^2 - \eta \|f\|_{P_\mathcal{X}}^2 = \sum_{i,j} f(x_i) f(x_j) K^{-1}(x_i, x_j) p(x_i) p(x_j) - \eta \sum_i f(x_i)^2 p(x_i).$$

*Proof.* Denote $\boldsymbol{f} := [f(x_1), \cdots, f(x_{n+m})]$. Let $\boldsymbol{G}_{K^{-1}}[i,j] = K^{-1}(x_i, x_j)$, *i.e.* $\boldsymbol{G}_{K^{-1}}$ is the Gram matrix of $K^{-1}$. Let $\boldsymbol{P} = \boldsymbol{D}/\text{Tr}(\boldsymbol{D}) = \text{diag}\{p(x_1), \cdots, p(x_{n+m})\}$. Then, we have

$$\|f\|_{\mathcal{H}_K}^2 - \eta \|f\|_{P_\mathcal{X}}^2 = \boldsymbol{f}^\top \boldsymbol{P} \boldsymbol{G}_{K^{-1}} \boldsymbol{P} \boldsymbol{f} - \eta \boldsymbol{f}^\top \boldsymbol{P} \boldsymbol{f}.$$

Let us first characterize $\boldsymbol{G}_{K^{-1}}$. For any $f \in \mathcal{H}_K$, there is

$$(\boldsymbol{G}_K \boldsymbol{P} \boldsymbol{G}_{K^{-1}} \boldsymbol{P} \boldsymbol{f})[t] = \sum_{i,j} f(x_i) K^{-1}(x_i, x_j) K(x_j, x_t) p(x_i) p(x_j)$$

$$= \sum_i f(x_i) K^0(x_i, x_t) p(x_i) = f(x_t),$$

meaning that $\boldsymbol{G}_K \boldsymbol{P} \boldsymbol{G}_{K^{-1}} \boldsymbol{P} \boldsymbol{f} = \boldsymbol{f}$. Moreover, let $\boldsymbol{w} = \boldsymbol{D}^{\frac{1}{2}} \boldsymbol{u}$, then $\boldsymbol{f} = \sqrt{\text{Tr}(\boldsymbol{D})} \boldsymbol{D}^{-1} \boldsymbol{W} \boldsymbol{D}^{-1} \boldsymbol{w} = \text{Tr}(\boldsymbol{D})^{-\frac{1}{2}} \boldsymbol{G}_K \boldsymbol{w}$, so we have

$$\boldsymbol{f}^\top \boldsymbol{P} \boldsymbol{G}_{K^{-1}} \boldsymbol{P} \boldsymbol{f} = \text{Tr}(\boldsymbol{D})^{-\frac{1}{2}} \boldsymbol{w}^\top \boldsymbol{G}_K \boldsymbol{P} \boldsymbol{G}_{K^{-1}} \boldsymbol{P} \boldsymbol{f} = \text{Tr}(\boldsymbol{D})^{-\frac{1}{2}} \boldsymbol{w}^\top \boldsymbol{f} = \boldsymbol{u}^\top \left(\boldsymbol{D}^{-\frac{1}{2}} \boldsymbol{W} \boldsymbol{D}^{-\frac{1}{2}}\right)^{\frac{1}{2}} \hat{\boldsymbol{y}} = \|\hat{\boldsymbol{y}}\|_2^2.$$

Besides, $\boldsymbol{f}^\top \boldsymbol{P} \boldsymbol{f} = \boldsymbol{f}^\top \boldsymbol{D} \text{Tr}(\boldsymbol{D})^{-1} \boldsymbol{f} = \hat{\boldsymbol{y}}^\top \boldsymbol{D}^{-\frac{1}{2}} \boldsymbol{W} \boldsymbol{D}^{-\frac{1}{2}} \hat{\boldsymbol{y}}$. So $\|f\|_{\mathcal{H}_K}^2 - \eta \|f\|_{P_\mathcal{X}}^2 = \hat{\boldsymbol{y}}^\top \boldsymbol{L} \hat{\boldsymbol{y}}$. $\quad\square$

*Remark.* The definition of $K$, *i.e.* $\boldsymbol{G}_K = \text{Tr}(\boldsymbol{D}) \boldsymbol{D}^{-1} \boldsymbol{W} \boldsymbol{D}^{-1}$, has a similar form as the positive-pair kernel in Johnson et al. (2023), Eqn. (1). The important difference between this and the normalized adjacency matrix $\boldsymbol{D}^{-\frac{1}{2}} \boldsymbol{W} \boldsymbol{D}^{-\frac{1}{2}}$ is that it has $\boldsymbol{D}^{-1}$ instead of $\boldsymbol{D}^{-\frac{1}{2}}$. However, the above result says that using a kernel with Gram matrix $\boldsymbol{D}^{-1} \boldsymbol{W} \boldsymbol{D}^{-1}$ in the inductive setting is equivalent to using the matrix $\boldsymbol{D}^{-\frac{1}{2}} \boldsymbol{W} \boldsymbol{D}^{-\frac{1}{2}}$ in the transductive setting. Moreover, this result assumes that $\hat{\boldsymbol{y}}$ belongs to the column space of $\boldsymbol{D}^{-\frac{1}{2}} \boldsymbol{W} \boldsymbol{D}^{-\frac{1}{2}}$ (which is what $\boldsymbol{u}$ is used for). This is necessary for $\hat{\boldsymbol{y}}$ to be representable by an $f \in \mathcal{H}_K$; otherwise, $\hat{\boldsymbol{y}}^\top \hat{\boldsymbol{y}}$ cannot be expressed by any $f \in \mathcal{H}_K$.

### B.4 PROOF OF THEOREM 2

First of all, note that for any $f = \sum_i u_i \psi_i \in \mathcal{H}_K$ such that $\|f\|_{\mathcal{H}_K} \leq T$, there is $f(x)^2 = \left(\sum_i u_i \psi_i(x)\right)^2 \leq \left(\sum_i \frac{u_i^2}{\lambda_i}\right)\left(\sum_i \lambda_i \psi_i(x)^2\right) \leq T^2 \kappa^2$, i.e. $|f(x)| \leq \kappa T$ for $P_{\mathcal{X}}$-almost all $x$. And for any $f \in \mathcal{H}_{K_s}$ such that $\|f\|_{\mathcal{H}_{K_s}} \leq T$, by Theorem 1 we have $\|f\|_{\mathcal{H}_K} \leq \sqrt{M}T$, so there is $|f(x)| \leq \kappa\sqrt{M}T$ for $P_{\mathcal{X}}$-almost all $x$.

The main tool to prove this result is Theorem 3.1 in Fischer & Steinwart (2020), stated below:

**Theorem 6** (Theorem 3.1, Fischer & Steinwart (2020))**.** *Let $P_{\mathcal{X}\mathcal{Y}}, P_{\mathcal{X}}$ and the regression function $f^*$ be defined as in Section 2. Let $\mathcal{H}_K$ be a separable RKHS on $\mathcal{X}$ with respect to a measurable and bounded kernel $K$, and $K(x, x) \leq \kappa^2$ for $P_{\mathcal{X}}$-almost all $x$. Define the integral operator $T_K : L^2(P_{\mathcal{X}}) \to L^2(P_{\mathcal{X}})$ as $(T_K f)(x) = \int f(x')K(x, x')dp(x')$. Let the eigenvalues/functions of $T_K$ be $\lambda_i, \psi_i$, with $\lambda_1 \geq \lambda_2 \geq \cdots$. Let $\mathcal{H}_{K^p}$ be defined as Eqn. (1). Assume that there exists a constant $B_\infty > 0$ such that $\|f^*\|_{L^\infty(P_{\mathcal{X}})} \leq B_\infty$, and the following four conditions holds:*

- ***Eigenvalue decay (EVD)**: $\lambda_i \leq c_1 i^{-\frac{1}{p}}$ for some constant $c_1 > 0$ and $p \in (0, 1]$.*

- ***Embedding condition (EMB)** for $\alpha \in (0, 1]$: The inclusion map $\mathcal{H}_{K^\alpha} \hookrightarrow L^\infty(P_{\mathcal{X}})$ is bounded, with $\|\mathcal{H}_{K^\alpha} \hookrightarrow L^\infty(P_{\mathcal{X}})\| \leq c_2$ for some constant $c_2 > 0$.*

- ***Source condition (SRC)** for $\beta \in (0, 2]$: $\|f^*\|_{\mathcal{H}_{K^\beta}} \leq c_3$ for some constant $c_3 > 0$.*

- ***Moment condition (MOM)**: There exist constants $\sigma, L > 0$ such that for $P_{\mathcal{X}}$-almost all $x \in \mathcal{X}$ and all $r \geq 2$, $\int |y - f^*(x)|^r p(dy|x) \leq \frac{1}{2}r!\sigma^2 L^{r-2}$.*

*Let $\tilde{f}$ be the KRR predictor with $\beta_n > 0$. Let $\gamma$ be any constant such that $\gamma \in [0, 1]$ and $\gamma < \beta$. If $\beta + p > \alpha$, and $\beta_n = \Theta(n^{-\frac{1}{\beta+p}})$, then there is a constant $A > 0$ independent of $n \geq 1$ and $\tau > 0$ such that:*

$$\left\|\tilde{f} - f^*\right\|_{\mathcal{H}_{K^\gamma}} \leq 2c_3^2\beta_n^{\beta-\gamma} + A\tau^2\left[\frac{(\sigma^2\kappa^2 + c_2^2 c_3^2)}{n\beta_n^{\gamma+p}} + \frac{c_2^2\max\left\{L^2, (B_\infty + c_2 c_3)^2\right\}}{n^2\beta_n^{\alpha+\gamma+(\alpha-\beta)_+}}\right] \quad (11)$$

*holds for sufficiently large $n$ with $P_{\mathcal{X}}^n$-probability at least $1 - 4e^{-\tau}$.*

This exact bound can be derived from the proof in Sections 6.1-6.10 of Fischer & Steinwart (2020). We apply this result by substituting $K = K_s$, $\alpha = 1$, $\beta = 1$, and $\gamma = 0$. Note that $\|f\|_{\mathcal{H}_{K^0}} = \|f\|_{P_{\mathcal{X}}}$ for $f \in \mathcal{H}_{K^0}$, and we have proved that $\tilde{f} - f^* \in \mathcal{H}_{K_s} \subset \mathcal{H}_{K^0}$. For the four conditions, we have:

- **Eigenvalue decay (EVD)**: This is assumed to be satisfied by condition.
- **Embedding condition (EMB)** for $\alpha = 1$: $\|\mathcal{H}_{K_s} \hookrightarrow L^\infty(P_{\mathcal{X}})\| \leq c_2$ for some constant $c_2 > 0$. This condition is satisfied with $c_2 = \kappa\sqrt{M}$, as mentioned at the beginning of this proof.
- **Source condition (SRC)** for $\beta = 1$: $\|f^*\|_{\mathcal{H}_{K_s}} \leq c_3$ for some constant $c_3 > 0$. By Assumption 3, this is satisfied with $c_3 = \sqrt{\epsilon}B$.
- **Moment condition (MOM)**: This is assumed to be satisfied by condition.

Finally, we have $K_s(x, x) \leq M\kappa^2$ a.e., and $B_\infty = \kappa\sqrt{M\epsilon}B$, as mentioned at the beginning of this proof. Thus, applying this result yields the desired bound. $\square$

### B.5 BOUNDING THE GAP BETWEEN $\hat{K}_s$ AND $K_s$

**Lemma 6.** *For any $\delta > 0$, the following holds with probability at least $1 - \delta$ for all $p \geq 1$:*

$$\left|\hat{K}^p(x, x_j) - K^p(x, x_j)\right| \leq (p-1)\lambda_{\max}^{p-2}\frac{\kappa^4}{\sqrt{n+m}}\left(2 + \sqrt{2\log\frac{1}{\delta}}\right) \quad (12)$$

*for all $x \in \mathcal{X}, j \in [n+m]$, and $\lambda_{\max} = \max\left\{\lambda_1, \hat{\lambda}_1\right\}$, which implies that*

$$\left|\hat{K}_s(x, x_j) - K_s(x, x_j)\right| \leq \left.\nabla_\lambda\left(\frac{s(\lambda)}{\lambda}\right)\right|_{\lambda=\lambda_{\max}} \frac{\kappa^4}{\sqrt{n+m}}\left(2 + \sqrt{2\log\frac{1}{\delta}}\right). \tag{13}$$

*Proof.* For any $x' \in \mathcal{X}$ and any $p \geq 1$, $K^p(x, x')$ as a function of $x$ satisfies

$$\|K^p(x, x')\|_{\mathcal{H}_K}^2 = \left\|\sum_i \lambda_i^p \psi_i(x)\psi_i(x')\right\|_{\mathcal{H}_K}^2 = \sum_i \frac{\lambda_i^{2p}\psi_i(x')^2}{\lambda_i} \leq \lambda_1^{2p-2}\kappa^2.$$

Now, for any $\boldsymbol{u} \in \mathbb{R}^{n+m}$ such that $\|\boldsymbol{u}\|_1 \leq 1$, consider $F_p(x) = \boldsymbol{u}^\top\left(\frac{\boldsymbol{G}_K}{n+m}\right)^p \boldsymbol{v}_K(x)$. Since $\langle K(x_i, \cdot), K(x_j, \cdot)\rangle_{\mathcal{H}_K} = K(x_i, x_j)$, we have $\langle \boldsymbol{v}_K, \boldsymbol{v}_K\rangle_{\mathcal{H}_K} = \boldsymbol{G}_K$, which implies that

$$\|F_p\|_{\mathcal{H}_K}^2 = \left\langle \boldsymbol{u}^\top\left(\frac{\boldsymbol{G}_K}{n+m}\right)^p \boldsymbol{v}_K, \boldsymbol{u}^\top\left(\frac{\boldsymbol{G}_K}{n+m}\right)^p \boldsymbol{v}_K\right\rangle_{\mathcal{H}_K} = \boldsymbol{u}^\top \frac{\boldsymbol{G}_K^{2p+1}}{(n+m)^{2p}}\boldsymbol{u}.$$

We now provide a bound for $\|F_p\|_{\mathcal{H}_K}$, which uses the following exercise from linear algebra:

**Proposition 7.** *For any p.s.d. matrices $\boldsymbol{A}, \boldsymbol{B} \in \mathbb{R}^{d\times d}$, there is $\mathrm{Tr}(\boldsymbol{A}\boldsymbol{B}) \leq \|\boldsymbol{A}\|_2 \mathrm{Tr}(\boldsymbol{B})$.*[*]

Since $\boldsymbol{G}_K$ is *p.s.d.*, we can define $\boldsymbol{G}_K^{1/2}$. Then, using the above exercise, we have

$$\boldsymbol{u}^\top \frac{\boldsymbol{G}_K^{2p+1}}{(n+m)^{2p}}\boldsymbol{u} = \mathrm{Tr}\left(\boldsymbol{u}^\top \boldsymbol{G}_K^{1/2}\left(\frac{\boldsymbol{G}_K}{n+m}\right)^{2p}\boldsymbol{G}_K^{1/2}\boldsymbol{u}\right)$$

$$= \mathrm{Tr}\left(\left(\frac{\boldsymbol{G}_K}{n+m}\right)^{2p}\boldsymbol{G}_K^{1/2}\boldsymbol{u}\boldsymbol{u}^\top\boldsymbol{G}_K^{1/2}\right) \leq \hat{\lambda}_1^{2p}\mathrm{Tr}\left(\boldsymbol{G}_K^{1/2}\boldsymbol{u}\boldsymbol{u}^\top\boldsymbol{G}_K^{1/2}\right).$$

And $\mathrm{Tr}\left(\boldsymbol{G}_K^{1/2}\boldsymbol{u}\boldsymbol{u}^\top\boldsymbol{G}_K^{1/2}\right) = \boldsymbol{u}^\top\boldsymbol{G}_K\boldsymbol{u} = \sum_{i,j=1}^{n+m} u_i u_j K(x_i, x_j) \leq \sum_{i,j=1}^{n+m}|u_i u_j K(x_i, x_j)| \leq \kappa^2\|\boldsymbol{u}\|_1^2 \leq \kappa^2$. Thus, we have $\|F_p\|_{\mathcal{H}_K} \leq \hat{\lambda}_1^p\kappa$ for all $p \geq 0$.

Define $\mathcal{F} := \{f = g_1 g_2 \mid g_1, g_2 \in \mathcal{H}_K, \|g_1\|_{\mathcal{H}_K}, \|g_2\|_{\mathcal{H}_K} \leq 1\}$. Then, as we proved in the proof of Theorem 2, $\|g_1\|_\infty \leq \kappa$ and $\|g_2\|_\infty \leq \kappa$, which means that for all $f \in \mathcal{F}$, $\|f\|_\infty \leq \kappa^2$. Moreover, by Proposition 13 of Zhai et al. (2024), we have $\mathfrak{R}_n(\mathcal{F}) \leq \frac{\kappa^2}{\sqrt{n}}$, where $\mathfrak{R}_n$ is the Rademacher complexity. Thus, by Theorem 4.10 of Wainwright (2019), for any $\delta > 0$,

$$\left|\frac{1}{n+m}\sum_{i=1}^{n+m}f(x_i) - \mathbb{E}_{X\sim P_{\mathcal{X}}}[f(X)]\right| \leq \frac{\kappa^2}{\sqrt{n+m}}\left(2 + \sqrt{2\log\frac{1}{\delta}}\right) \qquad \text{for all } f \in \mathcal{F} \tag{14}$$

holds with probability at least $1 - \delta$. In what follows, we suppose that this inequality holds.

For any $p$, define $\boldsymbol{v}_{K^p}(x) \in \mathbb{R}^{n+m}$ as $\boldsymbol{v}_{K^p}(x)[i] = K^p(x, x_i)$ for all $i \in [n+m]$. Then,

$$\left|K^p(x, x_j) - \hat{K}^p(x, x_j)\right|$$

$$= \left|K^p(x, x_j) - \frac{1}{(n+m)^{p-1}}\boldsymbol{v}_K(x)^\top\boldsymbol{G}_K^{p-2}\boldsymbol{v}_K(x_j)\right|$$

$$\leq \left|K^p(x, x_j) - \frac{1}{n+m}\boldsymbol{v}_{K^{p-1}}(x)^\top\boldsymbol{v}_K(x_j)\right|$$

$$+ \sum_{q=1}^{p-2}\frac{1}{(n+m)^q}\left|\boldsymbol{v}_{K^{p-q}}(x)^\top\boldsymbol{G}_K^{q-1\top}\boldsymbol{v}_K(x_j) - \boldsymbol{v}_{K^{p-q-1}}(x)^\top\frac{\boldsymbol{G}_K^q}{n+m}\boldsymbol{v}_K(x_j)\right|.$$

---

[*]An elementary proof can be found at https://math.stackexchange.com/questions/2241879/reference-for-trace-norm-inequality.

Let us start with bounding the first term:

$$
\left| K^p(x, x_j) - \frac{1}{n+m} \boldsymbol{v}_{K^{p-1}}(x)^\top \boldsymbol{v}_K(x_j) \right|
$$

$$
= \left| \int_{\mathcal{X}} K^{p-1}(x, z) K(x_j, z) dp(z) - \frac{1}{n+m} \sum_{i=1}^{n+m} K^{p-1}(x, x_i) K(x_j, x_i) \right|
$$

$$
\leq \lambda_1^{p-2} \frac{\kappa^4}{\sqrt{n+m}} \left( 2 + \sqrt{2 \log \frac{1}{\delta}} \right),
$$

because $\left\| K^{p-1}(x, \cdot) \right\|_{\mathcal{H}_K} \leq \lambda_1^{p-2} \kappa$, and $\| K(x_j, \cdot) \|_{\mathcal{H}_K} \leq \kappa$.

For the second term, note that $\boldsymbol{v}_K(x_j) = \boldsymbol{G}_K \boldsymbol{e}_j$, where $\boldsymbol{e}_j = [0, \cdots, 0, 1, 0, \cdots, 0]$. So we have:

$$
\frac{1}{(n+m)^q} \left| \boldsymbol{v}_{K^{p-q}}(x)^\top \boldsymbol{G}_K^{q-1} \boldsymbol{v}_K(x_j) - \frac{1}{n+m} \boldsymbol{v}_{K^{p-q-1}}(x)^\top \boldsymbol{G}_K^q \boldsymbol{v}_K(x_j) \right|
$$

$$
= \left| \int_{\mathcal{X}} K^{p-q-1}(x, z) \left[ \boldsymbol{e}_j^\top \left( \frac{\boldsymbol{G}_K}{n+m} \right)^q \boldsymbol{v}_K(z) \right] dp(z) - \frac{1}{n+m} \sum_{j=1}^{n+m} K^{p-q-1}(x, x_j) \left[ \boldsymbol{e}_j^\top \left( \frac{\boldsymbol{G}_K}{n+m} \right)^q \boldsymbol{v}_K(x_j) \right] \right|
$$

$$
\leq \lambda_1^{p-q-2} \hat{\lambda}_1^q \frac{\kappa^4}{\sqrt{n+m}} \left( 2 + \sqrt{2 \log \frac{1}{\delta}} \right),
$$

because $\left\| K^{p-q-1}(x, \cdot) \right\|_{\mathcal{H}_K} \leq \lambda_1^{p-q-2} \kappa$, and $\left\| \boldsymbol{e}_j^\top \left( \frac{\boldsymbol{G}_K}{n+m} \right)^q \boldsymbol{v}_K \right\|_{\mathcal{H}_K} \leq \hat{\lambda}_1^q \kappa$ since $\| \boldsymbol{e}_j \|_1 = 1$.

Combining the above two inequalities yields Eqn. (12). Then, note that

$$
\nabla_\lambda \left( \frac{s(\lambda)}{\lambda} \right) = \sum_{p=1}^\infty \pi_p (p-1) \lambda^{p-2},
$$

which together with $\pi_p \geq 0$ for all $p$ yields Eqn. (13). $\qquad \square$

**Corollary 8.** *If Eqn. (14) holds, then for all $i, j \in [n+m]$, and $\lambda_{\max} = \max \left\{ \lambda_1, \hat{\lambda}_1 \right\}$,*

$$
\left| K_{s^2}(x_i, x_j) - \langle \hat{K}_s(x_i, \cdot), K_s(x_j, \cdot) \rangle_{P_\mathcal{X}} \right| + \left| \langle \hat{K}_s(x_i, \cdot), \hat{K}_s(x_j, \cdot) \rangle_{P_\mathcal{X}} - \langle \hat{K}_s(x_i, \cdot), K_s(x_j, \cdot) \rangle_{P_\mathcal{X}} \right|
$$

$$
\leq 2 s(\lambda_{\max}) \left. \nabla_\lambda \left( \frac{s(\lambda)}{\lambda} \right) \right|_{\lambda = \lambda_{\max}} \frac{\kappa^4}{\sqrt{n+m}} \left( 2 + \sqrt{2 \log \frac{1}{\delta}} \right).
$$

*Proof.* Consider $F_{p,q}(x) = \boldsymbol{u}^\top \left( \frac{\boldsymbol{G}_K}{n+m} \right)^p \boldsymbol{v}_{K^q}(x)$ for any $\| \boldsymbol{u} \|_1 \leq 1$ and any $p \geq 0, q \geq 1$. If Eqn. (14) holds, then by Proposition 7, we have

$$
\| F_{p,q} \|_{\mathcal{H}_K}^2 = \boldsymbol{u}^\top \left( \frac{\boldsymbol{G}_K}{n+m} \right)^p \boldsymbol{G}_{K^{2q-1}} \left( \frac{\boldsymbol{G}_K}{n+m} \right)^p \boldsymbol{u}
$$

$$
= \mathrm{Tr} \left( \left( \frac{\boldsymbol{G}_K}{n+m} \right)^{p-1/2} \frac{\boldsymbol{G}_{K^{2q-1}}}{n+m} \left( \frac{\boldsymbol{G}_K}{n+m} \right)^{p-1/2} \boldsymbol{G}_K^{1/2} \boldsymbol{u} \boldsymbol{u}^\top \boldsymbol{G}_K^{1/2} \right)
$$

$$
\leq \hat{\lambda}_1^{2p-1} \left\| \frac{\boldsymbol{G}_{K^{2q-1}}}{n+m} \right\|_2 \mathrm{Tr} \left( \boldsymbol{G}_K^{1/2} \boldsymbol{u} \boldsymbol{u}^\top \boldsymbol{G}_K^{1/2} \right)
$$

$$
= \hat{\lambda}_1^{2p-1} \left\| \frac{\boldsymbol{G}_{K^{2q-1}}}{n+m} \right\|_2 \boldsymbol{u}^\top \boldsymbol{G}_K \boldsymbol{u} \leq \hat{\lambda}_1^{2p-1} \left\| \frac{\boldsymbol{G}_{K^{2q-1}}}{n+m} \right\|_2 \kappa^2.
$$

For any unit vector $\boldsymbol{w} \in \mathbb{R}^{n+m}$, we have

$$
\hat{\lambda}_1 \geq \boldsymbol{w}^\top \frac{\boldsymbol{G}_K}{n+m} \boldsymbol{w} = \frac{1}{n+m} \sum_{i,j=1}^{n+m} w_i w_j K(x_i, x_j) = \frac{1}{n+m} \sum_t \lambda_t \boldsymbol{w}^\top \boldsymbol{\Psi}_t \boldsymbol{w},
$$

where $\boldsymbol{\Psi}_t \in \mathbb{R}^{(n+m)\times(n+m)}$ such that $\boldsymbol{\Psi}_t[i,j] = \psi_t(x_i)\psi_t(x_j)$, so $\boldsymbol{\Psi}_t$ is *p.s.d.*. Thus, we have

$$\boldsymbol{w}^\top \frac{\boldsymbol{G}_{K^{2q-1}}}{n+m}\boldsymbol{w} = \frac{1}{n+m}\sum_t \lambda_t^{2q-1}\boldsymbol{w}^\top\boldsymbol{\Psi}_t\boldsymbol{w} \le \lambda_1^{2q-2}\frac{1}{n+m}\sum_t \lambda_t\boldsymbol{w}^\top\boldsymbol{\Psi}_t\boldsymbol{w} \le \lambda_1^{2q-2}\hat{\lambda}_1,$$

which implies that $\left\|\frac{\boldsymbol{G}_{K^{2q-1}}}{n+m}\right\|_2 \le \lambda_1^{2q-2}\hat{\lambda}_1$. Thus, $\|F_{p,q}\|_{\mathcal{H}_K}^2 \le \lambda_1^{2q-2}\hat{\lambda}_1^{2p}\kappa^2$.

Note that $\langle \boldsymbol{v}_K, \boldsymbol{v}_K\rangle_{P_{\mathcal{X}}} = \boldsymbol{G}_{K^2}$. So for any $p,q \ge 1$ and any $i,j \in [n+m]$, there is:

$$\left| K^{p+q}(x_i,x_j) - \left\langle \hat{K}^p(x_i,\cdot), K^q(x_j,\cdot)\right\rangle_{P_{\mathcal{X}}}\right| = \left| \boldsymbol{e}_i^\top \boldsymbol{G}_{K^{p+q}}\boldsymbol{e}_j - \boldsymbol{e}_i^\top \frac{\boldsymbol{G}_K^{p-1}}{(n+m)^{p-1}}\boldsymbol{G}_{K^{q+1}}\boldsymbol{e}_j\right|$$

$$\le \sum_{t=1}^{p-1}\left| \boldsymbol{e}_i^\top \frac{\boldsymbol{G}_K^{p-t}}{(n+m)^{p-t}}\boldsymbol{G}_{K^{q+t}}\boldsymbol{e}_j - \boldsymbol{e}_i^\top \frac{\boldsymbol{G}_K^{p-t-1}}{(n+m)^{p-t-1}}\boldsymbol{G}_{K^{q+t+1}}\boldsymbol{e}_j\right|$$

$$= \sum_{t=1}^{p-1}\left| \frac{1}{n+m}\sum_{l=1}^{n+m}\left[\boldsymbol{e}_i^\top \left(\frac{\boldsymbol{G}_K}{n+m}\right)^{p-t-1}\boldsymbol{v}_K\right](x_l)\left[\boldsymbol{e}_j^\top \boldsymbol{v}_{K^{q+t}}\right](x_l) - \left\langle \boldsymbol{e}_i^\top\left(\frac{\boldsymbol{G}_K}{n+m}\right)^{p-t-1}\boldsymbol{v}_K, \boldsymbol{e}_j^\top\boldsymbol{v}_{K^{q+t}}\right\rangle_{P_{\mathcal{X}}}\right|$$

$$\le \sum_{t=1}^{p-1}\lambda_1^{q+t-1}\hat{\lambda}_1^{p-t-1}\frac{\kappa^4}{\sqrt{n+m}}\left(2+\sqrt{2\log\frac{1}{\delta}}\right) \le (p-1)\lambda_{\max}^{p+q-2}\frac{\kappa^4}{\sqrt{n+m}}\left(2+\sqrt{2\log\frac{1}{\delta}}\right).$$

Thus, we have

$$\left| K_{s^2}(x_i,x_j) - \langle \hat{K}_s(x_i,\cdot), K_s(x_j,\cdot)\rangle_{P_{\mathcal{X}}}\right| = \sum_{p,q=1}^\infty \left| \pi_p\pi_q\left(K^{p+q}(x_i,x_j) - \left\langle \hat{K}^p(x_i,\cdot), K^q(x_j,\cdot)\right\rangle_{P_{\mathcal{X}}}\right)\right|$$

$$\le \sum_{p,q=1}^\infty \pi_p\pi_q(p-1)\lambda_{\max}^{p+q-2}\frac{\kappa^4}{\sqrt{n+m}}\left(2+\sqrt{2\log\frac{1}{\delta}}\right).$$

Similarly, we can show that:

$$\left| \left\langle \hat{K}^p(x_i,\cdot), \hat{K}^q(x_j,\cdot)\right\rangle_{P_{\mathcal{X}}} - \left\langle \hat{K}^p(x_i,\cdot), K^q(x_j,\cdot)\right\rangle_{P_{\mathcal{X}}}\right|$$

$$= \left| \boldsymbol{e}_i^\top \frac{\boldsymbol{G}_K^{p-1}}{(n+m)^{p-1}}\boldsymbol{G}_{K^2}\frac{\boldsymbol{G}_K^{q-1}}{(n+m)^{q-1}}\boldsymbol{e}_j - \boldsymbol{e}_i^\top \frac{\boldsymbol{G}_K^{p-1}}{(n+m)^{p-1}}\boldsymbol{G}_{K^{q+1}}\boldsymbol{e}_j\right|$$

$$\le \sum_{t=1}^{q-1}\left| \boldsymbol{e}_i^\top \frac{\boldsymbol{G}_K^{p-1}}{(n+m)^{p-1}}\boldsymbol{G}_{K^{t+1}}\frac{\boldsymbol{G}_K^{q-t}}{(n+m)^{q-t}}\boldsymbol{e}_j - \boldsymbol{e}_i^\top \frac{\boldsymbol{G}_K^{p-1}}{(n+m)^{p-1}}\boldsymbol{G}_{K^{t+2}}\frac{\boldsymbol{G}_K^{q-t-1}}{(n+m)^{q-t-1}}\boldsymbol{e}_j\right|$$

$$= \sum_{t=1}^{q-1}\left| \frac{1}{n+m}\sum_{l=1}^{n+m}\left[\boldsymbol{e}_i^\top\left(\frac{\boldsymbol{G}_K}{n+m}\right)^{p-1}\boldsymbol{v}_{K^{t+1}}\right](x_l)\left[\boldsymbol{e}_j^\top\left(\frac{\boldsymbol{G}_K}{n+m}\right)^{q-t-1}\boldsymbol{v}_K\right](x_l)\right.$$

$$\left. -\left\langle \boldsymbol{e}_i^\top\left(\frac{\boldsymbol{G}_K}{n+m}\right)^{p-1}\boldsymbol{v}_{K^{t+1}}, \boldsymbol{e}_j^\top\left(\frac{\boldsymbol{G}_K}{n+m}\right)^{q-t-1}\boldsymbol{v}_K\right\rangle_{P_{\mathcal{X}}}\right|$$

$$\le \sum_{t=1}^{q-1}\lambda_1^t\hat{\lambda}_1^{p+q-t-2}\frac{\kappa^4}{\sqrt{n+m}}\left(2+\sqrt{2\log\frac{1}{\delta}}\right) \le (q-1)\lambda_{\max}^{p+q-2}\frac{\kappa^4}{\sqrt{n+m}}\left(2+\sqrt{2\log\frac{1}{\delta}}\right),$$

which implies that

$$\left| \langle \hat{K}_s(x_i,\cdot), \hat{K}_s(x_j,\cdot)\rangle_{P_{\mathcal{X}}} - \langle \hat{K}_s(x_i,\cdot), K_s(x_j,\cdot)\rangle_{P_{\mathcal{X}}}\right|$$

$$= \sum_{p,q=1}^\infty \left| \pi_p\pi_q\left(\left\langle \hat{K}^p(x_i,\cdot), \hat{K}^q(x_j,\cdot)\right\rangle_{P_{\mathcal{X}}} - \left\langle \hat{K}^p(x_i,\cdot), K^q(x_j,\cdot)\right\rangle_{P_{\mathcal{X}}}\right)\right|$$

$$\le \sum_{p,q=1}^\infty \pi_p\pi_q(q-1)\lambda_{\max}^{p+q-2}\frac{\kappa^4}{\sqrt{n+m}}\left(2+\sqrt{2\log\frac{1}{\delta}}\right).$$

Combining the above inequalities, we obtain

$$\left| K_{s^2}(x_i, x_j) - \langle \hat{K}_s(x_i, \cdot), K_s(x_j, \cdot)\rangle_{P_{\mathcal{X}}} \right| + \left| \langle \hat{K}_s(x_i, \cdot), \hat{K}_s(x_j, \cdot)\rangle_{P_{\mathcal{X}}} - \langle \hat{K}_s(x_i, \cdot), K_s(x_j, \cdot)\rangle_{P_{\mathcal{X}}} \right|$$

$$\leq \sum_{p,q=1}^{\infty} \pi_p \pi_q (p + q - 2)\lambda_{\max}^{p+q-2} \frac{\kappa^4}{\sqrt{n+m}} \left( 2 + \sqrt{2\log\frac{1}{\delta}} \right)$$

$$= \lambda_{\max} \nabla_\lambda \left( \frac{s(\lambda)^2}{\lambda^2} \right)\Big|_{\lambda=\lambda_{\max}} \frac{\kappa^4}{\sqrt{n+m}} \left( 2 + \sqrt{2\log\frac{1}{\delta}} \right),$$

so we get the result by expanding the derivative. □

## B.6 PROOF OF THEOREM 3

Define $\boldsymbol{v}_{K_s,n}(x) \in \mathbb{R}^n$ such that $\boldsymbol{v}_{K_s,n}(x)[i] = K_s(x, x_i)$. Define $\boldsymbol{v}_{\hat{K}_s,n}(x)$ similarly. Recall the formulas $\tilde{f} = \tilde{\boldsymbol{\alpha}}^\top \boldsymbol{v}_{K_s,n}$ and $\hat{f} = \hat{\boldsymbol{\alpha}}^\top \boldsymbol{v}_{\hat{K}_s,n}$. Define $f^\dagger := \hat{\boldsymbol{\alpha}}^\top \boldsymbol{v}_{K_s,n}$. Since $\boldsymbol{G}_{\hat{K}_s,n}$ is p.s.d., we can see that $\|\hat{\boldsymbol{\alpha}}\|_2 \leq \frac{\|\boldsymbol{y}\|_2}{n\beta_n}$, and $\|\hat{\boldsymbol{\alpha}}\|_1 \leq \sqrt{n}\|\hat{\boldsymbol{\alpha}}\|_2$. So if Eqn. (13) in Lemma 6 holds, then by Corollary 8, we have:

$$\left\| \hat{f} - f^\dagger \right\|_{P_{\mathcal{X}}}^2 = \hat{\boldsymbol{\alpha}}^\top \left\langle \boldsymbol{v}_{\hat{K}_s,n} - \boldsymbol{v}_{K_s,n}, \boldsymbol{v}_{\hat{K}_s,n} - \boldsymbol{v}_{K_s,n} \right\rangle_{P_{\mathcal{X}}} \hat{\boldsymbol{\alpha}}$$

$$= \hat{\boldsymbol{\alpha}}^\top \left( \langle \hat{K}_s(x_i, \cdot), \hat{K}_s(x_j, \cdot)\rangle_{P_{\mathcal{X}}} + K_{s^2}(x_i, x_j) - 2\langle \hat{K}_s(x_i, \cdot), K_s(x_j, \cdot)\rangle_{P_{\mathcal{X}}} \right)\hat{\boldsymbol{\alpha}}$$

$$\leq 2s(\lambda_{\max}) \nabla_\lambda \left( \frac{s(\lambda)}{\lambda} \right)\Big|_{\lambda=\lambda_{\max}} \frac{\beta_n^{-2}\kappa^4}{\sqrt{n+m}} \left( 2 + \sqrt{2\log\frac{1}{\delta}} \right) \frac{\|\boldsymbol{y}\|_2^2}{n}.$$

By the definitions of $\tilde{\boldsymbol{\alpha}}$ and $\hat{\boldsymbol{\alpha}}$, we can also see that:

$$(\boldsymbol{G}_{K_s,n} + n\beta_n \boldsymbol{I}_n)(\hat{\boldsymbol{\alpha}} - \tilde{\boldsymbol{\alpha}}) = \left( \boldsymbol{G}_{K_s,n} - \boldsymbol{G}_{\hat{K}_s,n} \right)\hat{\boldsymbol{\alpha}}. \tag{15}$$

Note that $\left\| \boldsymbol{G}_{K_s,n} - \boldsymbol{G}_{\hat{K}_s,n} \right\|_2 \leq n\left\| \boldsymbol{G}_{K_s,n} - \boldsymbol{G}_{\hat{K}_s,n} \right\|_{\max}$. Thus, by Eqn. (15), we have:

$$\left\| \tilde{f} - f^\dagger \right\|_{\mathcal{H}_{K_s}}^2 = (\hat{\boldsymbol{\alpha}} - \tilde{\boldsymbol{\alpha}})^\top \boldsymbol{G}_{K_s,n}(\hat{\boldsymbol{\alpha}} - \tilde{\boldsymbol{\alpha}})$$

$$= (\hat{\boldsymbol{\alpha}} - \tilde{\boldsymbol{\alpha}})^\top \left( \boldsymbol{G}_{K_s,n} - \boldsymbol{G}_{\hat{K}_s,n} \right)\hat{\boldsymbol{\alpha}} - n\beta_n(\hat{\boldsymbol{\alpha}} - \tilde{\boldsymbol{\alpha}})^\top (\hat{\boldsymbol{\alpha}} - \tilde{\boldsymbol{\alpha}})$$

$$\leq \|\hat{\boldsymbol{\alpha}}\|_2 \left\| \boldsymbol{G}_{K_s,n} - \boldsymbol{G}_{\hat{K}_s,n} \right\|_2 \|\hat{\boldsymbol{\alpha}}\|_2 + \|\tilde{\boldsymbol{\alpha}}\|_2 \left\| \boldsymbol{G}_{K_s,n} - \boldsymbol{G}_{\hat{K}_s,n} \right\|_2 \|\hat{\boldsymbol{\alpha}}\|_2 - 0$$

$$\leq 2 \nabla_\lambda \left( \frac{s(\lambda)}{\lambda} \right)\Big|_{\lambda=\lambda_{\max}} \frac{\beta_n^{-2}\kappa^4}{\sqrt{n+m}} \left( 2 + \sqrt{2\log\frac{1}{\delta}} \right) \frac{\|\boldsymbol{y}\|_2^2}{n}.$$

And note that we have $\left\| \tilde{f} - f^\dagger \right\|_{P_{\mathcal{X}}}^2 \leq s(\lambda_1)\left\| \tilde{f} - f^\dagger \right\|_{\mathcal{H}_{K_s}}^2 \leq s(\lambda_{\max})\left\| \tilde{f} - f^\dagger \right\|_{\mathcal{H}_{K_s}}^2$. Thus,

$$\left\| \hat{f} - \tilde{f} \right\|_{P_{\mathcal{X}}}^2 \leq 2\left( \left\| \hat{f} - f^\dagger \right\|_{P_{\mathcal{X}}}^2 + \left\| \tilde{f} - f^\dagger \right\|_{P_{\mathcal{X}}}^2 \right)$$

$$\leq 8s(\lambda_{\max}) \nabla_\lambda \left( \frac{s(\lambda)}{\lambda} \right)\Big|_{\lambda=\lambda_{\max}} \frac{\beta_n^{-2}\kappa^4}{\sqrt{n+m}} \left( 2 + \sqrt{2\log\frac{1}{\delta}} \right) \frac{\|\boldsymbol{y}\|_2^2}{n},$$

as desired. □

## B.7 PROOF OF PROPOSITION 3

By Assumption 3, there is $\|f^*\|_{\mathcal{H}_{K_{s^*}}}^2 \leq \epsilon\|f^*\|_{P_{\mathcal{X}}}^2$. Let $f^* = \sum_i u_i \psi_i$, then this is equivalent to

$$\sum_i \frac{u_i^2}{s^*(\lambda_i)} \leq \epsilon \sum_i u_i^2.$$

Let $s(\lambda) = \frac{\lambda}{1-\eta\lambda}$ be the inverse Laplacian, then we have

$$\sum_i \frac{u_i^2}{s(\lambda_i)} \leq \sum_i \frac{u_i^2}{\lambda_i} \leq \sum_i \frac{u_i^2}{s^*(\lambda_i)/M} \leq M\epsilon \sum_i u_i^2,$$

which means that $f^*$ also satisfies Assumption 3 w.r.t. $s$ by replacing $\epsilon$ with $M\epsilon$. All other conditions are the same, so we can continue to apply Theorems 2 and 3. $\square$

### B.8   PROOF OF PROPOSITION 4

This proof is similar to the proof of Proposition 4 in Zhai et al. (2024).

Since $\hat{\Psi}$ is at most rank-$d$, there must be a function in $\text{span}\{\psi_1, \cdots, \psi_{d+1}\}$ that is orthogonal to $\hat{\Psi}$. Thus, we can find two functions $f_1, f_2 \in \text{span}\{\psi_1, \cdots, \psi_{d+1}\}$ such that: $\|f_1\|_{P_\mathcal{X}} = \|f_2\|_{P_\mathcal{X}} = 1$, $f_1$ is orthogonal to $\hat{\Psi}$, $f_2 = \boldsymbol{u}^\top \hat{\Psi}$ (which means that $f_1 \perp f_2$), and $\psi_1 \in \text{span}\{f_1, f_2\}$. Let $\psi_1 = \alpha_1 f_1 + \alpha_2 f_2$, and without loss of generality suppose that $\alpha_1, \alpha_2 \in [0,1]$. Then, $\alpha_1^2 + \alpha_2^2 = 1$. Let $f_0 = \alpha_2 f_1 - \alpha_1 f_2$, then $\|f_0\|_{P_\mathcal{X}} = 1$ and $\langle f_0, \psi_1 \rangle_{P_\mathcal{X}} = 0$. This also implies that $\langle f_0, \psi_1 \rangle_{\mathcal{H}_{K_s}} = 0$.

Let $\beta_1, \beta_2 \in [0,1]$ be any value such that $\beta_1^2 + \beta_2^2 = 1$. Let $f = B(\beta_1 \psi_1 + \beta_2 f_0)$, then $\|f\|_{P_\mathcal{X}} = B$. And we have $\|f\|_{\mathcal{H}_{K_s}}^2 = B^2 \left( \|\beta_1 \psi_1\|_{\mathcal{H}_{K_s}}^2 + \|\beta_2 f_0\|_{\mathcal{H}_{K_s}}^2 \right) \leq B^2 \left( \frac{\beta_1^2}{s(\lambda_1)} + \frac{\beta_2^2}{s(\lambda_{d+1})} \right) \leq \epsilon B^2$, as long as $\beta_2^2 \leq \frac{s(\lambda_{d+1})}{s(\lambda_1)-s(\lambda_{d+1})}[s(\lambda_1)\epsilon - 1]$. This means that $f \in \mathcal{F}_s$.

Let $F(\alpha_1) := \alpha_1 \beta_1 + \alpha_2 \beta_2 = \alpha_1 \beta_1 + \sqrt{1-\alpha_1^2}\beta_2$ for $\alpha_1 \in [0,1]$. It is easy to show that $F(\alpha_1)$ first increases and then decreases on $[0,1]$, so $F(\alpha_1)^2 \geq \min\left\{F(0)^2, F(1)^2\right\} = \min\left\{\beta_1^2, \beta_2^2\right\}$, which can be $\frac{s(\lambda_{d+1})}{s(\lambda_1)-s(\lambda_{d+1})}[s(\lambda_1)\epsilon - 1]$ given that it is at most $\frac{1}{2}$. Thus, for this $f$, we have

$$\min_{\boldsymbol{w} \in \mathbb{R}^d} \left\| \boldsymbol{w}^\top \hat{\Psi} - f \right\|_{P_\mathcal{X}}^2 = \|B(\alpha_1 \beta_1 + \alpha_2 \beta_2)f_1\|_{P_\mathcal{X}}^2 = B^2 F(\alpha_1)^2 \geq \frac{s(\lambda_{d+1})}{s(\lambda_1)-s(\lambda_{d+1})}[s(\lambda_1)\epsilon - 1]B^2.$$

If the equality is attained, then we must have $\|f_0\|_{\mathcal{H}_{K_s}}^2 = s(\lambda_{d+1})^{-1}$. So if $s(\lambda_d) > s(\lambda_{d+1})$, then $\hat{\Psi}$ must span the linear span of $\psi_1, \cdots, \psi_d$.

Finally, we prove that $\max_{f \in \mathcal{F}_s} \min_{\boldsymbol{w} \in \mathbb{R}^d} \left\| \boldsymbol{w}^\top \hat{\Psi} - f \right\|_{P_\mathcal{X}}^2 \leq \frac{s(\lambda_{d+1})}{s(\lambda_1)-s(\lambda_{d+1})}[s(\lambda_1)\epsilon - 1]B^2$ if $\hat{\Psi}$ spans $\text{span}\{\psi_1, \cdots, \psi_d\}$. For any $f = \sum_i u_i \psi_i \in \mathcal{F}_s$, we have $\sum_i u_i^2 \leq B^2$, and $\sum_i \frac{u_i^2}{s(\lambda_i)} \leq \epsilon \sum_i u_i^2$. Let $a = \sum_{i \geq d+1} u_i^2$, and $b = \sum_{i=1}^d u_i^2$. Then, $a + b \leq B^2$. So we have

$$0 \geq \sum_i \frac{u_i^2}{s(\lambda_i)} - \epsilon \sum_i u_i^2 \geq \left[\frac{1}{s(\lambda_1)} - \epsilon\right]b + \left[\frac{1}{s(\lambda_{d+1})} - \epsilon\right]a \qquad \text{(since $s$ is monotonic)}$$

$$\geq \left[\frac{1}{s(\lambda_1)} - \epsilon\right](B^2 - a) + \left[\frac{1}{s(\lambda_{d+1})} - \epsilon\right]a = \left[\frac{1}{s(\lambda_1)} - \epsilon\right]B^2 + \left[\frac{1}{s(\lambda_{d+1})} - \frac{1}{s(\lambda_1)}\right]a,$$

which implies that $\min_{\boldsymbol{w} \in \mathbb{R}^d} \left\| \boldsymbol{w}^\top \hat{\Psi} - f \right\|_{P_\mathcal{X}}^2 = a \leq \frac{s(\lambda_{d+1})}{s(\lambda_1)-s(\lambda_{d+1})}[s(\lambda_1)\epsilon - 1]B^2.$ $\square$

### B.9   PROOF OF THEOREM 4

For any $f = \sum_i u_i \psi_i \in \mathcal{H}_{K_s}$ satisfying Assumption 3, $\|f\|_{\mathcal{H}_K}^2 = \sum_i \frac{u_i^2}{\lambda_i} \leq \sum_i \frac{u_i^2}{s(\lambda_i)/M} \leq \epsilon M B^2$.

Define $\tilde{f}_d$ as the projection of $f^*$ onto $\hat{\Psi}$ w.r.t. $\mathcal{H}_K$. Define RKHS $\mathcal{H}_{\hat{\Psi}} := \text{span}\left\{\hat{\psi}_1, \cdots, \hat{\psi}_d\right\}$ as a subspace of $\mathcal{H}_K$, then $\tilde{f}_d \in \mathcal{H}_{\hat{\Psi}}$. Let $K_{\hat{\Psi}}$ be the reproducing kernel of $\mathcal{H}_{\hat{\Psi}}$. Let $\tilde{\boldsymbol{y}} := [\tilde{y}_1, \cdots, \tilde{y}_n]$, where $\tilde{y}_i := \tilde{f}_d(x_i) + y_i - f^*(x_i)$. Then, the KRR of $\tilde{f}_d$ with $K_{\hat{\Psi}}$ is given by

$$\hat{f}_d^\dagger = \tilde{\boldsymbol{w}}^{*\top}\hat{\Psi}, \quad \tilde{\boldsymbol{w}}^* = \left(\hat{\Psi}(\boldsymbol{X}_n)\hat{\Psi}(\boldsymbol{X}_n)^\top + n\beta_n \boldsymbol{I}_d\right)^{-1}\hat{\Psi}(\boldsymbol{X}_n)\tilde{\boldsymbol{y}}.$$

First, we show a lower bound for the eigenvalues of $\hat{\Psi}(\boldsymbol{X}_n)\hat{\Psi}(\boldsymbol{X}_n)^\top$. Similar to Eqn. (14), define $\mathcal{F} := \{f = g_1 g_2 \mid g_1, g_2 \in \mathcal{H}_K, \|g_1\|_{\mathcal{H}_K}, \|g_2\|_{\mathcal{H}_K} \leq 1\}$, then we have:

$$\begin{cases} \left| \dfrac{1}{n} \displaystyle\sum_{i=1}^n f(x_i) - \mathbb{E}_{X \sim P_{\mathcal{X}}}[f(X)] \right| \leq \dfrac{\kappa^2}{\sqrt{n}}\left(2 + \sqrt{2\log\dfrac{2}{\delta}}\right) & \text{for all } f \in \mathcal{F}; \qquad (16) \\[4mm] \left| \dfrac{1}{m} \displaystyle\sum_{i=n+1}^{n+m} f(x_i) - \mathbb{E}_{X \sim P_{\mathcal{X}}}[f(X)] \right| \leq \dfrac{\kappa^2}{\sqrt{m}}\left(2 + \sqrt{2\log\dfrac{2}{\delta}}\right) & \text{for all } f \in \mathcal{F} \qquad (17) \end{cases}$$

hold simultaneously with probability at least $1 - \delta$ for any $\delta > 0$. In what follows, we assume them to hold. Then for all $f \in \mathcal{F}$, we have $\left| \frac{1}{n}\sum_{i=1}^n f(x_i) - \frac{1}{m}\sum_{i=n+1}^{n+m} f(x_i) \right| \leq \frac{\kappa^2}{\sqrt{n}}\left(4 + 2\sqrt{2\log\frac{2}{\delta}}\right)$.

For any unit vector $\boldsymbol{u} \in \mathbb{R}^d$, let $f = \boldsymbol{u}^\top \hat{\Psi}$. Then, $\|f\|_{\mathcal{H}_K} \in 1$, so $f^2 \in \mathcal{F}$. And we have

$$\sum_{i=n+1}^{n+m} f(x_i)^2 = \|\boldsymbol{G}_{K,m}[\boldsymbol{v}_1, \cdots, \boldsymbol{v}_d]\boldsymbol{u}\|_2^2 = \left\|[m\tilde{\lambda}_1\boldsymbol{v}_1, \cdots, m\tilde{\lambda}_d\boldsymbol{v}_d]\boldsymbol{u}\right\|_2^2 = \sum_{j=1}^d m\tilde{\lambda}_j u_j^2 \geq m\tilde{\lambda}_d,$$

which implies that

$$\frac{1}{n}\sum_{i=1}^n f(x_i)^2 \geq \tilde{\lambda}_d - \frac{\kappa^2}{\sqrt{n}}\left(4 + 2\sqrt{2\log\frac{2}{\delta}}\right) \qquad \text{for all } f = \boldsymbol{u}^\top\hat{\Psi} \text{ where } \boldsymbol{u} \in \mathbb{R}^d \text{ is a unit vector.}$$

Thus, for any unit vector $\boldsymbol{u} \in \mathbb{R}^d$, $\|\boldsymbol{u}^\top\hat{\Psi}(\boldsymbol{X}_n)\|_2^2 \geq n\tilde{\lambda}_d - \kappa^2\sqrt{n}\left(4 + 2\sqrt{2\log\frac{2}{\delta}}\right)$.

Second, we bound $\|\hat{f}_d - \hat{f}_d^\dagger\|_{\mathcal{H}_K}^2$. Denote $\Delta\boldsymbol{y} := [f^*(x_1) - \tilde{f}_d(x_1), \cdots, f^*(x_n) - \tilde{f}_d(x_n)]^\top \in \mathbb{R}^n$. Note that $\langle \hat{\Psi}, \hat{\Psi} \rangle_{\mathcal{H}_K} = \boldsymbol{I}_d$, so we have

$$\begin{aligned} \left\|\hat{f}_d - \hat{f}_d^\dagger\right\|_{\mathcal{H}_K}^2 &= \left\|\left[\left(\hat{\Psi}(\boldsymbol{X}_n)\hat{\Psi}(\boldsymbol{X}_n)^\top + n\beta_n\boldsymbol{I}_d\right)^{-1}\hat{\Psi}(\boldsymbol{X}_n)(\boldsymbol{y} - \tilde{\boldsymbol{y}})\right]^\top \hat{\Psi}\right\|_{\mathcal{H}_K}^2 \\ &= \left\|\left(\hat{\Psi}(\boldsymbol{X}_n)\hat{\Psi}(\boldsymbol{X}_n)^\top + n\beta_n\boldsymbol{I}_d\right)^{-1}\hat{\Psi}(\boldsymbol{X}_n)\Delta\boldsymbol{y}\right\|_2^2. \end{aligned}$$

So it suffices to bound $\left\|\boldsymbol{Q}^{-1}\hat{\Psi}(\boldsymbol{X}_n)\right\|_2^2$ where $\boldsymbol{Q} = \hat{\Psi}(\boldsymbol{X}_n)\hat{\Psi}(\boldsymbol{X}_n)^\top + n\beta_n\boldsymbol{I}_d$, which is equal to the largest eigenvalue of $\hat{\Psi}(\boldsymbol{X}_n)^\top\boldsymbol{Q}^{-2}\hat{\Psi}(\boldsymbol{X}_n)$, which is further equal to the largest eigenvalue of $\boldsymbol{Q}^{-2}\hat{\Psi}(\boldsymbol{X}_n)\hat{\Psi}(\boldsymbol{X}_n)^\top$ by Sylvester's theorem. Let the eigenvalues of $\hat{\Psi}(\boldsymbol{X}_n)\hat{\Psi}(\boldsymbol{X}_n)^\top$ be $\mu_1 \geq \cdots \geq \mu_d \geq 0$, with corresponding eigenvectors $\boldsymbol{\alpha}_1, \cdots, \boldsymbol{\alpha}_d$ that form an orthonormal basis of $\mathbb{R}^d$. For all $i \in [d]$, if $\mu_i = 0$, then $\boldsymbol{Q}^{-2}\hat{\Psi}(\boldsymbol{X}_n)\hat{\Psi}(\boldsymbol{X}_n)^\top\boldsymbol{\alpha}_i = \boldsymbol{0}$, meaning that $\boldsymbol{\alpha}_i$ is also an eigenvector of $\boldsymbol{Q}^{-2}\hat{\Psi}(\boldsymbol{X}_n)\hat{\Psi}(\boldsymbol{X}_n)^\top$ with eigenvalue 0. And if $\mu_i > 0$, then we have

$$\boldsymbol{Q}\boldsymbol{\alpha}_i = \hat{\Psi}(\boldsymbol{X}_n)\hat{\Psi}(\boldsymbol{X}_n)^\top\boldsymbol{\alpha}_i + n\beta_n\boldsymbol{\alpha}_i = (\mu_i + n\beta_n)\boldsymbol{\alpha}_i,$$

which implies that $\boldsymbol{Q}^2\boldsymbol{\alpha}_i = \boldsymbol{Q}(\mu_i + n\beta_n)\boldsymbol{\alpha}_i = (\mu_i + n\beta_n)^2\boldsymbol{\alpha}_i = \frac{(\mu_i+n\beta_n)^2}{\mu_i}\hat{\Psi}(\boldsymbol{X}_n)\hat{\Psi}(\boldsymbol{X}_n)^\top\boldsymbol{\alpha}_i$. Thus, $\boldsymbol{\alpha}_i$ is an eigenvector of $\boldsymbol{Q}^{-2}\hat{\Psi}(\boldsymbol{X}_n)\hat{\Psi}(\boldsymbol{X}_n)^\top$ with eigenvalue $\frac{\mu_i}{(\mu_i+n\beta_n)^2}$. This means that $\boldsymbol{\alpha}_1, \cdots, \boldsymbol{\alpha}_d$ are all eigenvectors of $\boldsymbol{Q}^{-2}\hat{\Psi}(\boldsymbol{X}_n)\hat{\Psi}(\boldsymbol{X}_n)^\top$. On the other hand, we have $\mu_d \geq n\tilde{\lambda}_d - \kappa^2\sqrt{n}\left(4 + 2\sqrt{2\log\frac{2}{\delta}}\right)$, and suppose that $n$ is large enough so that $\mu_d \geq \frac{n\tilde{\lambda}_d}{2}$, i.e. $n \geq \frac{4\kappa^4}{\tilde{\lambda}_d^2}\left(4 + 2\sqrt{2\log\frac{2}{\delta}}\right)^2$. Then, we have

$$\left\|\boldsymbol{Q}^{-1}\hat{\Psi}(\boldsymbol{X}_n)\right\|_2^2 \leq \max_{i\in[d]}\frac{\mu_i}{(\mu_i+n\beta_n)^2} \leq \max_{i\in[d]}\frac{1}{\mu_i} \leq \frac{2}{n\tilde{\lambda}_d}.$$

Thus, $\left\|\hat{f}_d - \hat{f}_d^\dagger\right\|_{\mathcal{H}_K}^2 \leq \frac{2}{\tilde{\lambda}_d}\frac{\|\Delta\boldsymbol{y}\|_2^2}{n}$.

Third, we bound $\|\hat{f}_d - \hat{f}_d^{\dagger}\|_{P_{\mathcal{X}}}^2$. Denote $\Delta \boldsymbol{f} := [\hat{f}_d(x_1) - \hat{f}_d^{\dagger}(x_1), \cdots, \hat{f}_d(x_n) - \hat{f}_d^{\dagger}(x_n)]^{\top} \in \mathbb{R}^n$. Then, we have

$$\Delta \boldsymbol{f} = \hat{\Psi}(\boldsymbol{X}_n)^{\top}(\hat{\boldsymbol{w}}^* - \tilde{\boldsymbol{w}}^*) = \hat{\Psi}(\boldsymbol{X}_n)^{\top}\left(\hat{\Psi}(\boldsymbol{X}_n)\hat{\Psi}(\boldsymbol{X}_n)^{\top} + n\beta_n \boldsymbol{I}_d\right)^{-1}\hat{\Psi}(\boldsymbol{X}_n)\Delta \boldsymbol{y}.$$

Similarly, we can show that the eigenvalues of $\hat{\Psi}(\boldsymbol{X}_n)^{\top}\left(\hat{\Psi}(\boldsymbol{X}_n)\hat{\Psi}(\boldsymbol{X}_n)^{\top} + n\beta_n \boldsymbol{I}_d\right)^{-1}\hat{\Psi}(\boldsymbol{X}_n)$ are $\frac{\mu_i}{\mu_i + n\beta_n}$, which are no larger than 1. Thus, $\|\Delta \boldsymbol{f}\|_2 \leq \|\Delta \boldsymbol{y}\|_2$. And by Eqn. (16), we have

$$\frac{\|\Delta \boldsymbol{y}\|_2^2}{n} \leq \|f^* - \tilde{f}_d\|_{P_{\mathcal{X}}}^2 + \|f^* - \tilde{f}_d\|_{\mathcal{H}_K}^2 \frac{\kappa^2}{\sqrt{n}}\left(2 + \sqrt{2\log\frac{2}{\delta}}\right).$$

So by Eqn. (16), we have

$$\begin{aligned}
\|\hat{f}_d - \hat{f}_d^{\dagger}\|_{P_{\mathcal{X}}}^2 &\leq \frac{\|\Delta \boldsymbol{f}\|_2^2}{n} + \|\hat{f}_d - \hat{f}_d^{\dagger}\|_{\mathcal{H}_K}^2 \frac{\kappa^2}{\sqrt{n}}\left(2 + \sqrt{2\log\frac{2}{\delta}}\right) \\
&\leq \frac{\|\Delta \boldsymbol{y}\|_2^2}{n}\left[1 + \frac{2\kappa^2}{\tilde{\lambda}_d \sqrt{n}}\left(2 + \sqrt{2\log\frac{2}{\delta}}\right)\right] \\
&\leq \left[\|f^* - \tilde{f}_d\|_{P_{\mathcal{X}}}^2 + \|f^* - \tilde{f}_d\|_{\mathcal{H}_K}^2 \frac{\kappa^2}{\sqrt{n}}\left(2 + \sqrt{2\log\frac{2}{\delta}}\right)\right]\left[1 + \frac{2\kappa^2}{\tilde{\lambda}_d \sqrt{n}}\left(2 + \sqrt{2\log\frac{2}{\delta}}\right)\right] \\
&\leq \frac{3}{2}\left(\|f^* - \tilde{f}_d\|_{P_{\mathcal{X}}}^2 + \frac{\tilde{\lambda}_d}{4}\|f^* - \tilde{f}_d\|_{\mathcal{H}_K}^2\right),
\end{aligned}$$

where the final step uses $n \geq \frac{4\kappa^4}{\tilde{\lambda}_d^2}\left(4 + 2\sqrt{2\log\frac{2}{\delta}}\right)^2$.

Finally, we bound $\left\|\hat{f}_d^{\dagger} - \tilde{f}_d\right\|_{P_{\mathcal{X}}}$ using Theorem 6 with $K = K_{\hat{\Psi}}, \alpha = \beta = 1$, and $\gamma = 0$. Recall the four conditions:

- (EVD) is satisfied for any $p \in (0, 1]$ since $K_{\hat{\Psi}}$ has at most $d$ non-zero eigenvalues.
- (EMB) is satisfied with $c_2 = \kappa$ since $\|f\|_{\mathcal{H}_{\hat{\Psi}}} = \|f\|_{\mathcal{H}_K}$ for all $f \in \mathcal{H}_{\hat{\Psi}}$.
- (SRC) is satisfied with $c_3 = \sqrt{\epsilon M}B$ since $\|\tilde{f}_d\|_{\mathcal{H}_{\hat{\Psi}}} \leq \|f^*\|_{\mathcal{H}_K} \leq \sqrt{\epsilon M}B$.
- (MOM) is satisfied by condition.

And we have $B_{\infty} = \kappa\sqrt{\epsilon M}B$. Thus, when $\tau \geq \kappa^{-1}$, it holds with probability at least $1 - 4e^{-\tau}$ that

$$\left\|\hat{f}_d^{\dagger} - \tilde{f}_d\right\|_{P_{\mathcal{X}}}^2 \leq \frac{c_0}{2}\tau^2\left[(\kappa^2 M\epsilon B^2 + \kappa^2\sigma^2)n^{-\frac{1}{1+p}} + \kappa^2\max\{L^2, \kappa^2 M\epsilon B^2\}n^{-\frac{1+2p}{1+p}}\right].$$

Combining the two inequalities with $(a + b)^2 \leq 2(a^2 + b^2)$ yields the result. $\qquad\square$

### B.10 PROOF OF THEOREM 5

We start with the following lemma:

**Lemma 9.** *For any $g \in \mathcal{H}_K$ such that $\|g\|_{\mathcal{H}_K} = 1$ and $\langle g, \hat{\psi}_i \rangle_{\mathcal{H}_K} = 0$ for all $i \in [d]$, there is*

$$\|\hat{\psi}_1\|_{P_{\mathcal{X}}}^2 + \cdots + \|\hat{\psi}_d\|_{P_{\mathcal{X}}}^2 + \|g\|_{P_{\mathcal{X}}}^2 \leq \lambda_1 + \cdots + \lambda_{d+1}. \tag{18}$$

*Proof.* Let $\left[\hat{\psi}_1, \cdots, \hat{\psi}_d, g\right] = \boldsymbol{Q}\boldsymbol{D}_{\lambda}^{1/2}\boldsymbol{\Psi}^*$, where $\boldsymbol{D}_{\lambda} = \text{diag}(\lambda_1, \lambda_2, \cdots)$, and $\boldsymbol{\Psi}^* = [\psi_1, \psi_2, \cdots]$. Then, $\boldsymbol{Q}\boldsymbol{Q}^{\top} = \left\langle\left[\hat{\psi}_1, \cdots, \hat{\psi}_d, g\right], \left[\hat{\psi}_1, \cdots, \hat{\psi}_d, g\right]\right\rangle_{\mathcal{H}_K} = \boldsymbol{I}_{d+1}$, and

$$\|\hat{\psi}_1\|_{P_{\mathcal{X}}}^2 + \cdots + \|\hat{\psi}_d\|_{P_{\mathcal{X}}}^2 + \|g\|_{P_{\mathcal{X}}}^2 = \text{Tr}\left(\boldsymbol{Q}\boldsymbol{D}_{\lambda}\boldsymbol{Q}^{\top}\right).$$

So we obtain the result by applying Lemma 9 in Zhai et al. (2024). $\qquad\square$

Define $\mathcal{F}_d := \left\{ f = \sum_{i=1}^d g_i^2 \mid g_i \in \mathcal{H}_K, \langle g_i, g_j \rangle_{\mathcal{H}_K} = \delta_{i,j} \right\}$. We now bound its Rademacher complexity. For any $x$, denote $\Psi(x) = [\lambda_1^{1/2} \psi_1(x), \lambda_2^{1/2} \psi_2(x), \cdots]$. For any $S = \{x_1, \cdots, x_m\}$, denote $\Psi_k = \Psi(x_k)$ for $k \in [m]$. Let $g_i(x) = \boldsymbol{u}_i^\top \Psi(x)$, and denote $\boldsymbol{U} = [\boldsymbol{u}_1, \cdots, \boldsymbol{u}_d]$. Then, $\boldsymbol{U}^\top \boldsymbol{U} = \boldsymbol{I}_d$. Let $\boldsymbol{\sigma} = [\sigma_1, \cdots, \sigma_m]$ be Rademacher variates. Thus, the empirical Rademacher complexity satisfies:

$$\hat{\mathfrak{R}}_S(\mathcal{F}_d) \leq \mathbb{E}_{\boldsymbol{\sigma}} \left[ \sup_{\boldsymbol{u}_1, \cdots, \boldsymbol{u}_d} \left| \frac{1}{m} \sum_{k=1}^m \sum_{i=1}^d \sigma_k \boldsymbol{u}_i^\top \Psi_k \Psi_k^\top \boldsymbol{u}_i \right| \right]$$

$$= \mathbb{E}_{\boldsymbol{\sigma}} \left[ \sup_{\boldsymbol{U}: \boldsymbol{U}^\top \boldsymbol{U} = \boldsymbol{I}_d} \left| \mathrm{Tr} \left( \boldsymbol{U}^\top \left( \frac{1}{m} \sum_{k=1}^m \sigma_k \Psi_k \Psi_k^\top \right) \boldsymbol{U} \right) \right| \right]$$

$$= \mathbb{E}_{\boldsymbol{\sigma}} \left[ \sup_{\boldsymbol{U}: \boldsymbol{U}^\top \boldsymbol{U} = \boldsymbol{I}_d} \left| \mathrm{Tr} \left( \left( \frac{1}{m} \sum_{k=1}^m \sigma_k \Psi_k \Psi_k^\top \right) \boldsymbol{U} \boldsymbol{U}^\top \right) \right| \right].$$

Let $\mu_1 \geq \mu_2 \geq \cdots$ be the singular values of $\frac{1}{m} \sum_{k=1}^m \sigma_k \Psi_k \Psi_k^\top$. For any $\boldsymbol{U}$, the singular values of $\boldsymbol{U} \boldsymbol{U}^\top$ are $d$ ones and then zeros. So by von Neumann's trace inequality, we have:

$$\sup_{\boldsymbol{U}: \boldsymbol{U}^\top \boldsymbol{U} = \boldsymbol{I}_d} \left| \mathrm{Tr} \left( \left( \frac{1}{m} \sum_{k=1}^m \sigma_k \Psi_k \Psi_k^\top \right) \boldsymbol{U} \boldsymbol{U}^\top \right) \right| \leq \mu_1 + \cdots + \mu_d \leq \frac{\sqrt{d}}{m} \cdot \left\| \sum_{k=1}^m \sigma_k \Psi_k \Psi_k^\top \right\|_F,$$

where the last step is Cauchy-Schwarz inequality applied to the diagonalized matrix. So we have:

$$\hat{\mathfrak{R}}_S(\mathcal{F}_d) \leq \frac{\sqrt{d}}{m} \mathbb{E}_{\boldsymbol{\sigma}} \left[ \left\| \sum_{k=1}^m \sigma_k \Psi_k \Psi_k^\top \right\|_F \right] \leq \frac{\kappa^2 \sqrt{d}}{\sqrt{m}} \qquad \text{almost surely,}$$

where the last step was proved in Proposition 13 of Zhai et al. (2024). Thus, for the Rademacher complexity we have $\mathfrak{R}_m(\mathcal{F}_d) = \mathbb{E}_S[\hat{\mathfrak{R}}_S(\mathcal{F}_d)] \leq \frac{\kappa^2 \sqrt{d}}{\sqrt{m}}$. Moreover, for $P_\mathcal{X}$-almost all $x$, we have:

$$\sum_{i=1}^d g_i(x)^2 = \Psi(x)^\top \left( \sum_{i=1}^d \boldsymbol{u}_i \boldsymbol{u}_i^\top \right) \Psi(x) = \Psi(x)^\top \boldsymbol{U} \boldsymbol{U}^\top \Psi(x) \leq \|\Psi(x)\|_2^2 \|\boldsymbol{U} \boldsymbol{U}^\top\|_2 = \|\Psi(x)\|_2^2 \leq \kappa^2,$$

where the last step is because $\Psi(x)^\top \Psi(x) = \sum_i \lambda_i \psi_i(x)^2 \leq \kappa^2$ for all $x$. Hence, by Theorem 4.10 of Wainwright (2019), we have:

$$\left| \frac{1}{m} \sum_{i=n+1}^{n+m} f(x_i) - \mathbb{E}_{X \sim P_\mathcal{X}}[f(X)] \right| \leq \frac{\kappa^2}{\sqrt{m}} \left( 2\sqrt{d} + \sqrt{2 \log \frac{2}{\delta}} \right) \qquad \text{for all } f \in \mathcal{F}_d \qquad (19)$$

holds with probability at least $1 - \frac{\delta}{2}$. Let $F(x) = \sum_{i=1}^d \hat{\psi}_i(x)^2$. Then, $F \in \mathcal{F}_d$. And for all $i \in [d]$, there is $[\hat{\psi}_i(x_{n+1}), \cdots, \hat{\psi}_i(x_{n+m})]^\top = \boldsymbol{G}_{k,m} \boldsymbol{v}_i = m\tilde{\lambda}_i \boldsymbol{v}_i$, so $\hat{\psi}_i(x_{n+1})^2 + \cdots + \hat{\psi}_i(x_{n+m})^2 = m^2 \tilde{\lambda}_i^2 \|\boldsymbol{v}_i\|_2^2 = m\tilde{\lambda}_i$. Thus, $\frac{1}{m} \sum_{i=1}^{n+m} F(x_i) = \tilde{\lambda}_1 + \cdots + \tilde{\lambda}_d$. So if Eqn. (19) holds, then

$$\|\hat{\psi}_1\|_{P_\mathcal{X}}^2 + \cdots + \|\hat{\psi}_d\|_{P_\mathcal{X}}^2 \geq \tilde{\lambda}_1 + \cdots + \tilde{\lambda}_d - \frac{\kappa^2}{\sqrt{m}} \left( 2\sqrt{d} + \sqrt{2 \log \frac{2}{\delta}} \right).$$

Since $\hat{\lambda}_1, \cdots, \hat{\lambda}_d$ are the eigenvalues of $\frac{\boldsymbol{G}_{k,m}}{m}$, by Theorem 3.2 of Blanchard et al. (2007), we have:

$$\tilde{\lambda}_1 + \cdots + \tilde{\lambda}_d \geq \lambda_1 + \cdots + \lambda_d - \frac{\kappa^2}{\sqrt{m}} \sqrt{\frac{1}{2} \log \frac{6}{\delta}}$$

holds with probability at least $1 - \frac{\delta}{2}$. By union bound, it holds with probability at least $1 - \delta$ that

$$\|\hat{\psi}_1\|_{P_\mathcal{X}}^2 + \cdots + \|\hat{\psi}_d\|_{P_\mathcal{X}}^2 \geq \lambda_1 + \cdots + \lambda_d - \frac{\kappa^2}{\sqrt{m}} \left( 2\sqrt{d} + 3\sqrt{\log \frac{6}{\delta}} \right).$$

Let $f^* - \tilde{f}_d = bg$, where $b \in \mathbb{R}$, and $g \in \mathcal{H}_K$ satisfies $\|g\|_{\mathcal{H}_K} = 1$ and $\langle g, \hat{\psi}_i \rangle_{\mathcal{H}_K} = 0$ for $i \in [d]$. Then, by Lemma 9, we have

$$\|g\|_{P_\mathcal{X}}^2 \leq \lambda_{d+1} + \frac{\kappa^2}{\sqrt{m}} \left( 2\sqrt{d} + 3\sqrt{\log \frac{6}{\delta}} \right).$$

Let $a = \|\tilde{f}_d\|_{\mathcal{H}_K}$. Then, $\|f^*\|_{\mathcal{H}_K}^2 = a^2 + b^2$. Since $\|f\|_{\mathcal{H}_K}^2 \leq \epsilon M \|f^*\|_{P_\mathcal{X}}^2$, we have:

$$\frac{a^2 + b^2}{\epsilon M} \leq \|f^*\|_{P_\mathcal{X}}^2 \leq \left( \|\tilde{f}_d\|_{P_\mathcal{X}} + b\|g\|_{P_\mathcal{X}} \right)^2 \leq \left( a\sqrt{\lambda_1} + b\|g\|_{P_\mathcal{X}} \right)^2 \leq 2(a^2 \lambda_1 + b^2 \|g\|_{P_\mathcal{X}}^2),$$

which implies that

$$(\lambda_1 - \|g\|_{P_\mathcal{X}}^2)b^2 \leq \left( \lambda_1 - \frac{1}{2\epsilon M} \right)(a^2 + b^2) \leq \left( \lambda_1 \epsilon M - \frac{1}{2} \right)B^2,$$

which completes the proof. $\qquad\square$

## C  DETAILS OF NUMERICAL IMPLEMENTATIONS

---
**Algorithm 3** Directly solving STKR
---
**Input:** $K(x, x')$, $s(\lambda) = \sum_{p=1}^{q} \pi_p \lambda^p$, $\beta_n$, $\boldsymbol{y}$
    # $\boldsymbol{G}_{K,n+m,n}$ *is the left $n$ columns of $\boldsymbol{G}_K$*
1: Initialize: $\boldsymbol{M} \leftarrow \boldsymbol{G}_{K,n+m,n} \in \mathbb{R}^{(n+m) \times n}$
2: $\boldsymbol{A} \leftarrow n\beta_n \boldsymbol{I}_n + \pi_1 \boldsymbol{G}_{K,n} \in \mathbb{R}^{n \times n}$
3: **for** $p = 2, \cdots, q$ **do**
4:     $\boldsymbol{A} \leftarrow \boldsymbol{A} + \frac{\pi_p}{n+m} \boldsymbol{G}_{K,n+m,n}^\top \boldsymbol{M}$
5:     $\boldsymbol{M} \leftarrow \frac{1}{n+m} \boldsymbol{G}_K \boldsymbol{M}$
6: Solve $\boldsymbol{A}\hat{\boldsymbol{\alpha}} = \boldsymbol{y}$
---

STKR amounts to solving $\boldsymbol{A}\hat{\boldsymbol{\alpha}} = \boldsymbol{y}$ for $\boldsymbol{A} = \boldsymbol{G}_{\hat{K}_s,n} + n\beta_n \boldsymbol{I}_n$. First, consider $s(\lambda) = \sum_{p=1}^{q} \pi_p \lambda^p$ with $q < \infty$. Provided that computing $K(x, x')$ for any $x, x'$ takes $O(1)$ time, directly computing $\boldsymbol{A}$ and then solving $\boldsymbol{A}\hat{\boldsymbol{\alpha}} = \boldsymbol{y}$ as described above has a time complexity of $O((n+m)^2 nq)$ as it performs $O(q)$ matrix multiplications, and a space complexity of $O((n+m)n)$. Calculating $\boldsymbol{A}$ directly could be expensive since it may require many matrix-matrix multiplications.

Alternatively we can use iterative methods, such as Richardson iteration that solves $\boldsymbol{A}\hat{\boldsymbol{\alpha}} = \boldsymbol{y}$ with $\hat{\boldsymbol{\alpha}}_{t+1} = \hat{\boldsymbol{\alpha}}_t - \gamma(\boldsymbol{A}\hat{\boldsymbol{\alpha}}_t - \boldsymbol{y})$ for some $\gamma > 0$, as described in Algorithm 1 in the main text. This is faster than the direct method since it replaces matrix-matrix multiplication with matrix-vector multiplication.

It is well-known that with a proper $\gamma$, it takes $O(\kappa(\boldsymbol{A}) \log \frac{1}{\epsilon})$ Richardson iterations to ensure $\|\hat{\boldsymbol{\alpha}}_t - \hat{\boldsymbol{\alpha}}_*\|_2 < \epsilon \|\hat{\boldsymbol{\alpha}}_*\|_2$, where $\hat{\boldsymbol{\alpha}}_*$ is the real solution, and $\kappa(\boldsymbol{A})$ is the condition number of $\boldsymbol{A}$. Let $\lambda$ be a known upper bound of $\lambda_1$ (*e.g.* for augmentation-based pretraining, $\lambda = 1$ (Zhai et al., 2024)). With a sufficiently large $n$, we have $\kappa(\boldsymbol{A}) = O(\beta_n^{-1} s(\lambda))$ almost surely, so Richardson iteration has a total time complexity of $O((n+m)^2 \beta_n^{-1} s(\lambda) q \log \frac{1}{\epsilon})$, and a space complexity of $O(n+m)$. The method can be much faster if $K$ is sparse. For instance, if $K$ is the adjacency matrix of a graph with $|E|$ edges, then each iteration only takes $O(q \cdot |E|)$ time instead of $O(q(n+m)^2)$.

Next, we consider the case where $s$ could be complex, but $s^{-1}(\lambda) = \sum_{p=0}^{q-1} \xi_p \lambda^{p-r}$ is simple, such as the inverse Laplacian (Example 1). In this case we cannot compute $\boldsymbol{G}_{\hat{K}_s,n}$, but if we define $\boldsymbol{Q} := \sum_{p=0}^{q-1} \xi_p \left( \frac{\boldsymbol{G}_K}{n+m} \right)^p$, then there is $\boldsymbol{G}_{\hat{K}_s} \boldsymbol{Q} = (n+m) \left( \frac{\boldsymbol{G}_K}{n+m} \right)^r$. With this observation, we use an indirect approach to find $\hat{\boldsymbol{\alpha}}$, which is to find a $\boldsymbol{\theta} \in \mathbb{R}^{n+m}$ such that $\boldsymbol{Q}\boldsymbol{\theta} = [\hat{\boldsymbol{\alpha}}, \boldsymbol{0}_m]^\top$. Note that by the definition of $\hat{\boldsymbol{\alpha}}$, the first $n$ elements of $\left( \boldsymbol{G}_{\hat{K}_s} + n\beta_n \boldsymbol{I}_{n+m} \right)[\hat{\boldsymbol{\alpha}}, \boldsymbol{0}_m]^\top = \left( \boldsymbol{G}_{\hat{K}_s} + n\beta_n \boldsymbol{I}_{n+m} \right)\boldsymbol{Q}\boldsymbol{\theta} = \left[ (n+m) \left( \frac{\boldsymbol{G}_K}{n+m} \right)^r + n\beta_n \boldsymbol{Q} \right]\boldsymbol{\theta}$ is $\boldsymbol{y}$, which provides $n$ linear equations. The last $m$ elements of $\boldsymbol{Q}\boldsymbol{\theta}$ are zeros, which gives $m$ linear equations. Combining these two gives an $(n+m)$-dimensional linear system, which we simplify as:

$$\boldsymbol{M}\boldsymbol{\theta} = \tilde{\boldsymbol{y}}, \quad \text{where } \boldsymbol{M} := (n+m)\tilde{\boldsymbol{I}}_n \left( \frac{\boldsymbol{G}_K}{n+m} \right)^r + n\beta_n \boldsymbol{Q}, \quad \text{and } \tilde{\boldsymbol{y}} := [\boldsymbol{y}, \boldsymbol{0}_m]^\top. \quad (20)$$

Here, $\tilde{\boldsymbol{I}}_n := \text{diag}\{1, \cdots, 1, 0, \cdots, 0\}$, with $n$ ones and $m$ zeros. Again, we can find $\boldsymbol{\theta}$ by Richardson iteration, as described in Algorithm 2, with $O(n+m)$ space complexity. Let $\boldsymbol{v}_K$ be defined as in

Eqn. (5). Then, at inference time, one can compute $\hat{f}(x) = \boldsymbol{v}_K(x)^\top \left(\frac{\boldsymbol{G}_K}{n+m}\right)^{r-1}\boldsymbol{\theta}$ in $O(n+m)$ time by storing $\left(\frac{\boldsymbol{G}_K}{n+m}\right)^{r-1}\boldsymbol{\theta}$ in the memory.

We now discuss the time complexity of Algorithm 2. We cannot use the previous argument because now $\boldsymbol{M}$ is not symmetrical. Let $\rho(\lambda) := \frac{\lambda^r}{s(\lambda)} = \sum_{p=0}^{q-1}\xi_p\lambda^p$, where $\rho(0) = \xi_0 > 0$. Then, $\rho(\lambda)$ is continuous on $[0, +\infty)$. Denote its maximum and minimum on $[0, \lambda]$ by $\rho_{\max}$ and $\rho_{\min}$, where again $\lambda$ is a known upper bound of $\lambda_1$. Then, we have the following:

**Proposition 10.** *With $\gamma = (n\lambda^r)^{-1}$, Algorithm 2 takes $O\left(\frac{\lambda^r}{\beta_n\rho_{\min}}\log\left(\max\left\{\frac{1}{\epsilon}, \frac{\lambda^r\rho_{\max}\|\boldsymbol{y}\|_2}{n\beta_n^2\rho_{\min}^2\|\hat{\boldsymbol{\alpha}}_*\|_2}\right\}\right)\right)$ iterations so that $\|\hat{\boldsymbol{\alpha}} - \hat{\boldsymbol{\alpha}}_*\|_2 \leq \epsilon\|\hat{\boldsymbol{\alpha}}_*\|_2$ almost surely for sufficiently large $n$, where $\hat{\boldsymbol{\alpha}}_*$ is the ground truth solution and $\hat{\boldsymbol{\alpha}}$ is the output of the algorithm. Each iteration takes $O(\max\{q, r\}(n+m)^2)$ time. Thus, the total time complexity is $O\left((n+m)^2\beta_n^{-1}\frac{\max\{q,r\}\lambda^r}{\rho_{\min}}\log\left(\max\left\{\frac{1}{\epsilon}, \frac{\lambda^r\rho_{\max}\|\boldsymbol{y}\|_2}{n\beta_n^2\rho_{\min}^2\|\hat{\boldsymbol{\alpha}}_*\|_2}\right\}\right)\right)$.*

*Proof.* Let $\hat{\lambda}_1 \geq \cdots \geq \hat{\lambda}_{n+m}$ be the eigenvalues of $\frac{\boldsymbol{G}_K}{n+m}$. It is easy to show that $\boldsymbol{Q}$ has the same eigenvectors as $\frac{\boldsymbol{G}_K}{n+m}$, with eigenvalues $g(\hat{\lambda}_1), \cdots, g(\hat{\lambda}_2)$. By Lemma 2 and Borel-Cantelli lemma, as $n \to \infty$, $\hat{\lambda}_1 \xrightarrow{a.s.} \lambda_1$. For simplicity, let us assume that $\lambda$ is slightly larger than $\lambda_1$, so almost surely there is $\hat{\lambda}_1 \leq \lambda$. Then, all eigenvalues of $\boldsymbol{Q}$ are in $[\rho_{\min}, \rho_{\max}]$.

The first part of this proof is to bound $\|\boldsymbol{u}_t\|_2$, where $\boldsymbol{u}_t := (n+m)\tilde{\boldsymbol{I}}_n\left(\frac{\boldsymbol{G}_K}{n+m}\right)^r(\boldsymbol{\theta}_* - \boldsymbol{\theta}_t)$. Let $\boldsymbol{\theta}_t$ be the $\boldsymbol{\theta}$ at iteration $t$, and $\boldsymbol{\theta}_*$ be the solution to Eqn. (20). Since $\boldsymbol{\theta}_0 = \boldsymbol{0}$, we have

$$
\begin{aligned}
\boldsymbol{\theta}_* - \boldsymbol{\theta}_t =& \left[\left(\boldsymbol{I}_{n+m} - \gamma\left[(n+m)\tilde{\boldsymbol{I}}_n\left(\frac{\boldsymbol{G}_K}{n+m}\right)^r + n\beta_n\boldsymbol{Q}\right]\right)\boldsymbol{\theta}_* + \gamma\tilde{\boldsymbol{y}}\right] \\
& - \left[\left(\boldsymbol{I}_{n+m} - \gamma\left[(n+m)\tilde{\boldsymbol{I}}_n\left(\frac{\boldsymbol{G}_K}{n+m}\right)^r + n\beta_n\boldsymbol{Q}\right]\right)\boldsymbol{\theta}_{t-1} + \gamma\tilde{\boldsymbol{y}}\right] \\
=& \left(\boldsymbol{I}_{n+m} - \gamma\left[(n+m)\tilde{\boldsymbol{I}}_n\left(\frac{\boldsymbol{G}_K}{n+m}\right)^r + n\beta_n\boldsymbol{Q}\right]\right)(\boldsymbol{\theta}_* - \boldsymbol{\theta}_{t-1}) \\
=& \left(\boldsymbol{I}_{n+m} - \gamma\left[(n+m)\tilde{\boldsymbol{I}}_n\left(\frac{\boldsymbol{G}_K}{n+m}\right)^r + n\beta_n\boldsymbol{Q}\right]\right)^t\boldsymbol{\theta}_*.
\end{aligned}
\tag{21}
$$

Note that

$$
\begin{aligned}
& \left(\frac{\boldsymbol{G}_K}{n+m}\right)^{r/2}\left(\boldsymbol{I}_{n+m} - \gamma\left[(n+m)\tilde{\boldsymbol{I}}_n\left(\frac{\boldsymbol{G}_K}{n+m}\right)^r + n\beta_n\boldsymbol{Q}\right]\right) \\
=& \left(\boldsymbol{I}_{n+m} - \gamma\left[(n+m)\left(\frac{\boldsymbol{G}_K}{n+m}\right)^{r/2}\tilde{\boldsymbol{I}}_n\left(\frac{\boldsymbol{G}_K}{n+m}\right)^{r/2} + n\beta_n\boldsymbol{Q}\right]\right)\left(\frac{\boldsymbol{G}_K}{n+m}\right)^{r/2}.
\end{aligned}
$$

Thus, by propagating $\left(\frac{\boldsymbol{G}_K}{n+m}\right)^{r/2}$ from left to right, we get

$$
\left(\frac{\boldsymbol{G}_K}{n+m}\right)^{r/2}(\boldsymbol{\theta}_* - \boldsymbol{\theta}_t) = (\boldsymbol{I}_{n+m} - \gamma\boldsymbol{R})^t\left(\frac{\boldsymbol{G}_K}{n+m}\right)^{r/2}\boldsymbol{\theta}_*,
$$

where $\boldsymbol{R} := (n+m)\left(\frac{\boldsymbol{G}_K}{n+m}\right)^{r/2}\tilde{\boldsymbol{I}}_n\left(\frac{\boldsymbol{G}_K}{n+m}\right)^{r/2} + n\beta_n\boldsymbol{Q}$ is a *p.s.d.* matrix. Denote the smallest and largest eigenvalues of $\boldsymbol{R}$ by $\tilde{\lambda}_{\min}$ and $\tilde{\lambda}_{\max}$. Then, $\tilde{\lambda}_{\min} \geq n\beta_n\rho_{\min}$. In terms of $\tilde{\lambda}_{\max}$, we have

$$
(n+m)\left(\frac{\boldsymbol{G}_K}{n+m}\right)^{r/2}\tilde{\boldsymbol{I}}_n\left(\frac{\boldsymbol{G}_K}{n+m}\right)^{r/2} = \left(\frac{\boldsymbol{G}_K}{n+m}\right)^{\frac{r-1}{2}}\left(\boldsymbol{G}_K^{\frac{1}{2}}\tilde{\boldsymbol{I}}_n\boldsymbol{G}_K^{\frac{1}{2}}\right)\left(\frac{\boldsymbol{G}_K}{n+m}\right)^{\frac{r-1}{2}}.
$$

By Sylvester's theorem, all non-zero eigenvalues of $\boldsymbol{G}_K^{\frac{1}{2}}\tilde{\boldsymbol{I}}_n\boldsymbol{G}_K^{\frac{1}{2}}$ are the eigenvalues of $\tilde{\boldsymbol{I}}_n\boldsymbol{G}_K\tilde{\boldsymbol{I}}_n$, *i.e.* the non-zero eigenvalues of $\boldsymbol{G}_{K,n}$. By Lemma 2, $\frac{1}{n}\|\boldsymbol{G}_{K,n}\|_2 \xrightarrow{a.s.} \lambda_1$, so suppose $\|\boldsymbol{G}_{K,n}\|_2 \leq n\lambda$. Then, $\tilde{\lambda}_{\max} \leq n\lambda^r + n\beta_n\rho_{\max}$.

Since $M\theta_* = \tilde{y}$, and $\left(\frac{G_K}{n+m}\right)^{r/2} M = R\left(\frac{G_K}{n+m}\right)^{r/2}$, we have $R\left(\frac{G_K}{n+m}\right)^{r/2}\theta_* = \left(\frac{G_K}{n+m}\right)^{r/2}\tilde{y}$. Note that $R(I_{n+m} - \gamma R) = (I_{n+m} - \gamma R)R$. Thus, we have

$$\left(\frac{G_K}{n+m}\right)^{r/2}(\theta_* - \theta_t) = (I_{n+m} - \gamma R)^t\left(\frac{G_K}{n+m}\right)^{r/2}\theta_*$$

$$= R^{-1}(I_{n+m} - \gamma R)^t R\left(\frac{G_K}{n+m}\right)^{r/2}\theta_*$$

$$= R^{-1}(I_{n+m} - \gamma R)^t\left(\frac{G_K}{n+m}\right)^{r/2}\tilde{y}.$$

Now we bound $\|u_t\|_2$. First, note that for any matrices $A, B \in \mathbb{R}^{d\times d}$ where $B$ is p.s.d., there is $u^\top A^\top BAu \le \|B\|_2\|Au\|_2^2 \le \|B\|_2\|A^\top A\|_2\|u\|_2^2$ for any $u \in \mathbb{R}^d$, so $\|A^\top BA\|_2 \le \|B\|_2\|A^\top A\|_2$. Second, note that the last $m$ elements of $\tilde{y}$ are zeros, which means that $\tilde{y} = \tilde{I}_n\tilde{y}$. Thus, we have

$$\|u_t\|_2 = \left\|(n+m)\tilde{I}_n\left(\frac{G_K}{n+m}\right)^r(\theta_* - \theta_t)\right\|_2$$

$$= \left\|(n+m)\tilde{I}_n\left(\frac{G_K}{n+m}\right)^{r/2}R^{-1}(I_{n+m} - \gamma R)^t\left(\frac{G_K}{n+m}\right)^{r/2}\tilde{y}\right\|_2$$

$$= \left\|(n+m)\tilde{I}_n\left(\frac{G_K}{n+m}\right)^{r/2}(I_{n+m} - \gamma R)^{t/2}R^{-1}(I_{n+m} - \gamma R)^{t/2}\left(\frac{G_K}{n+m}\right)^{r/2}\tilde{I}_n\tilde{y}\right\|_2$$

$$\le \left\|(I_{n+m} - \gamma R)^{t/2}R^{-1}(I_{n+m} - \gamma R)^{t/2}\right\|_2\left\|(n+m)\tilde{I}_n\left(\frac{G_K}{n+m}\right)^r\tilde{I}_n\right\|_2\|\tilde{y}\|_2$$

$$\le \frac{1}{\tilde{\lambda}_{\min}}\|I_{n+m} - \gamma R\|_2^t(n\lambda_1^r)\|y\|_2,$$

where the last step is because we have already proved $\left\|(n+m)\tilde{I}_n\left(\frac{G_K}{n+m}\right)^r\tilde{I}_n\right\|_2 \le n\lambda_1^r$.

Now, for $\gamma = \frac{1}{n\lambda^r}$, when $n$ is sufficiently large it is less than $\frac{2}{\tilde{\lambda}_{\max}+\tilde{\lambda}_{\min}}$, because $\beta_n = o(1)$. Thus, $\|I_{n+m} - \gamma R\|_2 \le 1 - \frac{\tilde{\lambda}_{\min}}{n\lambda^r} \le 1 - \frac{\beta_n\rho_{\min}}{\lambda^r}$. Thus, we have

$$\|u_t\|_2 \le \left(1 - \frac{\beta_n\rho_{\min}}{\lambda^r}\right)^t\frac{\lambda^r}{\beta_n\rho_{\min}}\|y\|_2.$$

The second part of this proof is to bound $\|Q(\theta_* - \theta_t)\|_2$. Let us return to Eqn. (21), which says that
$$\|Q(\theta_* - \theta_{t+1})\|_2 = \|(I_{n+m} - \gamma n\beta_n Q)Q(\theta_* - \theta_t) - \gamma Qu_t\|_2$$

$$\le \left(1 - \frac{\beta_n\rho_{\min}}{\lambda^r}\right)\|Q(\theta_* - \theta_t)\|_2 + \frac{\rho_{\max}}{n\lambda^r}\|u_t\|_2.$$

Here again, we assume that $n$ is large enough so that $\lambda^r > \beta_n\rho_{\min}$. This implies that

$$\|Q(\theta_* - \theta_{t+1})\|_2 - t\left(1 - \frac{\beta_n\rho_{\min}}{\lambda^r}\right)^t\frac{\rho_{\max}\|y\|_2}{n\beta_n\rho_{\min}}$$

$$\le \left(1 - \frac{\beta_n\rho_{\min}}{\lambda^r}\right)\left[\|Q(\theta_* - \theta_t)\|_2 - (t-1)\left(1 - \frac{\beta_n\rho_{\min}}{\lambda^r}\right)^{t-1}\frac{\rho_{\max}\|y\|_2}{n\beta_n\rho_{\min}}\right]$$

$$\le \cdots \le \left(1 - \frac{\beta_n\rho_{\min}}{\lambda^r}\right)^t\left[\left(1 - \frac{\beta_n\rho_{\min}}{\lambda^r}\right)\|Q\theta_*\|_2 + \frac{\rho_{\max}\|y\|_2}{n\beta_n\rho_{\min}}\right].$$

Thus, there is $\|Q(\theta_* - \theta_t)\|_2 \le \left(1 - \frac{\beta_n\rho_{\min}}{\lambda^r}\right)^t\|Q\theta_*\|_2 + t\left(1 - \frac{\beta_n\rho_{\min}}{\lambda^r}\right)^{t-1}\frac{\rho_{\max}\|y\|_2}{n\beta_n\rho_{\min}}$. Using $1 - x \le e^{-x}$, we have

$$\|Q(\theta_* - \theta_t)\|_2 \le \exp\left(-\frac{\beta_n\rho_{\min}t}{\lambda^r}\right)\|Q\theta_*\|_2 + t\exp\left(-\frac{\beta_n\rho_{\min}(t-1)}{\lambda^r}\right)\frac{\rho_{\max}\|y\|_2}{n\beta_n\rho_{\min}}.$$

When $t = t_0 := \frac{4\lambda^r}{\beta_n \rho_{\min}} \log \frac{2\lambda^r \rho_{\max} \|\boldsymbol{y}\|_2}{n \beta_n^2 \rho_{\min}^2 \|\boldsymbol{Q\theta}_*\|_2}$, by $\log(2x) \le x$ for $x > 0$, we have

$$\exp\left(\frac{\beta_n \rho_{\min}}{\lambda^r} \frac{t}{2}\right) \ge \left(\frac{2\lambda^r \rho_{\max} \|\boldsymbol{y}\|_2}{n \beta_n^2 \rho_{\min}^2 \|\boldsymbol{Q\theta}_*\|_2}\right)^2 \ge \frac{4\lambda^r \rho_{\max} \|\boldsymbol{y}\|_2}{n \beta_n^2 \rho_{\min}^2 \|\boldsymbol{Q\theta}_*\|_2} \log\left(\frac{2\lambda^r \rho_{\max} \|\boldsymbol{y}\|_2}{n \beta_n^2 \rho_{\min}^2 \|\boldsymbol{Q\theta}_*\|_2}\right).$$

Let $F(t) := \exp\left(\frac{\beta_n \rho_{\min}}{2\lambda^r} t\right) - \frac{\rho_{\max} \|\boldsymbol{y}\|_2}{n \beta_n \rho_{\min} \|\boldsymbol{Q\theta}_*\|_2} t$. Then we have $F(t_0) \ge 0$. And it is easy to show that for all $t \ge \frac{t_0}{2}$, there is $F'(t) \ge 0$. This means that when $t \ge t_0$, there is $F(t) \ge 0$, so we have

$$\|\boldsymbol{Q}(\boldsymbol{\theta}_* - \boldsymbol{\theta}_t)\|_2 \le \exp\left(-\frac{\beta_n \rho_{\min} t}{\lambda^r}\right) \|\boldsymbol{Q\theta}_*\|_2 + \exp\left(-\frac{\beta_n \rho_{\min}}{\lambda^r}\left(\frac{t}{2} - 1\right)\right) \|\boldsymbol{Q\theta}_*\|_2.$$

Hence, when $t \ge \max\left\{\frac{2\lambda^r}{\beta_n \rho_{\min}} \log \frac{2}{\epsilon} + 2, t_0\right\}$, we have $\|\boldsymbol{Q}(\boldsymbol{\theta}_* - \boldsymbol{\theta}_t)\|_2 \le \epsilon \|\boldsymbol{Q\theta}_*\|_2$, which implies that the relative estimation error of $\hat{\boldsymbol{\alpha}}$ is less than $\epsilon$. $\qquad\square$

## D  EXPERIMENTS

The purpose of our experiments is threefold:

(i) Verify that STKR-Prop (Algorithms 1 and 2) works with general polynomial $s(\lambda)$ including inverse Laplacian under transductive and inductive settings on real datasets, and compare them to label propagation (for the transductive setting).

(ii) Explore possible reasons why canonical Laplacian works so well empirically, by examining the effect of $p$ on the performance when using STKR with $s(\lambda) = \lambda^p$.

(iii) Verify that STKR-Prop works with kernel PCA on real datasets, and compare it to other methods.

### D.1  SETUP

**Datasets.**  We focus on graph node classification tasks, and work with the publicly available datasets in the *PyTorch Geometric* library (Fey & Lenssen, 2019), among which Cora, CiteSeer and PubMed are based on Yang et al. (2016); Computers, Photos, CS and Physics are based on Shchur et al. (2018); DBLP and CoraFull are based on Bojchevski & Günnemann (2018). See Table 3 for a summary of the dataset statistics.

**Train/val/test/other splits.**  We split a dataset into four sets: train, validation (val), test and other. Among them, train and val contain labeled samples, while test and other contain unlabeled samples. **The test performance which we will report later is only evaluated on the test set.** The val set is used to select the best model, so it is used in a similar way as the test set as explained below:

- In the transductive setting, the learner can see all four sets at train time. The learner manually hides the labels of the samples in the val set (so that the val performance approximates the test performance). Thus, $n$ is the size of the train set, while $m$ is the size of the other three combined.

- In the inductive setting, the learner can see train, val and other, but not test. Neither can it see any edges connected to test nodes. Then, the learner manually hides the entire val set (nodes, outcoming edges and labels), so that the val performance approximates the test performance. Thus, $n$ is the size of the train set, while $m$ is the size of the other set.

In all our experiments, these four sets are randomly split. This means that with the same random seed, the four splits are exactly the same; With a different random seed, the four splits are different, but their sizes are kept the same for the same dataset.

**Sizes of the splits.**  First, we specify a hyperparameter $p_{\text{test}}$, and then $p_{\text{test}}$ of all the samples will be in the test set. For Cora, CiteSeer and PubMed, we use the default train/validation/test split size, and from the test set we take out $p_{\text{test}} \times$ #(all samples) of the samples to be the real test set, and the rest of the samples go to the other set. For the other six datasets, we set the train and validation set size to be $20 \times$ number of classes. For example, the Physics dataset has 5 classes, so we randomly sample 100 samples to be train data, and another 100 samples to be validation data. We also do an ablation study for $p_{\text{test}}$ in our experiments, where $p_{\text{test}}$ could range from 1% to 50%.

Table 3: Number of classes, nodes, edges, and fractions (%) of train and validation sets.

|                     | Classes | Nodes  | Edges   | Train | Validation |
|---------------------|---------|--------|---------|-------|------------|
| Cora                | 7       | 2,708  | 10,556  | 5.17  | 18.46      |
| CiteSeer            | 6       | 3,327  | 9,104   | 3.61  | 15.03      |
| PubMed              | 3       | 19,717 | 88,648  | 0.3   | 2.54       |
| Amazon - Computers  | 10      | 13,752 | 491,722 | 1.45  | 1.45       |
| Amazon - Photos     | 8       | 7,650  | 238,162 | 2.09  | 2.09       |
| Coauthor - CS       | 15      | 18,333 | 163,788 | 1.64  | 1.64       |
| Coauthor - Physics  | 5       | 34,493 | 495,924 | 0.29  | 0.29       |
| DBLP                | 4       | 17,716 | 105,734 | 0.45  | 0.45       |
| CoraFull            | 70      | 19,793 | 126,842 | 7.07  | 7.07       |

**Implementations.** For label propagation, we use the version in Zhou et al. (2003), which solves:

$$(\boldsymbol{I}_{n+m} - \eta \boldsymbol{S})\hat{\boldsymbol{y}} = \tilde{\boldsymbol{y}}, \qquad \text{where } \tilde{\boldsymbol{y}} := [\boldsymbol{y}, \boldsymbol{0}_m]^\top. \tag{22}$$

Here $\hat{\boldsymbol{y}}$ is the predicted labels for all $n + m$ samples under the transductive setting, and $\boldsymbol{S}$ is defined as $\boldsymbol{S} := \boldsymbol{D}^{-\frac{1}{2}}\boldsymbol{W}\boldsymbol{D}^{-\frac{1}{2}}$, where $\boldsymbol{W}$ is the adjacency matrix such that $\boldsymbol{W}[i, j] = 1$ if $x_i$ is connected to $x_j$ and 0 otherwise, and $\boldsymbol{D}$ is a diagonal matrix defined as $\boldsymbol{D}[i, i] = \sum_{j=1}^{n+m} \boldsymbol{W}[i, j]$ for $i \in [n + m]$. For STKR, we define the base kernel $K$ as:

$$K(x, x') = (n + m)\frac{W(x, x')}{\sqrt{D(x)D(x')}}, \tag{23}$$

where $W(x_i, x_j) = \boldsymbol{W}[i, j]$, and for the transductive setting there is $D(x_i) = \boldsymbol{D}[i, i]$, so that $\boldsymbol{S} = \frac{\boldsymbol{G}_K}{n+m}$. For the inductive setting, $D(x_i) = \sum_{j \notin \text{test nodes}} W(x_i, x_j)$ for all $i \in [n + m]$, *i.e.* the sum is only taken over the visible nodes at train time.

**Hyperparameters.** Below are the hyperparameters we use in the experiments. Best hyperparameters are selected with the validation split as detailed above.

- **Label Propagation**
    - Number of iteration $T \in [1, 2, 4, 8, 16, 32]$
    - $\eta \in [0.7, 0.8, 0.9, 0.99, 0.999, 0.9999, 0.99999, 0.999999]$
- **STKR transductive**
    - Number of iteration $T \in [1, 2, 4, 8, 16, 32]$
    - Laplacian $s^{-1}(\lambda) = \lambda^{-1} - \eta$: $\eta \in [0.7, 0.8, 0.9, 0.99, 0.999, 0.9999, 0.99999, 0.999999]$
    - Polynomial $s(\lambda) = \lambda^k$: $k \in [1, 2, 4, 6, 8]$
    - $\beta \in [10^3, 10^2, 10^1, 10^0, 10^{-1}, 10^{-2}, 10^{-3}, 10^{-4}, 10^{-5}, 10^{-6}, 10^{-7}, 10^{-8}]$
- **STKR inductive**
    - Laplacian $s^{-1}(\lambda) = \lambda^{-1} - \eta$: $\eta \in [0.7, 0.8, 0.9, 0.99, 0.999, 0.9999, 0.99999, 0.999999]$
    - Polynomial $s(\lambda) = \lambda^k$: $k \in [1, 2, 4, 6, 8]$
    - $\beta \in [10^3, 10^2, 10^1, 10^0, 10^{-1}, 10^{-2}, 10^{-3}, 10^{-4}, 10^{-5}, 10^{-6}, 10^{-7}, 10^{-8}]$
- **Kernel PCA**
    - Number of representation dimension $d \in [32, 64, 128, 256, 512]$
    - $\beta \in [10^3, 10^2, 10^1, 10^0, 10^{-1}, 10^{-2}, 10^{-3}, 10^{-4}, 10^{-5}, 10^{-6}, 10^{-7}, 10^{-8}]$

### D.2 RESULTS

We report the test accuracy of STKR-Prop with different transformations and the Label-Prop algorithm in Table 4. To make a fair comparison between the transductive and inductive setting, we report the test accuracy on the same $p_{\text{test}} = 0.01$ fraction of the data (for the same random seed). This is a part of the unlabeled data in the transductive setting, but is completely hidden from the learner in the inductive setting at train time.

First, our results show that STKR-Prop can work pretty well with a general $s(\lambda)$ under the inductive setting. The drops in the accuracy of the inductive STKR-Prop compared to transductive are small. In

Table 4: The test accuracy (%) of Label-Prop (LP), STKR-Prop (SP) with inverse Laplacian (Lap), with polynomial $s(\lambda) = \lambda^8$ (poly), with kernel PCA (topd), and with $s(\lambda) = \lambda$ (KRR). (t) and (i) indicate the transductive and inductive settings. We report the test accuracy when $p_{\text{test}} = 0.01$, *i.e.* test samples account for 1% of all samples. Standard deviations are given across ten random seeds. The best and second-best results for each dataset are marked in red and blue, respectively.

| | CS | CiteSeer | Computers | Cora | CoraFull | DBLP | Photo | Physics | PubMed |
|---|---|---|---|---|---|---|---|---|---|
| LP (t) | $79.07_{2.19}$ | $52.73_{7.72}$ | $77.30_{3.05}$ | $73.33_{6.00}$ | $54.47_{3.24}$ | $66.44_{3.78}$ | $83.95_{5.78}$ | $84.33_{4.86}$ | $72.28_{5.55}$ |
| SP-Lap (t) | $78.96_{2.53}$ | $52.12_{7.67}$ | $77.81_{3.94}$ | $77.04_{5.74}$ | $53.81_{2.34}$ | $65.42_{5.02}$ | $84.08_{6.52}$ | $84.22_{4.86}$ | $71.93_{4.86}$ |
| SP-poly (t) | $79.13_{2.29}$ | $48.79_{8.51}$ | $76.72_{4.12}$ | $71.48_{5.80}$ | $53.25_{3.54}$ | $64.52_{4.20}$ | $79.21_{7.20}$ | $84.45_{4.89}$ | $72.18_{4.66}$ |
| SP-topd (t) | $78.80_{3.22}$ | $46.06_{1.08}$ | $80.80_{3.06}$ | $69.26_{7.82}$ | $50.36_{2.85}$ | $64.86_{4.60}$ | $84.61_{6.30}$ | $83.20_{2.25}$ | $65.38_{5.66}$ |
| SP-Lap (i) | $78.42_{2.81}$ | $46.06_{6.97}$ | $77.15_{2.64}$ | $67.78_{7.62}$ | $53.30_{3.24}$ | $65.20_{4.92}$ | $84.87_{5.66}$ | $83.11_{5.09}$ | $70.36_{4.80}$ |
| SP-poly (i) | $79.02_{2.42}$ | $44.55_{9.15}$ | $71.97_{4.13}$ | $65.19_{9.11}$ | $51.98_{3.88}$ | $64.52_{4.05}$ | $78.42_{7.80}$ | $84.68_{4.83}$ | $70.76_{4.28}$ |
| SP-topd (i) | $79.13_{3.35}$ | $41.52_{6.71}$ | $80.80_{3.28}$ | $63.70_{6.00}$ | $47.41_{3.39}$ | $63.16_{3.41}$ | $85.53_{5.68}$ | $82.44_{3.88}$ | $64.31_{4.95}$ |
| KRR (i) | $13.11_{2.29}$ | $13.64_{5.93}$ | $26.35_{4.34}$ | $28.52_{8.56}$ | $19.80_{2.22}$ | $44.80_{3.86}$ | $33.95_{7.07}$ | $19.74_{1.46}$ | $20.76_{2.06}$ |

Table 5: Test accuracy (%) of STKR-Prop (SP) with inverse Laplacian (Lap), with polynomial $s(\lambda) = \lambda^8$ (poly) and with kernel PCA (topd) for inductive setting with different value of $p_{\text{test}} \in \{0.01, 0.05, 0.1, 0.2, 0.3\}$. Standard deviations are given across ten random seeds.

| Methods | $p_{\text{test}}$ | CS | CiteSeer | Computers | Cora | CoraFull | DBLP | Photo | Physics | PubMed |
|---|---|---|---|---|---|---|---|---|---|---|
| | 0.01 | $78.42_{2.81}$ | $46.06_{6.97}$ | $77.15_{2.64}$ | $67.78_{7.62}$ | $53.30_{3.24}$ | $65.20_{4.92}$ | $84.87_{5.66}$ | $83.11_{5.09}$ | $70.36_{4.80}$ |
| | 0.05 | $77.93_{1.41}$ | $41.75_{4.82}$ | $77.42_{2.25}$ | $62.37_{3.66}$ | $51.63_{0.90}$ | $65.57_{2.52}$ | $84.90_{2.50}$ | $82.72_{4.26}$ | $67.81_{3.56}$ |
| SP-Lap | 0.1 | $76.45_{1.17}$ | $39.97_{2.68}$ | $77.40_{1.88}$ | $59.74_{2.37}$ | $50.22_{0.75}$ | $65.34_{2.09}$ | $84.10_{1.84}$ | $82.02_{4.01}$ | $66.43_{3.62}$ |
| | 0.2 | $75.06_{1.01}$ | $35.50_{1.94}$ | $77.18_{2.03}$ | $55.10_{2.92}$ | $47.84_{0.79}$ | $63.24_{1.96}$ | $83.81_{1.23}$ | $81.33_{3.59}$ | $63.77_{3.36}$ |
| | 0.3 | $72.89_{0.95}$ | $30.92_{1.14}$ | $76.59_{1.71}$ | $50.22_{2.87}$ | $44.96_{0.77}$ | $60.44_{1.68}$ | $83.28_{1.11}$ | $80.27_{3.45}$ | $60.25_{2.75}$ |
| | 0.01 | $79.13_{3.35}$ | $44.55_{9.15}$ | $71.97_{4.13}$ | $65.19_{9.11}$ | $51.98_{3.88}$ | $64.52_{4.05}$ | $78.42_{7.80}$ | $84.68_{4.83}$ | $70.76_{4.28}$ |
| | 0.05 | $78.74_{1.42}$ | $40.36_{5.51}$ | $73.04_{1.80}$ | $61.85_{4.15}$ | $50.21_{1.69}$ | $64.64_{2.07}$ | $79.03_{4.61}$ | $84.05_{3.99}$ | $67.68_{2.91}$ |
| SP-poly | 0.1 | $77.51_{1.03}$ | $38.58_{2.90}$ | $73.10_{1.58}$ | $58.56_{2.59}$ | $49.06_{0.90}$ | $64.21_{1.93}$ | $78.59_{4.49}$ | $83.13_{3.69}$ | $66.08_{2.96}$ |
| | 0.2 | $75.74_{1.02}$ | $33.65_{1.78}$ | $72.70_{1.84}$ | $53.11_{2.39}$ | $46.53_{0.79}$ | $61.89_{2.03}$ | $78.82_{3.05}$ | $82.36_{3.27}$ | $62.97_{3.10}$ |
| | 0.3 | $73.35_{0.76}$ | $28.98_{1.21}$ | $72.35_{1.60}$ | $47.99_{2.62}$ | $43.42_{0.78}$ | $59.55_{2.02}$ | $78.20_{2.73}$ | $81.08_{3.25}$ | $59.32_{2.57}$ |
| | 0.01 | $79.13_{3.35}$ | $41.52_{6.71}$ | $80.80_{3.28}$ | $63.70_{6.00}$ | $47.41_{3.39}$ | $63.16_{3.41}$ | $85.53_{5.68}$ | $82.44_{3.88}$ | $64.31_{4.95}$ |
| | 0.05 | $78.37_{1.58}$ | $40.00_{5.14}$ | $80.17_{2.30}$ | $61.70_{3.53}$ | $47.17_{1.63}$ | $62.79_{3.36}$ | $85.47_{2.07}$ | $82.26_{1.88}$ | $62.79_{3.28}$ |
| SP-topd | 0.1 | $77.17_{1.02}$ | $38.67_{4.17}$ | $79.35_{2.57}$ | $58.81_{3.26}$ | $45.22_{0.89}$ | $61.51_{2.60}$ | $84.85_{1.70}$ | $80.59_{1.86}$ | $61.75_{2.26}$ |
| | 0.2 | $75.61_{0.64}$ | $35.10_{2.64}$ | $79.00_{2.02}$ | $55.51_{4.21}$ | $42.31_{0.91}$ | $60.30_{2.60}$ | $84.63_{1.45}$ | $80.15_{1.86}$ | $59.60_{3.22}$ |
| | 0.3 | $72.59_{0.70}$ | $32.92_{1.92}$ | $78.09_{1.27}$ | $51.71_{4.39}$ | $38.62_{0.79}$ | $59.18_{2.29}$ | $83.91_{1.69}$ | $79.07_{2.33}$ | $57.50_{2.50}$ |

Table 5, we further provide an ablation on the test accuracy as we increase $p_{\text{test}}$. As $p_{\text{test}}$ is larger, the performance of inductive STKR decreases across the board. Nevertheless, there are many datasets such as Photo, Physics, Computer where the performance drop is fairly small — at around $2 - 3$ percent even when $p_{\text{test}} = 0.3$. Our ablation study shows that inductive STKR is quite robust to the number of available unlabeled data at the training time. Our experiment clearly demonstrates that one can implement STKR with a general transformation such as $s(\lambda) = \lambda^p$ efficiently. The running time of STKR-Prop is similar to that of Label-Prop with the same number of iterations.

Second, we explore the impact of the "number of hop" $p$ on the performance of STKR with $s(\lambda) = \lambda^p$ (Figure 2). We consider $p \in \{1, 2, 4, 6, 8\}$, and note that for $p = 1$, this STKR is equivalent to performing a KRR with the base kernel. As we have already seen in Table 4, the performance of such KRR is extremely poor compared to the other methods. This is consistent with our analysis in Section 2 that KRR with the base kernel is not sufficient. We find that by increasing $p$ which in turns increase the additional smoothness requirement, the performance of STKR increases by a large margin for all datasets. This clearly illustrates the benefits of the transitivity of similarity, and offers a possible explanation why the inverse Laplacian transformation performs so well in practice: It uses multi-hop similarity information up to infinitely many hops.

Third, the results show that STKR also works pretty well with kernel PCA. Comparing between kernel PCA and LP (or STKR with inverse Laplacian), on 3 of the 9 datasets we use, the former is better. Thus, this experiment clearly demonstrates the parallel nature of these two methodologies — STKR with inverse Laplacian, and kernel PCA.

Finally, we provide an ablation study about the effect of $\eta$ in inverse Laplacian on the performance of SP-Lap. Recall that for inverse Laplacian we have $s^{-1}(\lambda) = \lambda^{-1} - \eta$. Our observation is that when $\eta$ is very close to 0, the performance is low; once it is bounded away from 0, the performance is fairly consistent, and gets slightly better with a larger $\eta$. Only on dataset CS do we observe a significant performance boost as $\eta$ increases.

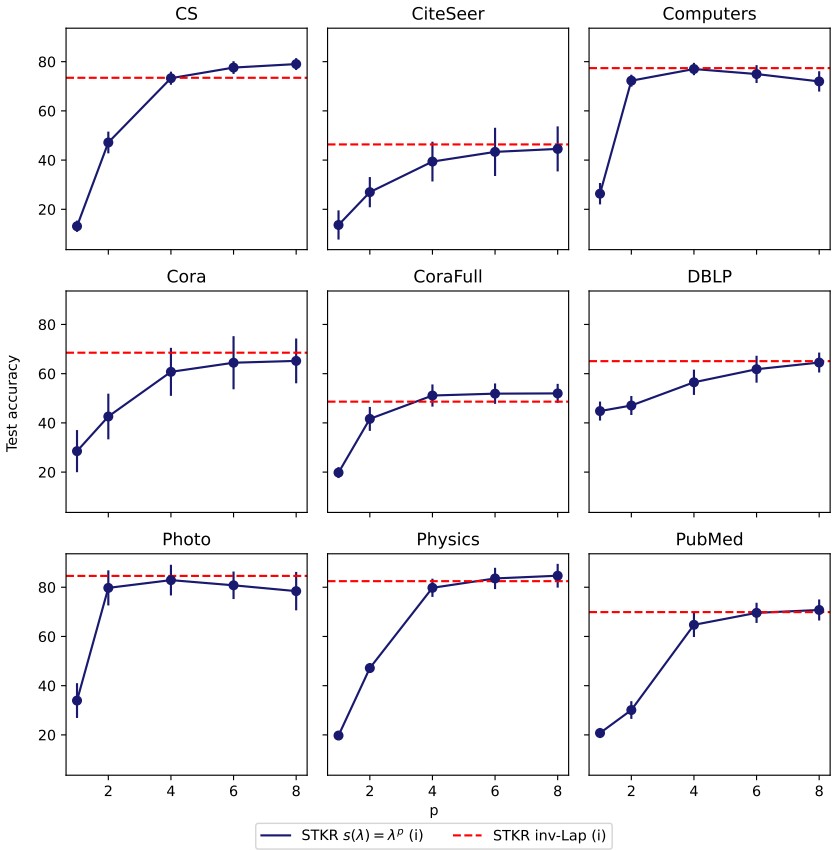

Figure 2: Test accuracy (%) of STKR-Prop (SP) with polynomial with $s(\lambda) = \lambda^p$ for $p \in \{1, 2, 4, 6, 8\}$. The test accuracy increases significantly as $p$ is larger than 1, illustrating the benefits of the transitivity of similarity.

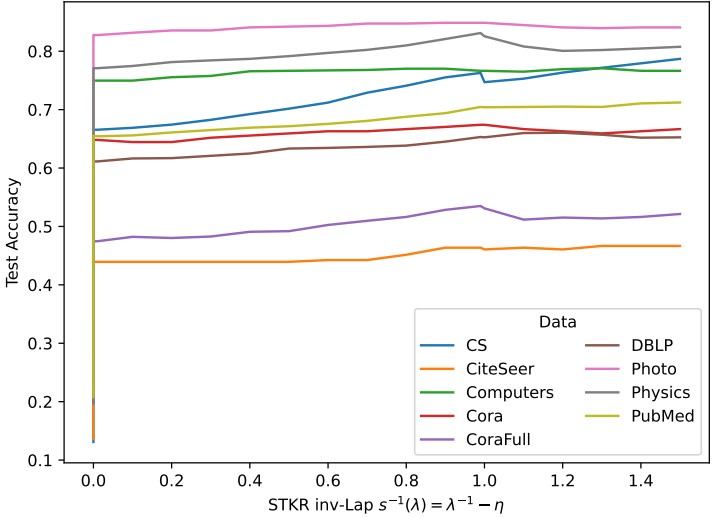

Figure 3: Test accuracy of SP-Lap with different values of $\eta$. The test accuracy is fairly consistent as long as $\eta$ is not too close to 0, and gets slightly better with a larger $\eta$. All reported performances are averaged across ten random seeds.

