# OpenReview forum: "Spectrally Transformed Kernel Regression"
_ICLR.cc/2024/Conference — ICLR 2024 spotlight_

### Official Review · Reviewer_aR59 · 2023-11-02

**Soundness:** 3 good
**Presentation:** 3 good
**Contribution:** 4 excellent
**Rating:** 8
**Confidence:** 4

**Summary:**

This work stands at the crossroads of kernel learning and regression, in close connection with manifold learning. The contribution is presented as a novel theory to incorporate both unlabeled and labeled data into the resolution of the regression task (see our comments).  The authors propose to address an inductive regression estimation problem with the additional assumption that the target function satisfies a smoothness constraint defined with respect to a given Mercer kernel-induced metric. A key result of the paper is that the target is proved to be attainable within a RKHS based on a spectrally transformed kernel (say $H_{K_s}$).
Empirically this problem can be handled with the choice of a single regularization term that writes as a variance term in the RKHS H_{K_s}.
The authors then propose two novel (closed-form) estimators that rely on both kernel and representation coefficients learning. In the first setting, the user is supposed to know exactly the spectral transformation of the eigenvalues at work in the spectrally transformed kernel built from the eigenvalues of the original base kernel. A computable kernel $\hat{K}_s$ is recursively built and a closed-form solution is provided for the empirical ridge regression problem in the RKHS associated with the approximate kernel. This novel estimator comes with an minimax optimal excess risk bound. In the second setting, a (two-step) transform-agnostic estimator is proposed with $K_s$ defined as the inverse Laplacian and a finite feature map representation based on kernel PCA applied on unlabeled data. The novel estimator is also studied at the lens of an excess risk bound. Numerical experiments complete the picture with a comparison of  the STKR different variants to Label-Propagation.

**Strengths:**

*Overall, the paper proposes a rich framework that tackles both representation learning and regression estimation within RKHSs. It follows a long line of research linked to manifold learning in a transductive or inductive way and a spectral approach to kernel learning. The works presented here are substantial with a solid theoretical back up and extensive discussions. I appreciate the elegance of the approach that inherently incorporates into the joint choice of the regularization term  (variance) and the kernel choice the smoothness contraints without leveraging the two regularization terms usually at work in manifold learning. It is a pleasant paper to read, even if it is dense.

*The proposed framework can be naturally applied to transductive or inductive setting, the latter being the most interesting.
Overall the work shed slight on how kernel learning can be considered in a systematic and powerful way for a general regression problem leveraging unlabeled and labeled data.
* The work appears as original and rather well written even if the writing can still be improved.

**Weaknesses:**

* Please note that my score is currently 7
* Claims: it is stated in the abstract and in the paper many times, that this contribution provides a unifyied theory for learning with unlabeled data and a base kernel. I think this message does not hold and is misleading for the reader. I consider that the paper provides a novel class of regression estimators able to exploit labeled and unlabeled data by considering that the right space to work in is the RKHS associated to a Spectrally transformed kernel or its approximation.
*I am surprised that the discussion in page 3 takes classic kernel ridge regression as a reference method while manifold learning with Laplacian regularization (in addition to $\ell_2$ norm ) would be more interesting to discuss here : see Belkin et al. 2006 JMLR to be cited (other papers of Belkin are cited, though)
Belkin, M., Niyogi, P., & Sindhwani, V. (2006). Manifold regularization: A geometric framework for learning from labeled and unlabeled examples. Journal of machine learning research, 7(11).
In particular, the example discussed in figure 1 could have been solved with manifold learning.
at this stage of the paper, I see the advantage of defining the right regularization term with the right kernel to empower the RKHS with the good smoothness properties. However I am a bit confused then by example 1 given in page 7.
* In Example 1 page 7 (Inverse laplacian), I am interested in a discussion about the pros and cons of the proposed method where $K_s$ is defined as a combination of two kernels with a hyperamater eta that seems to play a similar  role that the weight controlling the importance of $\ell_2$ norm and the Laplacian smoothness penalty. Here, I don't see how one can take into account the dependency of the excess risk on this "hidden" parameter.
* originality: the paper resembles a bit to the recent work of Zhai et al. (2023) and it should be interested to highlight the differences; I will advise the authors not to go to far in the direction of data augmentation to strengthen the differences.
* the paper is very dense and clarity can be improved. Some suggestions:
Please provide a table (can be in the first page of the supplements) with a reminder of all the notations for the various kernels, $K_s$, $K^p$, \tilde{f}, \hat{f}, ..
* clearly state when you introduce the intuition and then formally state the theoretical results, there is a mix of evreything all along the paper which makes the paper sometimes difficult to read
* in experimental results (last page): complete the legend so that the reader can see what is the used criterion here (accuracy with a post-processing of the regression estimation outputs).

**Questions:**

* There is no discussion about the base kernel K upon which the spectrally-transformed kernel is built. How to choose it ?
* finally to improve the clarity of the paper:
- please clearly state when you introduce the intuition and then formally state the theoretical results, there is a mix of evreything all along the paper which makes the paper sometimes difficult to read?
- express more clearly Theorem 1: how is defined r_t prior to the conclusion: only a function from $L^2(P_X)$ to $\mathbb{R}$ ?
- explain the importance of centering the functions or alternatively using a "variance term" as a penalty.
- give the analytic complexity in time of the algorithms
- in the experiments provide the behaviour of Mainfold learning (Belkin et al. 2006), especially the closed-form with ridge and laplacian regularisation.

I have read the author's rebuttal and am satisfied the answers. I increase my score from 7 to 8.

---

> ### Author Response · Authors · 2023-11-16
> **Author Response**
>
> Thank you for your review. We have uploaded an updated version of our paper. Please read our general rebuttal for the major edits we have made.
>
> We would like to respond to your review specifically as follows:
>
> ### Comments
> 1. Regarding the claims: We thank the reviewer for this excellent point. We agree with the reviewer that this paper is more about proposing a class of new regression estimators, than developing a unified theory. We have repositioned our paper and made this point very clear in the updated pdf, which we invite the reviewer to check out.
> 2. We have rewritten the analysis of the graph example entirely in Section 2.1 of the updated pdf and added a comparison with Belkin et al. (2006). Please see the third point of our general rebuttal. Here we use this simple graph example instead of a manifold learning example, because this paper does not contain lots of background of manifold learning, and we want to make sure that most people can understand the paper without too much background. We thank the reviewer for this great comment.
>
> 3. We added an ablation study on the effect of $\eta$ on the tasks we study in Appendix D. We also edited the definition of inverse Laplacian in Example 1. The observation of our ablation study is that: if $\eta$ is too close to 0, then the performance is very low; as long as it is bounded away from 0 (even as small as 0.001), the performance is good and fairly consistent as $\eta$ grows. There does have a slight performance increase as $\eta$ gets larger, but except for one dataset (CS), this increase is not significant. As for a theoretical analysis on the effect of $\eta$, it really depends on what the real unknown $s$ is, so a general result is hard to prove.
> 4. We have added a comparison with Zhai et al. (2023) on page 9 of the updated pdf.
> 5. A table of notation has been added at the beginning of the appendix in the updated pdf. Thank you for this great suggestion.
> 6. We have largely rearranged Sections 3 and 4, adding clear separations between different parts and labeling them with arrow marks. Please kindly let us know if you find that the new version is easier to read, and if there is any further improvement to be made.
> 7. We have edited all table and figure captions in the updated pdf.
> ### Questions
> 1. How to choose a good $K$: This is a very broad and hard open question with quite a long history, which we do not intend to answer in this work, as we have mentioned in the limitations section.
> 2. Regarding $r_t$ in Theorem 1: The key assumption here is that the target smoothness $r_t$ has the same form as the multiscale smoothness $r_{K^p}$, with respect to a certain metric $d_t$. We have clarified this point in the updated pdf, and revised Sections 2.1 and 2.2.
> 3. Regarding centering the kernel: We have added a paragraph about why the centeredness assumption is necessary on page 3 of the updated pdf. We have also added remarks on this point in Section 2.1. Thank you for bringing up this point.
> 4. Analytic complexity: The time complexity is analyzed in Appendix C.
> 5. Experiments of manifold learning: Thank you for this suggestion. Unfortunately, we don’t think we have any space left for the manifold learning experiment in this paper. We can put it in the appendix, but we feel that this experiment is important and should appear in the main body, which is currently over-stuffed. Thus, we intend to put it in a future paper, like what we also say in our response to other reviewers. This future paper will contain a much more extensive empirical study, including not only manifold learning but also modern self-supervised and semi-supervised learning.
>
> We hope that this response has addressed all your questions and concerns. We are very grateful that the reviewer wrote so many constructive comments, and they really helped us to improve this work. We welcome the reviewer to have further discussion with us.

---

> > ### Comment · Reviewer_aR59 · 2023-11-22
> > **Feedback on the rebuttal**
> >
> > I thank the authors for their answers to my comments and those of the other reviewers.

---

> > > ### Author Response · Authors · 2023-11-22
> > > **Thank you**
> > >
> > > Dear Reviewer aR59,
> > >
> > > Thank you for reviewing our paper and providing so many constructive comments! Happy Black Friday!
> > >
> > > Cheers,
> > > Authors

---

### Official Review · Reviewer_j9KG · 2023-11-08

**Soundness:** 3 good
**Presentation:** 3 good
**Contribution:** 3 good
**Rating:** 8
**Confidence:** 3

**Summary:**

The paper presents a universality type result in the context of Spectrally Transformed Kernel Regression (STKR): They show that any target smoothness that preserves multiscale smoothness can be encoded by a spectrally transformed kernel (theorem1).

The goal of the work is use unlabelled data in encoding the target smoothness and then perform (ridge) regression over labelled data. Accordingly, in the case the spectral transform is known, they propose to perform a montecarlo estimation of the STK using labelled as well as unlabelled examples and perform STKR using labelled samples. For this methodology both estimation and approximation errors are bounded (theorem2,3). Implementation details are also discussed.

For the case the transform is not known, they propose following a commonly used two stage process: self-supervised learning using unlabelled samples to learn a d-dim representation. Then, perform ridge regression using the labelled examples under the learnt representation. Again, theorem4,5 provide the estimation and approximation errors. Interestingly, the approximation is shown to be tight (theorem 5& prop5), proving that this methodoloy is no worse than the original proposal of STKR.

Simulations results on few benchmarks are provided.

**Strengths:**

1. I think theorem 1 is insightful and provides perhaps a first universality kind of result connecting smoothness and RKHS.

2. Theorem 5 is also interesting and seems to improve over current bounds i.e., Zhai et.al. 2023.

**Weaknesses:**

questions section

**Questions:**

1. Proposition 1 assumes functions extended to \bar{X}. But what if X itself is a vector space? then this extension basically means we are considering only linear functions over X. So proposition 1 would be restricted to linear functions over X. Am I missing something? Since prop1 is used in theorem 1, perhaps this is an issue?

Minor comments:
1. pg3 "p" is used as exponent as well as likelihood.

---

> ### Author Response · Authors · 2023-11-16
> **Author Response**
>
> Thank you for your review. We have uploaded an updated version of our paper. Please read our general rebuttal for the major edits we have made.
>
> We would like to respond to your review specifically as follows:
>
> 1. Regarding the definition of $\bar{X}$: In the updated pdf, we have added the clarification in Section 2.1 that the “addition” in $\bar{X}$ is an algebraic addition, not the Euclidean addition. Thus, $f(x)$ is a linear functional on $\bar{X}$, but it is not a “linear model”, because the “addition” is defined differently.
> 2. "$p$ is used as exponent as well as likelihood": $dp(x)$ is a shorthand of $d P _X (x)$. It could be better to use a different symbol, but we think that the notation should be clear from context. We have also added a table of notations at the beginning of the Appendix in the updated pdf.
>
> We hope that this response has addressed all your questions and concerns. Once again, we would like to thank the reviewer, and we welcome you to have further discussion with us.

---

> > ### Comment · Reviewer_j9KG · 2023-12-02
> >
> > Thanks for your reponse. My concerns have been addressed. After reading other reviews and the rebuttals, I would like to increase my score.

---

### Official Review · Reviewer_bYf4 · 2023-11-08

**Soundness:** 4 excellent
**Presentation:** 3 good
**Contribution:** 4 excellent
**Rating:** 8
**Confidence:** 2

**Summary:**

This paper studies regression problems in indusctive and transductive settings. The authors establish that if the target function space satisfies a multiscale smoothness assumption, then its reproducing kernel can be represented using a spectral transformed kernel (STK) of on the original space.  Under inductive settings when the spectral transformation function is known, they arrive at convergence guarantees of the learnt model to that of the true one, and the approximation guarantees for using an estimated kernel. Algorithm(s) SKTR-prop estimate the parameters \alpha when s(\lambda) or s^{-1}(\lambda) is analytical, which are in turn used for the prediction. They also show convergence and approximation bounds when s(\lambda) is not known, while constraining the space using regularized inverse laplacian kernel or top-d components from kernel PCA. They illustrate on node classification tasks comparing well against label prop and kernel ridge regression.

**Strengths:**

- As the authors mentioned, though STK, STKR are known before, the theoretical results seem original (to the understanding of the reviewer).

- The regularized laplacian and kernel pca strategies makes good illustration for the applicability of the results, along with the experimental results.

**Weaknesses:**

- Though the flow of the paper is neat, the presentation may be improved which may help with reading such dense set of results. For instance, the definitions of the (target)-smoothness may be done clearer.

**Questions:**

- The laplacian kernel and kernel PCA are known in the literature. In terms of final application, how exactly this work exactly differs in applying them for the problems? Are there other recommendations for choosing the transformation functions for practical use cases ?

- Notation: p is an integer in eqn 7, while a real number in eqn 10. Are they consistent ?

---

> ### Author Response · Authors · 2023-11-16
> **Author Response**
>
> Thank you for your review. We have uploaded an updated version of our paper. Please read our general rebuttal for the major edits we have made.
>
> We would like to respond to your review specifically as follows:
> 1. We have largely improved the writing of our paper and we invite the reviewer to check out the updated pdf. Regarding the definition of the target smoothness, we have largely rewritten Section 2.1, and we believe that the motivation and intuition of how the target smoothness is defined should be much clearer now. If the reviewer has any other suggestions in terms of presentation, please kindly let us know.
> 2. "The laplacian kernel and kernel PCA are known in the literature": As we stated in our second contribution, the novelty of our methods is that they work for general transformation under the inductive setting. Most prior work only focuses on the transductive setting, and to our best knowledge, this work is the first to propose a scalable and general method for the inductive setting. Thus, previous methods such as label propagation need to know the entire graph and all test nodes during training, while our method only requires to know the training nodes and the edges between them, which we demonstrate in our experiments.
> 3. Other recommendations for choosing the transformation function: Ideally, we would want to select the transformation function that matches the oracle transformation function that depicts the “target smoothness”. However, such an oracle rarely exists. The two solutions we provide in Section 4, namely inverse Laplacian and kernel PCA, have been widely used and observed to perform well on a variety of domains. Even better, one can try to learn the best kernel, which we briefly introduced in the related work section, but this is beyond the scope of this work.
> 4. $p$ is real-valued. It is only an integer in Section 3, where we have Assumption 1 that explicitly restricts $s(\lambda)$ to be a polynomial.
>
> We hope that this response has addressed all your questions and concerns. Once again, we would like to thank the reviewer, and we welcome you to have further discussion with us.

---

> > ### Comment · Reviewer_bYf4 · 2023-11-23
> > **Thanks for the clarifications**
> >
> > I thank the authors for the clarifications.

---

### Official Review · Reviewer_SncV · 2023-11-09

**Soundness:** 3 good
**Presentation:** 4 excellent
**Contribution:** 3 good
**Rating:** 8
**Confidence:** 3

**Summary:**

This paper considers spectrally transformed kernel regression, a way of performing unsupervised or semi-supervised learning with kernels. Unlabeled data can be leveraged to obtain better estimates of a spectral transformation of the kernel, which can then be used for kernel regression. The paper proposes scalable algorithms to implement STKR for two different types of spectral transformations in the inductive setting, which is more practical than the transductive setting that is often used. Moreover, it provides a characterization of target smoothness of functions and provides theoretical guarantees on how fast such smooth functions can be learned using STKR.

**Strengths:**

Originality: While STKR itself is apparently well-known, the authors advance the understanding of this method both in algorithmical and theoretical aspects.
Quality: While I have not verified the proofs in the appendix except for Proposition 1, the quality of the results seems to be good.
Clarity: The paper is well-written, in particular it provides many useful comments on the motivation and interpretation of the theoretical results.
Significance: I am not familiar with previous literature on semi-supervised learning with kernels, but judging from the description in the paper, the existence of theoretical guarantees seems to be an advantage compared to previous work, and the introduction of practical algorithms for the inductive setting also appears to be relevant. I am not sure about the practical relevance due to limited experimental results.

**Weaknesses:**

The experiments are rather limited. First, they are limited to node classification in graphs, which is certainly an interesting class of problems, but it leaves me wondering whether the proposed method, despite its generality, is useful on other types of problems. Moreover, the results are only compared to one competitor method, label propagation. It would be interesting to see
- how the proposed method compares to label propagation in terms of (training and) inference time,
- if there is another known feasible inductive method, how this method compares to STKR,
- how the proposed method compares to other (non-kernel) methods, surely there have to be some deep learning methods for these problems?
Of course, the proposed method is already relevant through its theoretical analysis, but a better experimental evaluation would help to better understand the practical relevance.

The assumptions used in the theoretical analysis appear to be relatively strong compared to what I know from the supervised learning literature, at least when thinking about continuous input spaces rather than graphs. For example, if the base kernel was a Sobolev kernel, the associated RKHS would have to have a smoothness $s > d/2$, which might be unrealistic in high dimensions. In this case, many theoretical results such as the ones in Fischer & Steinwart (2020) allow the target function to lie in an interpolation space that is larger than the RKHS. In the case of this paper, I am wondering whether such an assumption could be sensible, as having higher smoothness than the base kernel appears to be crucial.

**Questions:**

**Questions**:
In Example 1, is $K^0(x, x')$ just $\mathrm{tr}(K)$?

Is there a setting in which a provable benefit of semi-supervised learning over supervised learning can be proven?

Could label propagation be made inductive by using it to label the unlabeled "other" set transductively, and then using these pseudo-labels to fit a supervised kernel regression method ("distillation")? How would you expect this to compare to STKR in terms of runtime and accuracy?

In the proposed algorithms, could it be beneficial to use more advanced linear system solvers like the CG method instead of Richardson iteration?


**Major comments**:
Paragraph before Proposition 1: The definition of $\overline{\mathcal{X}}$ is very imprecise. It looks like you are summing inside of $\mathcal{X}$, even though $\mathcal{X}$ might not be a vector space. It is also not clear whether the sums are allowed to be infinite. On the one hand, I would assume them to be finite because you didn't impose any assumptions on the $a_i$. On the other hand, you say later that $\overline{\mathcal{X}}$ is a Banach space, but it wouldn't be complete if you only allowed finite sums. I assume that you would want $\overline{\mathcal{X}}$ to be the space of finite signed measures on $\mathcal{X}$, then the norm / distance could be defined through kernel mean embeddings, and $f(\mu) = \mu_f$ would be the pushforward measure.

Proposition 1 seems to be mathematically elegant, at least when reformulating it with measures as discussed in the previous comment. However, the notion of this "alternative" Lipschitz constant seems rather unintuitive to me, and I am not sure what benefit it brings compared to just using the RKHS norm.

The centeredness assumption on the base kernel is worrying me, as I do not fully understand its practical and theoretical implications. Where is this assumption necessary? I understand that it is necessary for Proposition 1. Is the purpose to make the smoothness notion shift invariant and not having to worry about the constant part of the target function that is not covered by the smoothness $r_t(f)$?

In Theorem 1, it was not fully clear to me whether the assumptions above Section 2.2 are also assumed in the theorem; for example, the assumption $\mathcal{H}_t \subset \mathcal{H}_K$ is repeated in the theorem.

**Minor comments**:
- Section 2, page 2: You write $dp(x)$ in the integral multiple times, but you did not define $p$. You could just write $dP_{\mathcal{X}}(x)$.
- Footnote on page 2: Since $L^1(P_{\mathcal{X}}) \subseteq L^2(P_{\mathcal{X}})$, it is not necessary to assume boundedness for the existence of the expectation. Is boundedness also required for other things?
"$\mathcal{H}_{K_p}$ are also known as interpolation Sobolev spaces": I think from the definition these are known as power spaces, see e.g. [1], although they are often (?) identical to interpolation spaces. In the case where the RKHS of $K$ is a Sobolev space, these are usually also Sobolev spaces, but you did not assume this.
- Small typesetting observation: In Proposition 1, $\overline{\mathrm{Lip}}$ is italic while it is not italic above. Maybe you used \text{Lip} instead of other commands such as \DeclareMathOperator or \operatorname or \mathrm?
- Before Section 2.2, you use the absolute value on $\overline{\mathcal{X}}$, is this supposed to be the total variation norm?
- In Section 3 (page 6), you should explain that $s^{-1}(\lambda) = 1/s(\lambda)$ (if I understand correctly), or perhaps directly write $s(\lambda)^{-1}$. I thought that $s^{-1}$ was the inverse of $s$ until I noticed that it didn't fit with the implementation.

[1] https://link.springer.com/article/10.1007/s00365-012-9153-3

**Summary of discussion:**
While the authors did not extend their experiments much, the theoretical analysis is interesting in its own right and the authors fixed some technical issues and improved some explanations, so I raised my score from 6 to 8.

---

> ### Author Response · Authors · 2023-11-16
> **Author Response (1/2)**
>
> Thank you for your review. We have uploaded an updated version of our paper. Please read our general rebuttal for the major edits we have made.
>
> We would like to respond to your review specifically as follows:
> 1. Experiments: We agree that a comprehensive empirical investigation would be an interesting study. Since our work focuses on developing theory and the paper in its current form is already very dense, we believe that it would be better to defer a more extensive empirical study on STKR to a future paper, especially within the context of modern self-supervised and semi-supervised learning. This is definitely something we plan to work on. Regarding your specific points:
>     - STKR-Prop vs Label-Prop: We have pointed out in the “implementation” paragraph in Section 3 that STKR-Prop (with inverse Laplacian) has the same time complexity as Label-Prop on a graph under the transductive setting. The difference here is that STKR-Prop uses the inductive version of inverse Laplacian, while Label-Prop uses the transductive version, whose difference and connection are discussed in Appendix B.3.
>     - Other feasible inductive methods: To the best of our knowledge, this is the first work that proposes an efficient and scalable STKR implementation for the inductive setting. We have also pointed out that for the inductive setting, the inference of STKR-Prop is much faster than previous methods, such as Chapelle et al. (2002).
>     - Non-kernel methods: There are deep learning methods for these problems (we mentioned them in the second contribution in the introduction section), but these methods do not have rigorous guarantees. In contrast, the goal of this work is to propose an approach that is general, scalable, and comes with provable guarantees. We intend to do a more comprehensive empirical study of our method and deep learning methods in future work.
> 2. The reviewer asked if the assumption $H_t \subset H_K$ in Theorem 1 can be relaxed. We thank the reviewer for this great question. The answer is yes.
>     - In theory, we can use $H_{K^p}$ for $p < 1$, up to $p \ge 0$. The tricky thing here is that in this case, $H_{K^p}$ is not necessarily an RKHS, and consequently $H_t$ could no longer be an RKHS, so we cannot explicitly write out the kernel $K_s$ as in Theorem 1. But with the extra assumption that $H_{K^p} (p < 1)$ is still an RKHS, our results can be easily extended. We made a new remark on this point after Theorem 1 in the updated pdf.
>     - Implementation-wise, using $H_{K^p}$ for $p < 1$ makes no difference for kernel PCA (Section 4), because the optimal d-dimensional representation still consists of the top-d eigenfunctions. And this does not apply to Section 3, because $s(\lambda) = \lambda^p$ with $p < 1$ is not differentiable at $\lambda = 0$, so it violates Assumption 1.
>
> ### Questions
> 1. Regarding $K^0$: Thank you for pointing out this ambiguity. We have added clarifications of the definition of $K^0$ in the updated pdf. It corresponds to the STK with $s(\lambda) = \lambda^0 (\lambda > 0)$.
> 2. Provable benefit over supervised learning: In Section 2.1 of the updated pdf, we added an intuitive explanation of why STKR is better than supervised learning using our graph example. The TL;DR is that the procedure of constructing the STK exploits $P _X$, and the unlabeled data can provide additional information about $P _X$. In our experiments, we have also shown that only supervised KRR is not enough for real graph tasks. Moreover, in Appendix D we have an ablation study on the effect of $p _{test}$, and this study shows that more unlabeled samples leads to better performance.
> 3. Whether inductive Label-Prop is possible: Label-Prop can be seen as a special case of STKR-Prop with $s(\lambda) = 1 / (1 - \eta \lambda)$. The first problem is that $H_{K _s}$ with this $s$ is not an RKHS unless it is finite-dimensional, because $s(0) > 0$. So we cannot prove the same guarantees for it. The second problem is that to make it work under the inductive setting, we need to estimate $K^0$, which is (a) not doable if $K _s$ is infinite-dimensional, and (b) very hard to estimate if $K _s$ is finite-dimensional because essentially one would need to estimate all eigenfunctions with positive eigenvalues. An viable alternative would be to add an extra $L^2$ regularizer on top of STKR-Prop with inverse Laplacian. This could approximate the inductive Label-Prop though not exactly equivalent, and we don’t see why this would be significantly better than just STKR-Prop with inverse Laplacian.

---

> > ### Author Response · Authors · 2023-11-16
> > **Author Response (2/2)**
> >
> > 4. Whether one can use other linear solvers: Absolutely yes. With Richardson iteration, we showed that both training and inference of STKR-Prop can be very scalable. Our goals here are (i) to demonstrate the existence of a good scalable solver (Richardson), and (ii) to show how to do training and inference in a scalable way. We believe that for the second case where $s^{-1}$ is simple (such as the inverse Laplacian), the equation $M \theta = \tilde{y}$ (Eqn. (28)) is a major contribution because it offers an indirect but scalable way of finding the solution to an otherwise hard-to-solve equation. But once we derive this equation, we can use whatever linear solver we find that works the best for us, and there could be solvers that are faster or more numerically stable than Richardson iteration. We have added this new remark at the end of the implementation paragraph of Section 3.
> >
> > ### Major comments
> > 1. Regarding $\bar{X}$: Thank you for pointing out this obscurity. We have revised the definition of $\bar{X}$ in the updated pdf.
> >
> > 2. Regarding your question about the Lipschitz constant:
> >     - One main goal of Section 2.1 is to formally characterize the type of “smoothness” KRR and STKR are imposing on the predictor, which is crucial as it reflects the prior knowledge about the target function that our semi-supervised or self-supervised learning method depends on. For power spaces $H _{K^p}$, Fischer & Steinwart (2020) showed their connection to Sobolev spaces in their Section 4, and that is why they are also called interpolation Sobolev spaces. So essentially, STKR with $K^p$ is imposing a type of Sobolev smoothness, but this does not give us a clear picture of what smoothness a general $K _s$ is imposing.
> >     - Our Lipschitz characterization in Section 2.1 provides another perspective, which is based on the diffusion maps introduced in Coifman & Lafon (2006). What we show here is that: $\lVert f \rVert _{H _{K^p}}$ is equivalent to the Lipschitzness of $f$ w.r.t. diffusion distance $d _{K^p}$ defined by Coifman & Lafon. So for the unknown target smoothness that the target function satisfies, we naturally assume that it is the Lipschitzness of $f$ w.r.t. another metric. This shows that the target smoothness which $K _s$ is imposing has a similar nature to the Sobolev smoothness imposed by $K^p$.
> >
> > 3. Regarding the centeredness assumption: please see the second point in our general rebuttal, as well as the new Section 2 of the updated pdf.
> >
> > 4. We have revised the sentence before Theorem 1. Please let us know if this helps clarify the premise of Theorem 1.
> > ### Minor comments
> > 1. $dp(x)$ is a shorthand for $d P _X(x)$. We have clarified this point in Section 2 in the updated pdf.
> > 2. This footnote is from an older draft and is obsolete. We have removed it in the updated pdf. $H _{K^p}$ is called power spaces as well as interpolation Sobolev spaces. We have clarified this point in Section 2 of the updated pdf.
> > 3. Typesetting: Thanks for this suggestion. The problem is now fixed.
> > 4. We couldn’t find which absolute value of $\bar{X}$ the reviewer is referring to. Maybe this problem has already been fixed in the updated pdf. Please let us know if the problem is still there.
> > 5. We have added a clarification of the meaning of $s^{-1}$ in Example 1 of the updated pdf.
> >
> > We hope that this response has addressed all your questions and concerns. We are very grateful that the reviewer wrote such a detailed, thoughtful and constructive review. We welcome the reviewer to have further discussion with us.

---

> > > ### Comment · Reviewer_SncV · 2023-11-16
> > > **Response to second part of the answer**
> > >
> > > **Regarding questions**:
> > > 4. Thanks for adding the remark.
> > >
> > > **Regarding major comments**:
> > > 1. Your revision does not address my remarks. I think you should use a measure-theoretic formulation. If it is unclear what I mean by the measure-theoretic formulation, I can try to write it more formally.
> > > 2.
> > > - The whole Sobolev smoothness discussion does not apply to the graph kernels that you mostly consider. (Additionally, the connection of $H_{K^p}$ to Sobolev spaces should only hold in the case where $P_X$ has an upper- and lower-bounded density, which is probably a setting where semi-supervised learning is not very helpful.)
> > > - Okay, but my point is that when you apply $f$ to measures, the resulting notion of a Lipschitz constant is rather unintuitive.
> > > 4. It is still there, in the second-last line before Section 2.2, in the definition of $d_t(x, x')$. Note that for $x \in \overline{\mathcal{X}}$, $f(x) \in \overline{\mathcal{X}}$.

---

> > > > ### Author Response · Authors · 2023-11-17
> > > > **[Author Response] Discussion**
> > > >
> > > > Thank you for your follow-up discussion! We really appreciate that the reviewer is so responsive and gives so much care for our paper. We would like to answer your new questions as follows:
> > > >
> > > > 1. Relaxing $f^* \in H _{K _s}$:
> > > >     - Assumption 3 is used in Theorems 2 and 3: In Theorem 2, the bound holds “for all $f^*$ satisfying Assumption 3”; Theorem 3 contains $\frac{ \lVert y \rVert_2^2 }{n}$, and with assumptions 3 and 4, we can prove an $O(B^2 + \sigma^2 + L^2)$ for it, which we mentioned in the remark of this theorem.
> > > >     - We assume $f^* \in H _{K _s}$, because $K _s$ is assumed to have captured the “target smoothness” that $f^*$ possesses. One possible way of relaxing this is to use the power spaces in Fischer & Steinwart (2020) with a smaller power, and our results shouldn’t be too hard to extend (similar to how we relaxed $H_t \subset H_K$).
> > > > 2. Whether unlabeled data helps reduce the sample complexity of supervised learning: The reviewer asked if we can prove a theorem that with sufficient unlabeled samples, we can improve the minimax optimal learning rate w.r.t. n. Thank you for this excellent question! This is indeed a super interesting question. We believe that the answer is yes. The empirical evidence is that with a huge amount of unlabeled data, modern foundation models are able to use very few-shot learning to achieve SOTA performances on downstream tasks, which otherwise would require a lot more samples. Of course, since we have yet to prove a rigorous theorem, we cannot 100% say yes at this point, but we will definitely study this problem in a future work. We have added this question to the open problem section.
> > > > 3. Inductive Label-Prop: The reviewer asked if in practice, one can implement a version of inductive Label-Prop, by first “pseudo-labelling” the unlabeled samples using Label-Prop, and then running KRR on labeled and unlabeled samples. This is a good idea and is indeed implementable. We implement this idea and compare it to inductive STKR with inverse Laplacian on five datasets. The results are the following:
> > > > | Dataset      | STKR-Lap |  Pseudo-label   |
> > > > | ----------- | ----------- | ----------- |
> > > > | Computers      |  $76.64 \pm 2.31$     |  $ 73.43 \pm 4.52 $    |
> > > > | Cora   | $ 67.41 \pm 7.96 $        |  $ 70.37 \pm 7.81$    |
> > > > | DBLP  | $ 65.25 \pm 4.28 $   | $ 65.25 \pm 4.54 $    |
> > > > | Photo  |  $ 84.87 \pm 5.66 $  | $ 80.13 \pm 7.88$   |
> > > > | Physics   |  $ 82.56 \pm 5.26 $   | $ 84.42 \pm 4.71 $   |
> > > >
> > > > The observation is that on some datasets STKR-Lap is better, and on some other datasets Pseudo-label is better, but neither is significantly better than the other one. Currently there is no theoretical guarantee for Pseudo-label, though it won’t be surprising if one can be proved.
> > > >
> > > > 4. Definition of $\bar{X}$: The reviewer suggested defining $\bar{X}$ as the space of finite signed measures on $X$. We apologize for our earlier misunderstanding of your remark. We have adopted your definition and updated Section 2.1. Please let us know if the current definition is correct and rigorous.
> > > >
> > > > 5. Lipschitzness over the space of measures:
> > > >     - First, regarding your fourth comment, we would like to clarify that (using the notations in the new version) $\bar{f}(\mu) = \int _{x \in X} f(x) d \mu(x)$ for $\mu \in \bar{X}$. Thus, $\bar{f}$ is a linear functional over $\bar{X}$, and $\bar{f}(\mu) \in \mathbb{R}$, not in $\bar{X}$. We have added this clarification in the new version.
> > > >     - Second, we provided an intuition on the Lipschitzness we defined in the new version. Essentially, our Lipschitz regards a function $f$ to be smooth, if its average value w.r.t. a measure $\mu$ does not change too much when $\mu$ changes by a little bit.
> > > >
> > > >
> > > > We have edited the paper according to your new comments. Please let us know if you have any further questions.

---

> > > > > ### Comment · Reviewer_SncV · 2023-11-17
> > > > > **[Reviewer Response] Discussion**
> > > > >
> > > > > 1.
> > > > > - Thanks, I overlooked it.
> > > > > - Good.
> > > > > 2. Good.
> > > > > 3. Thanks, this is interesting.
> > > > > 4. This looks much better. Technically, you need a $\sigma$-Algebra to define a space of measures. If you have a suitable topology on $X$, you could take the Borel $\sigma$-algebra. Maybe you will need that your $\sigma$-Algebra contains single-point sets. Maybe it is also sufficient to take the $\sigma$-algebra of countable and co-countable sets.
> > > > >
> > > > > Notationally, using $\int f(x) d\mu(x)$ instead of $\int f(x)\mu(x)$ would be more standard, but I see that the space is tight on this line.
> > > > >
> > > > > 5.
> > > > > - Yes, sorry, this was my mistake.
> > > > > - Maybe this helps, but I have a poor intuition on which measures are close in the (MMD) distance. Clearly, the Lipschitz constant w.r.t. measures can be much larger than the standard Lipschitz constant, so there must be some measures that are quite close in MMD that wouldn't be close w.r.t. some other metrics like the Wasserstein distance. But there is probably not much that you can do about this.
> > > > >
> > > > > I thank the authors for their responsiveness. In response to the improvements to the paper, I am raising my score to 8.

---

> > ### Comment · Reviewer_SncV · 2023-11-16
> > **Response to first part of the answer**
> >
> > Thank you for your detailed response. Please find some answers below. Note that I am not yet updating my main review and score, since some questions are still open (see below).
> >
> > **Regarding weaknesses**:
> >
> > 1. I agree that an extensive experimental evaluation, while interesting, is a lot of work and not necessary for this paper. With the current evaluation, the focus of the paper is more on the theoretical contribution, and the theoretical contribution seems good to me.
> >
> > 2. Thank you for the clarification. My question was not specifically about Theorem 1. I forgot to mention this in the review, but from my point of view, Theorem 1 is just rephrasing an assumption in terms of another assumption (preserves relative smoothness) that is not necessarily more intuitive or plausible to me (same issue as with Proposition 1). I was rather thinking about relaxing the assumption $f^* \in H_{K_s}$ in Assumption 3 for Theorems 2 and 3 etc. (I don't actually see Assumption 3 being required in Theorem 2 and 3?!) Ideally (but out of the scope of this paper) one could make some statements as to whether there can still be a provable benefit over supervised learning in this case -- certainly there would be no benefit for $p=1$. See also my answer to question 2.
> >
> >
> > **Regarding questions**:
> >
> > 2. Thank you, I understood the intuition but it is probably helpful for some readers that you added an explanation. My question was aimed at the form of the theoretical results. One could say that on a class $\mathcal{P}$ of distributions, semi-supervised learning has a provable benefit over supervised learning if a semi-supervised learning method provably achieves a convergence rate in terms of $n$ (with sufficiently large $m$) that is faster than the minimax-optimal rate on the same distribution class in the supervised setting. For example, denote by $K^p_{P_X}$ the $p$-th power kernel w.r.t. $P_X$ and by $f_P^*$ the target function w.r.t. $P$. Then, you could consider a class of distributions like $\mathcal{F} =$ set of distributions $P$ such that $f_P^* \in H_{K^p_{P_X}}$ and additional assumptions hold.
> >
> > 3. I understand that a theoretical analysis is difficult. However, I asked from a practical viewpoint whether Label-Prop can be used transductively to label the unsupervised set, such that one can obtain an inductive kernel model by performing KRR on top of the generated pseudo-labels.

---

### Official Review · Reviewer_nUyP · 2023-11-10

**Soundness:** 3 good
**Presentation:** 4 excellent
**Contribution:** 4 excellent
**Rating:** 8
**Confidence:** 3

**Summary:**

The paper focuses on utilising unlabelled data within the framework of Spectrally Transformed Kernel Regression (STKR). The rough idea is that unlabelled data can be used to infer the smoothness of the kernel. This is achieved by considering spectrally transformed (Mercer) kernels:
$$ K_s(x,y) = \sum_{i=1}^\infty s(\lambda_i) \psi_i(x)\psi_i(y). $$
The form of the transformation $s(\lambda)$ influences the smoothness of the kernel and this is what is learned from the unlabelled data. The paper investigates this theoretically by assuming the target function $f^*$ has a target smoothness constraint $f^* \in \mathcal{H}_t$ and assumes the target smoothness is at least as smooth as the base kernel $\mathcal{H}_t \subset \mathcal{H}_K$.

Theorem 1 states that under certain conditions, $\mathcal{H}_t$ is the RKHS of a spectrally transformed kernel. This can be viewed as an existence result of a suitable $s$ and so motivates the rest of the paper.

Next, the authors consider the theoretical implications of this and construct algorithms of STKR in two different settings: (1) $s$ is known and (2) $s$ is unknown. Setting (2) is the real-world situation and so is of more practical benefit.

Setting (1):

The problem here is that $K_s$ may not be computable. Therefore, the authors utilise the unlabelled data to estimate $K_s$ by a Monte-Carlo approximation of the kernels $K^p$, which is $K_s$ with $s(\lambda) = \lambda^p$ (it is assumed that the true $s$ is expressible as a power series). Under this scenario, it is shown that KRR is minimax optimal under exact evaluation of $K_s$ (Theorem 2). The approximation error of using $\hat{K}_s$ over $K_s$ is also shown to be bounded (Theorem 3).

Setting (2):

Firstly, the authors consider $s$ as the inverse regularised Laplacian. This is motivated by the fact that this $s$ has been shown to work well empirically in prior work and can be analysed theoretically.

Next, the authors consider a kernel where the first $d$ eigenfunctions are learned through kernel PCA on the unlabelled data. This is equivalent to STKR with $s$ as a truncation function, all eigenvalues smaller than $\lambda_d$ are eliminated. The authors provide theoretical guarantees that lower bound and upper bound the worst and best case errors respectively.

Main Contributions:
- Establishing STKR as a principled way to utilise unlabelled data, showing that under certain smoothness conditions, a target function must be smooth with respect to a certain Spectrally Transformed Kernel (STK).
- Implementing practicable STKR algorithm in an inductive setting with general transformations, which is more practical than previous transductive approaches. This implementation is scalable, has closed-form formulas for the predictor, and comes with statistical guarantees.
- Developing rigorous theoretical bounds for this general inductive STKR, proving estimation and approximation error bounds, some of which are tight or near-tight.

**Strengths:**

Overall, I find the paper very good. The authors provide a theoretical framework that unifies previous works of incorporating unlabelled data into learning algorithms. The author's approach, to my knowledge, has not been considered before.

The theoretical results are very interesting. The methodology developed is motivated and justified from the author's theoretical work. The authors developed a rigorous theoretical underpinning for inductive STKR and provide tight statistical learning bounds for prediction errors. This addresses the issue of generalisability in STKR, by offering strong statistical guarantees. Not only this, the methodology is practicable and this is evidenced from the experimental results and computational complexity calculations.

The paper is very well written and easy to understand. A lot of background material is presented, which contextualises the work. The appendices are also well written and include extensive discussions on related works.

**Weaknesses:**

The main weakness, in my view, is the lack of a comprehensive empirical investigation.

While the paper does explore the effect of different transformations $s(\lambda)$ and provides a comparison to other methods such as label propagation and kernel PCA, it does not delve deeply into the conditions or characteristics of datasets that would lead to STKR's superior performance. It is probably very difficult to do this theoretically, but a more comprehensive empirical investigation could lead to insights. This could be seen as a limitation in fully understanding the potential and limitations of STKR.

Also, although the experiments cover several datasets, they are all within the realm of graph node classification. Expanding to other types of data could demonstrate the generality of the approach.

While the paper mentions the efficiency of the STKR-Prop, a detailed computational complexity analysis, especially in comparison with other methods, could add more depth to the evaluation of the methodology.

Finally, I don't see where the regularisation term $\eta$ in SP-Lap is specified within the paper.

**Questions:**

Can you discuss when the conditions of theorem 1 are satisfied? The conditions being that if $r_{K^p}(f_1) \geq r_{K^p}(f_2)$ for all $p\geq 1$ then $r_t(f_1) \geq r_t(f_2)$. Could there be situations where this condition is violated and so the resulting theory doesn't hold? Does this have any practical consequences?

Could proposition 3 be generalised for other transformations $s(\lambda)$? With $s(\lambda) \geq \lambda$?

Do you have any intuition as to why STKR performance degrades as $p_{test}$ increases? Does this imply scalability issues?

Will the code used in the experiments be provided for reproducibility?

---

> ### Author Response · Authors · 2023-11-16
> **Author Response**
>
> Thank you for your review. We have uploaded an updated version of our paper. Please read our general rebuttal for the major edits we have made.
>
> We would like to respond to your review specifically as follows:
> 1. Empirical investigation: We agree that a comprehensive empirical investigation would be an interesting study. Since our work focuses on developing theory and the paper in its current form is already very dense, we believe that it would be better to defer a more extensive empirical study on STKR to a future paper, especially within the context of modern self-supervised and semi-supervised learning. This is definitely something we plan to work on.
> 2. Computational complexity and comparison with other methods: We have an analysis of the computational complexity of STKR in Appendix C. Regarding its comparison with Label-Prop, we have pointed out in the “implementation” paragraph in Section 3 that STKR-Prop (with inverse Laplacian) has the same time complexity as Label-Prop on a graph under the transductive setting. The difference here is that STKR-Prop uses the inductive version of inverse Laplacian, while Label-Prop uses the transductive version, whose difference and connection are discussed in Appendix B.3. As for the inductive setting, to our best knowledge this is the first work that proposes an efficient and scalable STKR implementation for the inductive setting, so there isn’t really a good baseline to compare to. We have pointed out that for the inductive setting, the inference of STKR-Prop is much faster than previous methods, such as Chapelle et al. (2002).
> 3. Values of $\eta$: In the updated pdf, we added the list of hyperparameters we used in our experiments in Appendix D, which includes $\eta$. Note that the best hyperparameters are always selected using a separate validation set.
> 4. Regarding your first question about the assumption of Theorem 1, our belief is that the “relative smoothness preserving” condition of Theorem 1 is a very weak one, because the premise of this condition $r _{K^p}(f _1) \ge r _{K^p}(f _2)$ needs to hold **for all** $p \ge 1$ (if the premise of a condition is strong, then the condition itself is weak). Thus, it is quite hard to find a realistic example that violates this condition. The only two possible examples we can think of are: (i) if the similarity does not have transitivity (that is, (a, b) are similar and (b, c) are similar do not imply (a, c) are similar), or (ii) if the base kernel K is really bad and has no alignment with the target smoothness. For (i), there could be some real-world tasks where similarity has no transitivity, which we are not really familiar with; for (ii), this pertains to how to choose the right kernel. As we pointed out in the limitations section, this work does not discuss how to choose the right base kernel, which is itself a very broad and hard open question that is beyond the scope of this work.
> 5. Regarding your second question about generalizing $s(\lambda)$, please note that $s(\lambda) = O(\lambda)$, so $s(\lambda) \ge \lambda$ is possible.
> 6. Regarding your third question about $p _{test}$: When $p _{test}$ increases, the set of unlabeled samples (“other” set in Appendix D.1) becomes smaller (as test samples are unseen during training in the inductive setting). Therefore, we have fewer unlabeled samples to approximate $K_s$ in Section 3, or perform kernel PCA in Section 4. This naturally leads to performance degradation. However, We don’t believe that this would cause a scalability issue because, in modern practice, we usually have a huge amount of unlabeled training data.
> 7. Please find our code in the attached supplementary material.
>
> We hope that this response has addressed all your questions and concerns. The reviewer raises some great questions which we really appreciate, and we welcome the reviewer to have further discussions with us.

---

### Author Response · Authors · 2023-11-16
**[Author Response] General Rebuttal**

We would like to thank all our reviewers for their time and effort in reviewing our paper. We really appreciate that we have got excellent reviewers who provide thoughtful comments and helpful feedback. Based on the reviews, we have made lots of improvements to our paper, and have uploaded a new pdf, which the reviewers can find on openreview. Apart from adding colors to all hyperlinks and a number of typo fixes, we have made several major edits and here is a summary:
### List of Major Edits

1. The contribution of this paper has been rephrased (credit to Reviewer aR59). In the previous version, this paper was positioned as “developing a unified theory”, which is not the precise depiction of the contribution. In the updated version, this paper is repositioned as “providing a new class of general and scalable STKR estimators able to leverage unlabeled data”.
2. An explanation of why the centeredness assumption is necessary has been added in Section 2, as well as after Proposition 1 (credit to Reviewer SncV and Reviewer aR59). Generally speaking, we view the smoothness and the scale of a function $f$ as two orthogonal axes, because the smoothness studied in this work pertains to inter-sample similarity. This implies two important properties: (i) If $f _1$ and $f _2$ differ by a constant a.e., then they are viewed as equally smooth; (ii) $f$ and $2f$ are viewed as equally smooth. For (i), the centeredness assumption is necessary; For (ii), we define $r _{K^p}(f)$ as a ratio so that it is homogeneous. Note that classical smooth function classes like Sobolev spaces are different (the $L^p$ norm is a part of the Sobolev norm), and they do not decouple smoothness from the scale. Moreover, in practice the centering step is not necessary (though it is often recommended in kernel PCA).
3. We rewrote the analysis of the graph example in Section 2.1 entirely, and discussed the connection between STKR and Belkin et al. (2006) (credit to Reviewer aR59). Intuitively, unlabeled samples are useful because they can provide additional information about $P _X$. However, the definition of the base kernel $K$ need not depend on $P _X$. And if it does not depend on $P _X$, then KRR exploits no more information about $P _X$ than supervised learning, and that’s why the unlabeled samples are useless in KRR as per Representer Theorem. To solve this, Belkin et al. (2006) explicitly introduced an extra $\lVert f \rVert _I^2$ regularizer that reflects the structure of $P _X$. Likewise, STKR also exploits $P _X$, in its procedure of constructing the STK, and that is where the unlabeled samples become useful. For example, if the STK is $K^2$, then since $K^2$ is essentially an integral over $P _X$, it exploits $P _X$. And indeed, in Section 3, we approximate $K^2$ via a Monte-Carlo approximation with the labeled and unlabeled data, and that is how the unlabeled data becomes useful again despite the Representer Theorem.
4. We added a remark after Theorem 1 that the assumption $H _t \subset H _K$ can be relaxed (credit to Reviewer nUyP and Reviewer SncV). The main caveat is that for $p _0 < 1$, $H _{K^{p _0}}$ might not be an RKHS. But if we additionally assume that it is an RKHS, and $H _t \subset H _{K^{p _0}}$, then we can still prove Theorem 1, with (ii) changed to $s(\lambda) \le M \lambda^{p _0}$.
5. The definition of $\bar{X}$ and the explanation of $r _t(f)$ in Section 2.1 were revised (credit to Reviewer SncV and Reviewer j9KG).
6. The structure of Sections 3 and 4 has been largely rearranged. Also to make these two sections easier to read, we added bullet points for Method, Results overview, and Implementation (credit to Reviewer aR59).
7. “Analytic transformation” is rephrased as “polynomial transformation”, which is more precise.
8. In Example 1, we edited the definition of the inverse Laplacian. Now, $\eta \in (0, \lambda _1^{-1})$. Previously, we assumed $\lambda _1 \le 1$, and $\eta \in (0,1)$.
9. We added a comparison to Zhai et al. (2023) on page 9 (credit to Reviewer aR59).
10. A table of notations was added at the beginning of the Appendix (credit to Reviewer aR59).
11. We rewrote the discussion paragraph in Appendix B.2, and added a discussion on the connection between $r _{K^p}(f)$ and the Poincaré constant.

12. All table and figure captions have been edited (credit to Reviewer aR59).

13. In Appendix D, we included a list of hyperparameters we used in our experiments (credit to Reviewer nUyP). Note that the best hyperparameters are always chosen on separate validation sets.

14. In Appendix D, we added an ablation study on the effect of $\eta$ (credit to Reviewer aR59).

Other minor edits can be found in our individual responses to the reviewers. Once again, a great thank you to all reviewers! We also welcome all reviewers to have further discussions with us. Just a kindly reminder that **the discussion deadline is Nov 22nd.**

---

### Meta-Review · Area_Chair_rqMD · 2023-12-05

**Metareview:**

The paper presents results for unsupervised and semi-supervised learning with kernels. The reviewers are unanimously positive but highlight a few short comings that would be worth addressing in the final version. For example, it would be useful to cite and discuss Belkin et al. 2006, to comment on adaptive estimation where the unknown function does not lie in the RKHS etc.

**Justification For Why Not Higher Score:**

The paper is a solid contribution to the literature but, according to the reviewers, falls into the category of incremental improvements and is not breaking substantial new ground.

**Justification For Why Not Lower Score:**

The paper contains interesting ideas that are worth being highlighted at the conference.

---

### Decision · Program_Chairs · 2024-01-16

Accept (spotlight)